# Optimization for Neural Operators can Benefit from Width

**Pedro Cisneros-Velarde** [* 1]   **Bhavesh Shrimali** [* 2]   **Arindam Banerjee** [2]

## Abstract

Neural Operators that directly learn mappings between function spaces, such as Deep Operator Networks (DONs) and Fourier Neural Operators (FNOs), have received considerable attention. Despite the universal approximation guarantees for DONs and FNOs, there is currently no optimization convergence guarantee for learning such networks using gradient descent (GD). In this paper, we address this open problem by presenting a unified framework for optimization based on GD and applying it to establish convergence guarantees for both DONs and FNOs. In particular, we show that the losses associated with both of these neural operators satisfy two conditions—restricted strong convexity (RSC) and smoothness—that guarantee a decrease on their loss values due to GD. Remarkably, these two conditions are satisfied for each neural operator due to different reasons associated with the architectural differences of the respective models. One takeaway that emerges from the theory is that wider networks benefit optimization convergence guarantees for both DONs and FNOs. We present empirical results on canonical operator learning problems to support our theoretical results and find that larger widths benefit training.

## 1. Introduction

Replicating the success of deep learning in scientific computing such as developing neural partial differential equation (PDE) solvers, constructing surrogate models, and developing hybrid numerical solvers, has recently captured the interest of the broader scientific community (Kutz & Brunton, 2024; Kovachki et al., 2023). In relevant applications to scientific computing, we often need to learn mappings between input and output function spaces. Neural operators have emerged as the prominent class of deep learning models used to learn such mappings (Lu et al., 2021). They have become a natural choice for learning solution operators of parametric PDEs and of inverse problems where multiple evaluations are needed under different parameters of the problem. Two of the arguably most widely adopted neural operators are Deep Operator Networks (DONs) (Lu et al., 2021; Wang et al., 2021b) and Fourier Neural Operators (FNOs) (Li et al., 2021a;b).

The fundamental idea of a neural operator is to parameterize mappings between function spaces with deep neural networks and proceed with its *learning*, i.e., *optimization*, as in a standard supervised learning setup. However, contrary to a classical supervised learning setting where we learn mappings between two finite-dimensional vector spaces, here we learn mappings between *infinite-dimensional* function spaces. While there exist results on the universal approximation properties of DONs and FNOs (Deng et al., 2022; Kovachki et al., 2021), to the best of our knowledge, there are *no formal optimization convergence results* for the training of these two popular neural operator models.

To address this open problem, in this paper, **we establish formal optimization convergence guarantees for learning DONs and FNOs with gradient descent (GD)**. To achieve this, we first **propose a general framework that ensures the optimization of any loss using GD as long as two conditions are satisfied across iterations**. The first condition is based on restricted strong convexity (RSC), a recently introduced alternative (Banerjee et al., 2023a) to the widely used neural tangent kernel (NTK) analysis (Liu et al., 2021a; 2022b; Allen-Zhu et al., 2019). The second condition is based on a smoothness property of the loss function. For feedforward neural networks, the RSC condition relies on the second-order Taylor expansion of the loss (Banerjee et al., 2023a; Cisneros-Velarde et al., 2025), using the *Hessian* of the neural network (i.e., second order structure), whereas the NTK approach relies on a kernel approximation of the training dynamics (Jacot et al., 2018), using the *gradient* of the network (i.e., first order structure). For a specific model such as neural operators, **the *technical challenge* in using our optimization framework** is to establish suitable properties of the loss, its gradient, and its Hessian in order

---

[*]Equal contribution   [1]VMware Research   [2]University of Illinois Urbana-Champaign.   Correspondence to: Pedro Cisneros-Velarde <pacisne@gmail.com>, Bhavesh Shrimali <bhavesh.shrimali@gmail.com>, Arindam Banerjee <arindamb@illinois.edu>.

*Proceedings of the $42^{nd}$ International Conference on Machine Learning*, Vancouver, Canada. PMLR 267, 2025. Copyright 2025 by the author(s).

to show that the RSC and smoothness conditions are indeed satisfied. We point out that convexity of the loss is not one of the required conditions, so we are not attempting to show that the Hessian of the loss is positive semi-definite, as that will not be true for most neural models, including neural operators.

Having defined a general optimization framework based on RSC and smoothness conditions, **the key novelty of our current work is showing that the losses for DONs and FNOs provably satisfy these two conditions when the neural operators are wide, despite the substantial differences in their architectures and mathematical analyses**. For both DONs and FNOs, we need to bound the Hessian of their respective empirical losses and of the neural operator models themselves in order to determine whether the RSC and smoothness properties are satisfied.

The *challenge* in the analysis of DONs stems from the fact that the output of this neural operator is the inner product of two neural networks. This greatly complicates the Hessian structure of the loss compared to standard neural networks. Indeed, the Hessian now contains cross-interaction terms between two neural networks which have to be carefully analyzed and which require a more complex definition of the restricted set over which the RSC property is defined compared to standard neural networks.

The *challenge* in the analysis of FNOs stems from the fact that it contains, inside their neural network structure, learnable weights that define transformations in the Fourier domain—something absent in standard neural networks. This complicates the Hessian structure of the FNO since it contains parameters both in the data domain and transformed Fourier domain leading to cross-derivatives between parameters from both domains. Thus, a more involved analysis than of standard neural networks is required.

Remarkably, we find that **the *widths* of both neural operator models benefit our optimization guarantees in similar ways**. First, the widths appear in the RSC condition such that larger widths make this condition less restrictive. Second, larger widths enlarge the neighborhood around the initialization point where our optimization guarantees hold. Similar benefits from larger widths were found for standard neural networks by Banerjee et al. (2023a), despite the substantial differences between our analyses and theirs (as mentioned in the *challenges* above).

Finally, to complement our theoretical results, we present empirical evaluations of DONs and FNOs and show the benefits of width on learning three popular operators in the literature (Li et al., 2021a; Lu et al., 2021): antiderivative, diffusion-reaction, and Burgers' equation. Our experiments show that increasing the width leads to lower training losses and generally leads to faster convergence.

**Paper Organization**. Section 2 presents related literature. Section 3 outlines the architectures and learning problems for DONs and FNOs. Section 4 establishes our general optimization framework, and Section 5 and Section 6 establish convergence guarantees using this framework for DONs and FNOs respectively, highlighting the benefits of width. Section 7 compares our results and known ones for standard neural networks. Section 8 presents empirical evaluations on the benefits of width. Section 9 is the conclusion.

**Notation**. $\|\cdot\|_2$ denotes the $L_2$-norm or the induced matrix $L_2$-norm when the argument is a vector or a matrix, respectively. Given an operator/function $f$, $\mathrm{ran}(f)$ and $\mathrm{dom}(f)$ denote the range and domain of $f$, respectively.

## 2. Related Work

We only present a brief overview of the literature related to our work and provide a more extensive treatment in Appendix A. In the case of DONs, approximation (Lu et al., 2021) and generalization (Kontolati et al., 2022) properties have been formally studied, as well as several applications (Goswami et al., 2022; Wang & Perdikaris, 2021; Diab & Al Kobaisi, 2024; Centofanti et al., 2024; Sun et al., 2023). Nevertheless, optimization guarantees for DONs is an open problem. Approximation properties for FNOs have been formally studied (Kovachki et al., 2021), and diverse applications of FNOs and various Fourier-based operators have been formulated (Li et al., 2020a; Liu et al., 2022a; Wen et al., 2022; Pathak et al., 2022; Centofanti et al., 2024; Li et al., 2023; Yang et al., 2023; Harder et al., 2023). Nevertheless, optimization guarantees for DONs is also an open problem. Though formal optimization guarantees for neural operators are largely absent, there is a more established literature on such guarantees for neural networks. We highlight two particular approaches for optimization analysis: based on the NTK approach (Jacot et al., 2018; Liu et al., 2021a; Banerjee et al., 2023b; Du et al., 2019; Allen-Zhu et al., 2019) and on the RSC approach (Banerjee et al., 2023a; Cisneros-Velarde et al., 2025)—our work is related to the latter.

## 3. Learning Neural Operators

A neural operator (Li et al., 2021a; 2020b; Lu et al., 2021) is a parametric model based on neural networks that aims to best approximate a mapping between two function spaces, which can be linear, such as the antiderivative or integral operator, or nonlinear such as the solution operator of a nonlinear PDE. Thus, letting $G^\dagger$ denote the ground-truth operator we are trying to approximate and $G_{\boldsymbol{\theta}}$ denote the neural operator parameterized by the parameter vector $\boldsymbol{\theta}$, the objective is to *learn* $\boldsymbol{\theta}$ such that, given an input function $\boldsymbol{u}$, we have $G_{\boldsymbol{\theta}}(\boldsymbol{u}) \approx G^\dagger(\boldsymbol{u})$. Such learning is done by solving

an optimization problem using data samples consisting of tuples of input and output function values of $G^\dagger$. This optimization problem is analogous to the notion of learning in finite dimensions, which is precisely the setup for which classical deep learning is used.

We now introduce DONs and FNOs. More information about neural operators and the schematics of both DONs and FNOs are found in Appendix B.

### 3.1. Learning Deep Operator Networks (DONs)

The DON model (Lu et al., 2021) is defined as the inner product of two deep feedforward neural networks, each one with $K$ output neurons. Given the branch net $\boldsymbol{f} = \{f_k\}_{k=1}^K$ and the trunk net $\boldsymbol{g} = \{g_k\}_{k=1}^K$, the DON is

$$G_{\boldsymbol{\theta}}(\boldsymbol{u})(\boldsymbol{y}) := \sum_{k=1}^K f_k(\boldsymbol{\theta}_f; \boldsymbol{u}) g_k(\boldsymbol{\theta}_g; \boldsymbol{y}) , \qquad (1)$$

where the input function $\boldsymbol{u}$ has $\mathrm{ran}(\boldsymbol{u}) \subseteq \mathbb{R}^{d_u}$ and $\boldsymbol{y} \in \mathrm{dom}(G_{\boldsymbol{\theta}}(\boldsymbol{u})) \subseteq \mathbb{R}^{d_y}$ is the output location on which the operator is evaluated. The training data is composed of $n$ input functions $\{\boldsymbol{u}^{(i)}\}_{i=1}^n$ and $q_i$ output locations for each $G^\dagger(\boldsymbol{u}^{(i)})$, i.e., $\{\{\boldsymbol{y}_j^{(i)}\}_{j=1}^{q_i}\}_{i=1}^n$ with $\boldsymbol{y}_j^{(i)} \in \mathbb{R}^{d_y}$ denoting the $j$-th output location for $G_{\boldsymbol{\theta}}^\dagger(\boldsymbol{u}^{(i)})$. Each $\boldsymbol{u}^{(i)}$ is represented in $R$ locations $\{\boldsymbol{x}_r\}_{r=1}^R$ so that $\boldsymbol{u}^{(i)}(\boldsymbol{x}_r) \in \mathbb{R}^{d_u}$, $r \in [R]$. The entire set of learnable parameters is $\boldsymbol{\theta} = [\boldsymbol{\theta}_f^\top \ \boldsymbol{\theta}_g^\top]^\top \in \mathbb{R}^{p_f + p_g}$, where $\boldsymbol{\theta}_f \in \mathbb{R}^{p_f}$ and $\boldsymbol{\theta}_g \in \mathbb{R}^{p_g}$ are the parameter vectors of $\boldsymbol{f}$ and $\boldsymbol{g}$ respectively.

We only consider scalar input functions, i.e., $d_u = 1$. For each $i \in [n]$, we stack $\{u^{(i)}(\boldsymbol{x}_r)\}_{r=1}^R$ as an input vector to $\boldsymbol{f}$, thus, $\boldsymbol{f} : \mathbb{R}^R \to \mathbb{R}^K$. Note that $\boldsymbol{g} : \mathbb{R}^{d_y} \to \mathbb{R}^K$. Then, the DON learning problem is the minimization:

$$\boldsymbol{\theta}_{(\mathrm{don})}^\dagger \in \underset{\boldsymbol{\theta} \in \mathbb{R}^{p_f + p_g}}{\operatorname{argmin}} \ \mathcal{L}\left(G_{\boldsymbol{\theta}}, G^\dagger\right) \qquad (2)$$

where

$$\begin{aligned} &\mathcal{L}\left(G_{\boldsymbol{\theta}}, G^\dagger\right) \\ &= \frac{1}{n} \sum_{i=1}^n \frac{1}{q_i} \sum_{j=1}^{q_i} \left(G_{\boldsymbol{\theta}}(u^{(i)})(\boldsymbol{y}_j^{(i)}) - G^\dagger(u^{(i)})(\boldsymbol{y}_j^{(i)})\right)^2 \end{aligned}$$
$$(3)$$

is the empirical loss function that measures the approximation between $G_{\boldsymbol{\theta}}$ and $G^\dagger$, and where $G_{\boldsymbol{\theta}}(u^{(i)})(\boldsymbol{y}_j^{(i)}) = \sum_{k=1}^K f_k\left(\boldsymbol{\theta}_f; \{u^{(i)}(\boldsymbol{x}_r)\}_{r=1}^R\right) g_k\left(\boldsymbol{\theta}_g; \boldsymbol{y}_j^{(i)}\right)$.

Note that the ground truth operator $G^\dagger$ can either be explicit, e.g., integral of a function, or implicit, e.g., the solution to a nonlinear PDE.

### 3.2. Learning Fourier Neural Operators (FNOs)

The FNO model (Li et al., 2021a) is defined as follows: $G_{\boldsymbol{\theta}}(\boldsymbol{u})(\boldsymbol{x}) := f(\boldsymbol{\theta}; \boldsymbol{x})$ with

$$\begin{aligned} \boldsymbol{\alpha}^{(0)}(\boldsymbol{x}) &= P(\boldsymbol{u}; \boldsymbol{\theta}_p)(\boldsymbol{x}) \\ \boldsymbol{\alpha}^{(l)}(\boldsymbol{x}) &= \mathcal{F}^{(l)}(\boldsymbol{\alpha}^{(l-1)}(\boldsymbol{x}); \boldsymbol{\theta}_{F^{(l)}}) , \ l \in [L+1] \quad (4) \\ f(\boldsymbol{\theta}; \boldsymbol{x}) &= Q(\boldsymbol{\alpha}^{(L+1)}; \boldsymbol{\theta}_q)(\boldsymbol{x}) , \end{aligned}$$

where the input function $\boldsymbol{u}$ has $\mathrm{ran}(\boldsymbol{u}) \subseteq \mathbb{R}^{d_u}$, $G_{\boldsymbol{\theta}}(\boldsymbol{u})(\boldsymbol{x}) \in \mathbb{R}$ is the output of the FNO evaluated at output location $\boldsymbol{x} \in \mathbb{R}^{d_x}$, $\{\mathcal{F}^{(l)}\}_{l=1}^{L+1}$ are nonlinear transformations with learnable parameters $\boldsymbol{\theta}_F = [\boldsymbol{\theta}_{F^{(1)}}^\top, \ldots, \boldsymbol{\theta}_{F^{(L+1)}}^\top]^\top \in \mathbb{R}^F$ and which may contain operations in the Fourier domain, $P$ is an encoder that maps $\boldsymbol{u}$ and $\boldsymbol{x}$ to an ambient space of dimension $d$ and has parameter vector $\boldsymbol{\theta}_p \in \mathbb{R}^p$, and $Q$ is a decoder that maps the output from the block $\boldsymbol{\alpha}^{(L+1)}(\boldsymbol{x})$ to a scalar output with parameter vector $\boldsymbol{\theta}_q \in \mathbb{R}^q$. The entire set of parameters for the FNO can be written as $\boldsymbol{\theta} = [\boldsymbol{\theta}_p^\top \ \boldsymbol{\theta}_F^\top \ \boldsymbol{\theta}_q^\top]^\top$. With a slight abuse of notation, the FNO is simply written as $G_{\boldsymbol{\theta}}(\boldsymbol{u})(\boldsymbol{x}) = f(\boldsymbol{\theta}; \boldsymbol{x})$ in (4) when the input function $\boldsymbol{u}$ is known by the context.

The training data is composed of $n$ input-output pairs $\{(\boldsymbol{u}^{(i)}, G^\dagger(\boldsymbol{u}^{(i)})\}_{i=1}^n$ and a computational grid of evaluations $\{\boldsymbol{x}_r\}_{r=1}^R$. We let $f^{(i)}(\boldsymbol{\theta}; \boldsymbol{x}_r)$ denote the FNO model (4) with input function $\boldsymbol{u}^{(i)}$ and evaluated at $\boldsymbol{x}_r$. Then, the FNO learning problem is the minimization:

$$\boldsymbol{\theta}_{(\mathrm{fno})}^\dagger \in \underset{\boldsymbol{\theta} \in \mathbb{R}^{p+F+q}}{\operatorname{argmin}} \ \mathcal{L}(G_{\boldsymbol{\theta}}, G^\dagger) \qquad (5)$$

with empirical loss function

$$\begin{aligned} &\mathcal{L}(G_{\boldsymbol{\theta}}, G^\dagger) \\ &= \frac{1}{n} \sum_{i=1}^n \frac{1}{R} \sum_{r=1}^R \left(G_{\boldsymbol{\theta}}(\boldsymbol{u}^{(i)})(\boldsymbol{x}_r) - G^\dagger(\boldsymbol{u}^{(i)})(\boldsymbol{x}_r)\right)^2 \end{aligned} \quad (6)$$

and where $G_{\boldsymbol{\theta}}(\boldsymbol{u}^{(i)})(\boldsymbol{x}_r) = \boldsymbol{f}^{(i)}(\boldsymbol{\theta}; \boldsymbol{x}_r)$.

## 4. Optimization Convergence Framework

We now establish *two conditions*—Conditions 1 and 2 below—for the convergence of gradient descent (GD) when minimizing a loss function $\mathcal{L}$. We show that as long as these two conditions are satisfied, the loss will decrease in value. In the following sections we show how the empirical losses used for training DONs (Section 5) and FNOs (Section 6), as in (3) and (6) respectively, satisfy these two conditions.

We consider $\boldsymbol{\theta} \mapsto \mathcal{L}(\boldsymbol{\theta})$ to be continuously differentiable. Let $\boldsymbol{\theta}_0 \in \mathbb{R}^p$ be a suitable initialization point and $\{\boldsymbol{\theta}_t\}_{t \geq 1}$ be the sequence of iterates obtained by GD on loss $\mathcal{L}$ for some step-size $\eta_t > 0$, i.e.,

$$\boldsymbol{\theta}_{t+1} = \boldsymbol{\theta}_t - \eta_t \nabla_{\boldsymbol{\theta}} \mathcal{L}(\boldsymbol{\theta}_t) . \qquad (7)$$

We consider a non-empty set $\mathcal{B}(\boldsymbol{\theta}_0) \subseteq \mathbb{R}^p$ around and including $\boldsymbol{\theta}_0$.

**Assumption 1** (**Iterates inside** $\mathcal{B}(\boldsymbol{\theta}_0)$)**.** *All iterates* $\{\boldsymbol{\theta}_t\}_{t \geq 1}$ *follow GD as in* (7) *and are inside the set* $\mathcal{B}(\boldsymbol{\theta}_0)$.

The first condition is based on the concept of Restricted Strong Convexity (RSC) being satisfied for $\mathcal{L}$.

**Definition 1** (**Restricted strong convexity (RSC)**)**.** *A function* $\mathcal{L}$ *is said to satisfy* $\alpha$-*restricted strong convexity (*$\alpha$-*RSC) w.r.t. the tuple* $(\mathcal{S}, \boldsymbol{\theta})$ *if for any* $\boldsymbol{\theta}' \in \mathcal{S} \subseteq \mathbb{R}^p$ *and some fixed* $\boldsymbol{\theta} \in \mathbb{R}^p$, *we have*

$$\mathcal{L}(\boldsymbol{\theta}') \geq \mathcal{L}(\boldsymbol{\theta}) + \langle \boldsymbol{\theta}' - \boldsymbol{\theta}, \nabla_{\boldsymbol{\theta}}\mathcal{L}(\boldsymbol{\theta}) \rangle + \frac{\alpha}{2} \|\boldsymbol{\theta}' - \boldsymbol{\theta}\|_2^2 , \tag{8}$$

*with* $\alpha > 0$.

**Condition 1** (**RSC**)**.** *Consider Assumption 1. At step t, there exists a non-empty set* $\mathcal{N}_t$ *such that:*

*(a)* $\mathcal{N}_t \subseteq \mathcal{B}(\boldsymbol{\theta}_0)$;

*(b) one of these two conditions hold:*

*(b.1)* $\boldsymbol{\theta}_{t+1} \in \mathcal{N}_t$ *with either* $\boldsymbol{\theta}_t \notin \mathcal{N}_t$ *or* $\mathcal{L}(\boldsymbol{\theta}_t) \neq \inf_{\boldsymbol{\theta} \in \mathcal{N}_t} \mathcal{L}(\boldsymbol{\theta})$,

*(b.2) there exists some* $\boldsymbol{\theta}' \in \mathcal{N}_t$ *such that* $\mathcal{L}(\boldsymbol{\theta}') < \mathcal{L}(\boldsymbol{\theta}_t)$;

*(c)* $\mathcal{L}$ *satisfies* $\alpha_t$-*RSC w.r.t.* $(\mathcal{N}_t, \boldsymbol{\theta}_t)$ *for some* $\alpha_t > 0$.

Note that $\mathcal{L}$ need not be convex for it to satisfy $\alpha_t$-RSC.

The second condition is based on the smoothness of $\mathcal{L}$.

**Condition 2** (**Smoothness**)**.** *The function* $\mathcal{L}$ *is* $\beta$-*smooth in* $\mathcal{B}(\boldsymbol{\theta}_0)$, *i.e., for any* $\boldsymbol{\theta}', \boldsymbol{\theta} \in \mathcal{B}(\boldsymbol{\theta}_0)$, *we have* $\mathcal{L}(\boldsymbol{\theta}') \leq \mathcal{L}(\boldsymbol{\theta}) + \langle \boldsymbol{\theta}' - \boldsymbol{\theta}, \nabla_{\boldsymbol{\theta}}\mathcal{L}(\boldsymbol{\theta}) \rangle + \frac{\beta}{2}\|\boldsymbol{\theta}' - \boldsymbol{\theta}\|_2^2$, *with* $\beta > 0$.

As long as Conditions 1 and 2 are satisfied at step $t$ of the GD update in (7), the loss is guaranteed to decrease with a suitable step-size choice.

**Theorem 1** (**Global loss reduction**)**.** *Consider Assumption 1 and Conditions 1 and 2 with* $\alpha_t \leq \beta$ *at step t of the GD update* (7) *with step-size* $\eta_t = \frac{\omega_t}{\beta}$ *for some* $\omega_t \in (0, 2)$. *If* $\mathcal{L}(\boldsymbol{\theta}_t) \neq \inf_{\boldsymbol{\theta} \in \mathcal{B}(\boldsymbol{\theta}_0)} \mathcal{L}(\boldsymbol{\theta})$, *then we have*

$$0 \leq \gamma_t := \frac{\inf_{\boldsymbol{\theta} \in \mathcal{N}_t} \mathcal{L}(\boldsymbol{\theta}) - \inf_{\boldsymbol{\theta} \in \mathcal{B}(\boldsymbol{\theta}_0)} \mathcal{L}(\boldsymbol{\theta})}{\mathcal{L}(\boldsymbol{\theta}_t) - \inf_{\boldsymbol{\theta} \in \mathcal{B}(\boldsymbol{\theta}_0)} \mathcal{L}(\boldsymbol{\theta})} < 1 \text{ and}$$

$$\mathcal{L}(\boldsymbol{\theta}_{t+1}) - \inf_{\boldsymbol{\theta} \in \mathcal{B}(\boldsymbol{\theta}_0)} \mathcal{L}(\boldsymbol{\theta})$$
$$\leq \left( 1 - \frac{\alpha_t \omega_t (1 - \gamma_t)}{\beta}(2 - \omega_t) \right) \left( \mathcal{L}(\boldsymbol{\theta}_t) - \inf_{\boldsymbol{\theta} \in \mathcal{B}(\boldsymbol{\theta}_0)} \mathcal{L}(\boldsymbol{\theta}) \right) . \tag{9}$$

Theorem 1's proof is found in Appendix C. We note that if the infimum loss inside $\mathcal{B}(\boldsymbol{\theta}_0)$ is attained at time $t$, i.e., $\mathcal{L}(\boldsymbol{\theta}_t) = \inf_{\boldsymbol{\theta} \in \mathcal{B}(\boldsymbol{\theta}_0)} \mathcal{L}(\boldsymbol{\theta})$, then there is nothing to prove—hence the conditional in the second sentence of Theorem 1.

**Remark 1** (The RSC to smoothness ratio)**.** Theorem 1 requires $\alpha_t/\beta \leq 1$, which needs to be proved for the particular function $\mathcal{L}$ being considered. $\square$

**Remark 2** (RSC and local strong convexity)**.** If equation (8) were to hold for any $\boldsymbol{\theta}, \boldsymbol{\theta}' \in \mathcal{B}(\boldsymbol{\theta}_0)$, then $\mathcal{L}$ would be a *locally strongly convex* function in the set $\mathcal{B}(\boldsymbol{\theta}_0)$ (Boyd & Vandenberghe, 2004). This is a stronger condition on $\mathcal{L}$ which makes $\alpha$ in (8) *independent* from the choice of $\boldsymbol{\theta}$ (in the context of Theorem 1, $\alpha_t$ would be independent from $t$). Moreover, it satisfies $\alpha/\beta < 1$. $\square$

Our analysis is inspired by the recent works (Banerjee et al., 2023a) and (Cisneros-Velarde et al., 2025), where optimization guarantees were done for feedforward networks and normalization. We abstract out from those special cases, and demonstrate that our analysis works for any losses satisfying Conditions 1 and 2—indeed, (Cisneros-Velarde et al., 2025) particularly satisfies Condition 1(b.1) and (Banerjee et al., 2023a) satisfies Condition 1(b.2). Thus, in the context of our paper, **the largest effort in establishing optimization guarantees for DONs and FNOs is to show these two models satisfy Conditions 1 and 2 with** $\alpha_t/\beta \leq 1$.

## 5. Optimization Analysis for DON

We consider, analogous to (Liu et al., 2021b), the branch net as a fully connected feedforward neural network:

$$\boldsymbol{\alpha}_f^{(0)} = \boldsymbol{u}(\boldsymbol{x})$$
$$\boldsymbol{\alpha}_f^{(l)} = \phi \left( \frac{1}{\sqrt{m_\mathrm{f}}} W_f^{(l)} \boldsymbol{\alpha}_f^{(l-1)} \right), \ l \in [L-1] \tag{10}$$
$$\boldsymbol{f} = \boldsymbol{\alpha}_f^{(L)} = \frac{1}{\sqrt{m_f}} W_f^{(L)} \boldsymbol{\alpha}_f^{(L-1)}$$

where with some abuse of notation $\boldsymbol{u}(\boldsymbol{x}) := [u(\boldsymbol{x}_1), \ldots, u(\boldsymbol{x}_R)]^\top$ is the vector of all scalar evaluations of $u$ at each of the $R$ locations, $\phi$ is a pointwise smooth activation function, $\boldsymbol{\alpha}_f^{(l)}$ is the output at layer $l \in [L]$, and the weight matrices are $W_f^{(1)} \in \mathbb{R}^{m_f \times R}$ and $W_f^{(l)} \in \mathbb{R}^{m_f \times m_f}$ at layer $l \in \{2, \ldots, L-1\}$. The branch net has width $m_f$ (all hidden layers have the same width). Similarly, the trunk net is a fully connected feedforward network:

$$\boldsymbol{\alpha}_g^{(0)} = \boldsymbol{y}$$
$$\boldsymbol{\alpha}_g^{(l)} = \phi \left( \frac{1}{\sqrt{m_g}} W_g^{(l)} \boldsymbol{\alpha}_g^{(l-1)} \right), \ l \in [L-1] \tag{11}$$
$$\boldsymbol{g} = \boldsymbol{\alpha}_g^{(L)} = \frac{1}{\sqrt{m_g}} W_g^{(L)} \boldsymbol{\alpha}_g^{(L-1)}$$

where $\boldsymbol{y} \in \mathbb{R}^{d_y}$ is the output location, and the weight matrices are $W_g^{(1)} \in \mathbb{R}^{m_g \times d_y}$ and $W_g^{(l)} \in \mathbb{R}^{m_g \times m_g}$ at layer $l \in \{2, \ldots, L-1\}$. The trunk net has width $m_g$ (all hidden layers have the same width). Finally, we recall that we have $K$ outputs on each network, i.e., $W_f^{(L)} \in \mathbb{R}^{K \times m_f}$ and $W_g^{(L)} \in \mathbb{R}^{K \times m_g}$. Given $l \in [L]$, we denote by $(w_{f,k}^{(l)})^\top$ and $(w_{g,k}^{(l)})^\top$ the $k$-th row of the matrices $W_f^{(l)}$ and $W_g^{(l)}$ respectively, and by $w_{f,ij}^{(l)}$ and $w_{g,ij}^{(l)}$ their respective $ij$-entry. Using the notation in Section 3.1, the set of trainable parameters is $\boldsymbol{\theta} = [\boldsymbol{\theta}_f^\top \ \boldsymbol{\theta}_g^\top]^\top \in \mathbb{R}^{p_f + p_g}$, with $\boldsymbol{\theta}_f = [\mathrm{vec}(W_f^{(1)})^\top, \ldots, \mathrm{vec}(W_f^{(L)})^\top]^\top$ and $\boldsymbol{\theta}_g = [\mathrm{vec}(W_g^{(2)})^\top, \ldots, \mathrm{vec}(W_g^{(L)})^\top]^\top$. Let $\boldsymbol{\theta}_0$ be the parameter vector at initialization and $\boldsymbol{\theta}_t$ be it at time step $t$.

We make the following assumptions for our analysis:

**Assumption 2** (**Activation functions**). *The activation function $\phi$ of the DON is 1-Lipschitz and $\beta_\phi$-smooth (i.e. $\phi'' \leq \beta_\phi$) for some $\beta_\phi > 0$.*

**Assumption 3** (**Initialization of weights**). *All weights of the branch and trunk nets are initialized independently as follows: (i) $w_{f_0, \, ij}^{(l)} \sim \mathcal{N}(0, \sigma_{f,0}^2)$ and $w_{g_0, \, ij}^{(l)} \sim \mathcal{N}(0, \sigma_{g,0}^2)$ for $l \in [L-1]$ where $\sigma_{f,0} = \frac{\sigma_0}{2(1 + \frac{\sqrt{\log m_f}}{\sqrt{2m_f}})}$ and $\sigma_{g,0} = \frac{\sigma_0}{2(1 + \frac{\sqrt{\log m_g}}{\sqrt{2m_g}})}$, $\sigma_0 > 0$; (ii) $w_{f_0,k}^{(L)}$ and $w_{g_0,k}^{(L)}$, $k \in [K]$, are random vectors with unit norms, i.e., $\|w_{f_0,k}^{(L)}\|_2 = 1$ and $\|w_{g_0,k}^{(L)}\|_2 = 1$. Further, we assume the input to the branches are normalized as $\|\boldsymbol{u}(\boldsymbol{x})\|_2 = \sqrt{R}$ and $\|\boldsymbol{y}\|_2 = \sqrt{d_y}$.*

For a given parameter vector $\bar{\boldsymbol{\theta}} = [\bar{\boldsymbol{\theta}}_f^\top, \bar{\boldsymbol{\theta}}_g^\top] \in \mathbb{R}^{p_f + p_g}$, we introduce the neighborhood set $B_{\rho,\rho_1}^{\mathrm{Euc}}(\bar{\boldsymbol{\theta}}) = \{\boldsymbol{\theta} \in \mathbb{R}^{p_f + p_g} : \|W_f^{(l)} - \bar{W}_f^{(l)}\|_2 \leq \rho, \|W_g^{(l)} - \bar{W}_g^{(l)}\|_2 \leq \rho, l \in [L-1], \|w_{f,k}^{(L)} - \bar{w}_{f,k}^{(L)}\|_2 \leq \rho_1, \|w_{g,k}^{(L)} - \bar{w}_{g,k}^{(L)}\|_2 \leq \rho_1, k \in [K]\}$ for radii $\rho, \rho_1 > 0$. We say that an element of $B_{\rho,\rho_1}^{\mathrm{Euc}}(\bar{\boldsymbol{\theta}})$ is *strictly inside* $B_{\rho,\rho_1}^{\mathrm{Euc}}(\bar{\boldsymbol{\theta}})$ when it satisfies every inequality in the set's definition without equality. We also define $B_\rho^{\mathrm{Euc}}(\bar{\boldsymbol{\theta}})$ as an Euclidean ball around $\bar{\boldsymbol{\theta}}$ with radius $\rho > 0$.

The following is an assumption analogous to the general Assumption 1.

**Assumption 4** (**Iterates inside** $B_{\rho,\rho_1}^{\mathrm{Euc}}(\boldsymbol{\theta}_0)$). *All iterates $\{\boldsymbol{\theta}_t\}_{t \geq 1}$ follow GD as in (7) and are strictly inside the set $B_{\rho,\rho_1}^{\mathrm{Euc}}(\boldsymbol{\theta}_0)$ for fixed $\rho, \rho_1 > 0$.*

We now focus on showing that the two conditions needed for optimization using GD as discussed in Section 4 are indeed satisfied by DONs. We start with the definition of a set $Q_\kappa^t$ parameterized by $\kappa \in (0, 1)$, which will help construct the set $\mathcal{N}_t$ in Condition 1 for RSC. Due to the interaction of two neural networks (branch and trunk), the definition of $Q_\kappa^t$ looks seemingly involved. However, note that $Q_\kappa^t$ is only

needed for establishing the RSC condition for the analysis and does not change the computation of the optimization algorithm, which is simply GD run over all the branch and trunk network parameters.

**Definition 2** ($Q_\kappa^t$ sets for DONs). *For an iterate $\boldsymbol{\theta}_t = [\boldsymbol{\theta}_{f,t}^\top \ \boldsymbol{\theta}_{g,t}^\top]^\top$ and $\kappa \in (0, 1)$, we define the set:*

$$
\begin{aligned}
Q_\kappa^t := \Bigg\{ &\boldsymbol{\theta}' = [\boldsymbol{\theta}_f'^\top \ \boldsymbol{\theta}_g'^\top]^\top \in \mathbb{R}^{p_f + p_g} : \\
&|\cos(\boldsymbol{\theta}' - \boldsymbol{\theta}_t, \nabla_{\boldsymbol{\theta}} \bar{G}_t)| \geq \kappa, \\
&(\boldsymbol{\theta}_f' - \boldsymbol{\theta}_{f,t})^\top \left( \frac{1}{n} \sum_{i=1}^n \frac{1}{q_i} \sum_{j=1}^{q_i} \ell_{i,j}' \sum_{k=1}^K \nabla_{\boldsymbol{\theta}_f} f_k^{(i)} \nabla_{\boldsymbol{\theta}_g} g_{k,j}^{(i) \ \top} \right) \\
&\times (\boldsymbol{\theta}_g' - \boldsymbol{\theta}_{g,t}) \geq 0, \\
&(\boldsymbol{\theta}_f' - \boldsymbol{\theta}_{f,t})^\top \left( \sum_{k=1}^K \nabla_{\boldsymbol{\theta}_f} f_k^{(i)} \nabla_{\boldsymbol{\theta}_g} g_{k,j}^{(i) \ \top} \right) \\
&\times (\boldsymbol{\theta}_g' - \boldsymbol{\theta}_{g,t}) \leq 0, \forall i \in [n], \forall j \in [q_i] \Bigg\},
\end{aligned}
$$

(12)

*where $\nabla_{\boldsymbol{\theta}} \bar{G}_t = \frac{1}{n} \sum_{i=1}^n \frac{1}{q_i} \sum_{j=1}^{q_i} \nabla_{\boldsymbol{\theta}} G_{\boldsymbol{\theta}_t}(u^{(i)})(\boldsymbol{y}_j^{(i)})$, $\ell_{i,j} = (G_{\boldsymbol{\theta}_t}(u^{(i)})(\boldsymbol{y}_j^{(i)}) - G^\dagger(u^{(i)})(\boldsymbol{y}_j^{(i)}))^2$, and both $\nabla_{\boldsymbol{\theta}_f} f_k^{(i)}$ and $\nabla_{\boldsymbol{\theta}_g} g_{k,j}^{(i)}$ are evaluated on $\boldsymbol{\theta}_t$.*

We now prove the RSC and smoothness conditions (corresponding to Conditions 1 and 2, respectively). Using the nomenclature of Section 4, the set $B_{\rho,\rho_1}^{\mathrm{Euc}}(\boldsymbol{\theta}_0)$ corresponds to $\mathcal{B}(\boldsymbol{\theta}_0)$, and $B_\kappa^t := Q_\kappa^t \cap B_{\rho,\rho_1}^{\mathrm{Euc}}(\boldsymbol{\theta}_0) \cap B_{\rho_2}^{\mathrm{Euc}}(\boldsymbol{\theta}_t)$ with $\rho_2 > 0$ corresponds to $\mathcal{N}_t$.

**Theorem 2** (**RSC for DONs**). *Consider Assumptions 2, 3, and 4, and $Q_\kappa^t$ as in Definition 2 with $\|\nabla_{\boldsymbol{\theta}} \bar{G}_t\|_2 \neq 0$. Then, the set $B_\kappa^t := Q_\kappa^t \cap B_{\rho,\rho_1}^{\mathrm{Euc}}(\boldsymbol{\theta}_0) \cap B_{\rho_2}^{\mathrm{Euc}}(\boldsymbol{\theta}_t)$ is a non-empty set that satisfies Condition 1(a) and (b) for suitable $\rho_2$ and sufficiently small $\kappa \in (0, 1)$. Moreover, with probability at least $1 - 2KL(\frac{1}{m_f} + \frac{1}{m_g})$, at step $t$ of GD, the DON loss $\mathcal{L}$ (3) satisfies equation (8) with*

$$
\alpha_t = 2\kappa^2 \|\nabla_{\boldsymbol{\theta}} \bar{G}_t\|_2^2 - c_1 K^2 \left( \frac{1}{\sqrt{m_f}} + \frac{1}{\sqrt{m_g}} \right) \quad (13)
$$

*where $\nabla_{\boldsymbol{\theta}} \bar{G}_t = \frac{1}{n} \sum_{i=1}^n \frac{1}{q_i} \sum_{j=1}^{q_i} \nabla_{\boldsymbol{\theta}} G_{\boldsymbol{\theta}_t}(u^{(i)})(\boldsymbol{y}_j^{(i)})$, and for some constant $c_1 > 0$ which depends polynomially on the depth $L$, and the radii $\rho$, $\rho_1$, and $\rho_2$ whenever $\sigma_0 \leq 1 - \rho \max\{\frac{1}{\sqrt{m_f}}, \frac{1}{\sqrt{m_g}}\}$. Thus, the loss $\mathcal{L}$ satisfies RSC w.r.t $(B_\kappa^t, \boldsymbol{\theta}_t)$, i.e., Condition 1(c), whenever $\|\nabla_{\boldsymbol{\theta}} \bar{G}_t\|_2^2 = \Omega(\frac{1}{\sqrt{m_f}} + \frac{1}{\sqrt{m_g}})$.*

**Theorem 3** (**Smoothness for DONs**). *Under Assumptions 2 and 3, with probability at least $1 - 2KL(\frac{1}{m_f} + \frac{1}{m_g})$, the DON loss $\mathcal{L}$ (3) is $\beta$-smooth in $B_{\rho,\rho_1}^{\mathrm{Euc}}(\boldsymbol{\theta}_0)$ with $\beta = c_2 K^2$, where $c_2 > 0$ is a constant which depends polynomially on the depth $L$, and the radii $\rho$, $\rho_1$, and $\rho_2$ whenever $\sigma_0 \leq 1 - \rho \max\{\frac{1}{\sqrt{m_f}}, \frac{1}{\sqrt{m_g}}\}$.*

**Remark 3** (Ensuring that $\alpha_t/\beta < 1$). As mentioned in Remark 2, in order to use the optimization framework from Section 4, the statement of Theorem 1 requires $\alpha_t/\beta \le 1$. We prove that this condition is satisfied with a strict inequality for DONs in Proposition 1 in Appendix D. $\square$

**Optimization Under Gradient Descent for DONs.** We have that Theorem 2 satisfies Condition 1 and Theorem 3 satisfies Condition 2. We also proved that $\alpha_t/\beta < 1$. Thus, when $\|\nabla_{\boldsymbol{\theta}}\bar{G}_t\|_2^2 = \Omega(\frac{1}{\sqrt{m_f}} + \frac{1}{\sqrt{m_g}})$, i.e., $\alpha_t > 0$, a decrease on the loss function by GD is ensured with probability at least $1 - 2KL(\frac{1}{m_f} + \frac{1}{m_g})$ towards its minimum value taken within the set $B_{\rho,\rho_1}^{\text{Euc}}(\boldsymbol{\theta}_0)$ due to Theorem 1.

**Remark 4** (The benefit of over-parameterization for the RSC property). According to (13), $\|\nabla_{\boldsymbol{\theta}}\bar{G}_t\|_2^2 = \Omega(\frac{1}{\sqrt{m_f}} + \frac{1}{\sqrt{m_g}})$ is needed to ensure that $\alpha_t > 0$, i.e., to ensure that the empirical loss $\mathcal{L}$ satisfies the RSC property at time $t$. Thus, as both widths $m_f$ and $m_g$ increase, $\mathcal{L}$ attains the RSC property at a lower value of $\|\nabla_{\boldsymbol{\theta}}\bar{G}_t\|_2^2$. $\square$

**Remark 5** (Over-parameterization allows for a larger neighborhood around initialization). The condition $\sigma_0 \le 1 - \rho \max\{\frac{1}{\sqrt{m_f}}, \frac{1}{\sqrt{m_g}}\}$ (required for obtaining a polynomial dependence on $L$ for both RSC and smoothness parameters) implies $\rho \le \min\{\sqrt{m_f}, \sqrt{m_g}\}$ since $\sigma_0$ must be positive. Thus, it is possible to increase the radius $\rho$ as we increase both $m_f$ and $m_g$. Thus, we can enlarge the neighborhood around the initialization point where our guarantees hold as the widths increase. $\square$

## 6. Optimization Analysis for FNO

As in the case of DONs, we also focus on scalar input functions $u$. To pass the input function $u$, we discretize it by sampling it on $\bar{R}$ locations, forming a vector of dimension $\bar{R}$. Thus, the encoder $P(u; \boldsymbol{\theta}_p)(\boldsymbol{x})$ in equation (4) takes a vector of dimension $\bar{R} + d_x$ ($\bar{R}$ from the sampled $u$ and $d_x$ from the output location where we evaluate the operator on). For our purposes, we consider a fixed (not trainable) encoder with output dimension $d$: $P(u; \boldsymbol{\theta}_p)(\boldsymbol{x}) \equiv P(u)(\boldsymbol{x}) \in \mathbb{R}^d$; and a linear decoder $Q(\boldsymbol{\alpha}^{(L+1)}; \boldsymbol{\theta}_q)(\boldsymbol{x}) = \frac{1}{\sqrt{m}}\mathbf{v}^\top \boldsymbol{\alpha}^{(L+1)}(\boldsymbol{x}) \in \mathbb{R}$ with $\boldsymbol{\theta}_q \equiv \mathbf{v} \in \mathbb{R}^m$ assuming $\boldsymbol{\alpha}^{(L+1)}(\boldsymbol{x}) \in \mathbb{R}^m$. Thus, following (Li et al., 2021a), the FNO model is:

$$\boldsymbol{\alpha}^{(0)} = P(u)(\boldsymbol{x})$$

$$\boldsymbol{\alpha}^{(1)} = \phi\left(\frac{1}{\sqrt{m}}W^{(1)}\boldsymbol{\alpha}^{(0)}\right)$$

$$\boldsymbol{\alpha}^{(l)} = \phi\left(\frac{1}{\sqrt{m}}W^{(l)}\boldsymbol{\alpha}^{(l-1)} + \frac{1}{\sqrt{m}}F^*R^{(l)}F\boldsymbol{\alpha}^{(l-1)}\right),$$

$$l \in \{2, \dots, L+1\}$$

$$f(\boldsymbol{\theta}; \boldsymbol{x}) = \frac{1}{\sqrt{m}}\mathbf{v}^\top \boldsymbol{\alpha}^{(L+1)},$$

where $\phi$ is a pointwise smooth activation function, $F$ is the discrete Fourier transform kernel (as a matrix) with $F^*$ being its conjugate transpose, the weight matrices are $W^{(1)} \in \mathbb{R}^{m \times d}$, $W^{(l)} \in \mathbb{R}^{m \times m}$ and $R^{(l)} \in \mathbb{R}^{m \times m}$ for layer $l \in \{2, \dots, L+1\}$ (all hidden layers have the same width $m$). The $ij$-entries of $W^{(l)}$ and $R^{(l)}$ are $w_{ij}^{(l)}$ and $r_{ij}^{(l)}$, respectively, for an appropriate $l$. With some abuse of notation, we denote the entire set of trainable parameters by $\boldsymbol{\theta} = [\boldsymbol{\theta}_w^\top \ \boldsymbol{\theta}_r^\top]^\top$, with $\boldsymbol{\theta}_w = [\text{vec}(W^{(1)})^\top, \dots, \text{vec}(W^{(L+1)})^\top \ \mathbf{v}^\top]^\top$ and $\boldsymbol{\theta}_r = [\text{vec}(R^{(2)})^\top, \dots, \text{vec}(R^{(L+1)})^\top]^\top$. We denote the number of parameters by $p_w + p_r$, where $\boldsymbol{\theta}_w \in \mathbb{R}^{p_w}$ and $\boldsymbol{\theta}_r \in \mathbb{R}^{p_r}$. Let $\boldsymbol{\theta}_0$ be the parameter vector at initialization and $\boldsymbol{\theta}_t$ be it at time step $t$.

We remark that our model uses an $m \times m$ Discrete Fourier Transform kernel $F$, whose $kj$-entry is $F_{kj} = e^{-\frac{2\pi\iota}{m}(k-1)(j-1)}$, with $\iota$ representing the imaginary unit.

**Assumption 5** (**Activation functions**). *The activation function $\phi$ of the FNO is 1-Lipschitz and $\beta_\phi$-smooth (i.e. $\phi_l'' \le \beta_\phi$) for some $\beta_\phi > 0$.*

**Assumption 6** (**Initialization of weights**). *All weights of the FNO are initialized independently as follows: (i) $w_{0,ij}^{(l)} \sim \mathcal{N}(0, \sigma_{0w}^2)$ and $r_{0,ij}^{(l)} \sim \mathcal{N}(0, \sigma_{0r}^2)$ for every appropriate $l$, where $\sigma_{0,w} = \frac{\sigma_{1,w}}{2(1 + \frac{\sqrt{\log m}}{\sqrt{2m}})}$ and $\sigma_{0,r} = \frac{\sigma_{1,r}}{2(1 + \frac{\sqrt{\log m}}{\sqrt{2m}})}$ with $\sigma_{1,w}, \sigma_{1,r} > 0$; (ii) the decoder parameter $\mathbf{v}$ is a random vector with unit norm $\|\mathbf{v}\|_2 = 1$. Further, we assume the encoder output satisfies $\|\boldsymbol{\alpha}^{(0)}\|_2 = \sqrt{d}$.*

For a given parameter vector $\bar{\boldsymbol{\theta}} \in \mathbb{R}^{p_w + p_r}$, we introduce the neighborhood set $B_{\rho_w, \rho_r, \rho_1}^{\text{Euc}}(\bar{\boldsymbol{\theta}}) = \{\boldsymbol{\theta} \in \mathbb{R}^{p_w + p_r} : \|W^{(l)} - \bar{W}^{(l)}\|_2 \le \rho_w, l \in [L+1], \|R^{(l)} - \bar{R}^{(l)}\|_2 \le \rho_r, l \in \{2, \dots, L+1\}, \|\mathbf{v} - \bar{\mathbf{v}}\|_2 \le \rho_1\}$ for radii $\rho_w, \rho_r, \rho_1 > 0$. We say that an element of $B_{\rho_w, \rho_r, \rho_1}^{\text{Euc}}(\bar{\boldsymbol{\theta}})$ is *strictly inside* $B_{\rho_w, \rho_r, \rho_1}^{\text{Euc}}(\bar{\boldsymbol{\theta}})$ when it satisfies every inequality in the set's definition without equality.

The following assumption is analogous to Assumption 1.

**Assumption 7** (**Iterates inside** $B_{\rho_w, \rho_r, \rho_1}^{\text{Euc}}(\boldsymbol{\theta}_0)$). *All iterates $\{\boldsymbol{\theta}_t\}_{t \ge 1}$ follow GD as in (7) and are strictly inside the set $B_{\rho_w, \rho_r, \rho_1}^{\text{Euc}}(\boldsymbol{\theta}_0)$ for fixed $\rho_w, \rho_r, \rho_1 > 0$.*

We also introduce the following auxiliary set.

**Definition 3** ($Q_\kappa^t$ sets for FNOs). *For an iterate $\boldsymbol{\theta}_t$, let $\nabla_{\boldsymbol{\theta}}\bar{G}_t = \frac{1}{n}\sum_{i=1}^n \frac{1}{R}\sum_{j=1}^R \nabla_{\boldsymbol{\theta}}G_{\boldsymbol{\theta}_t}(u^{(i)})(\boldsymbol{x}_j)$. For $\kappa \in (0, 1)$, define $Q_\kappa^t := \{\boldsymbol{\theta} \in \mathbb{R}^{p_w + p_r} \mid |\cos(\boldsymbol{\theta} - \boldsymbol{\theta}_t, \nabla_{\boldsymbol{\theta}}\bar{G}_t)| \ge \kappa\}$.*

Note that unlike DONs, the $Q_\kappa^t$ sets for FNOs are relatively simpler due to a single network architecture.

Next, we prove the RSC and smoothness conditions (corresponding to Conditions 1 and 2, respectively). Using the nomenclature of Section 4, the set $B_{\rho_w, \rho_r, \rho_1}^{\text{Euc}}(\boldsymbol{\theta}_0)$ corre-

sponds to $\mathcal{B}(\boldsymbol{\theta}_0)$, and $B_\kappa^t := Q_\kappa^t \cap B_{\rho_w,\rho_r\rho_1}^{\text{Euc}}(\boldsymbol{\theta}_0) \cap B_{\rho_2}^{\text{Euc}}(\boldsymbol{\theta}_t)$ with $\rho_2 > 0$ corresponds to $\mathcal{N}_t$.

**Theorem 4** (**RSC for FNOs**). *Consider Assumptions 5, 6, and 7, and $Q_\kappa^t$ as in Definition 3 with $\|\nabla_{\boldsymbol{\theta}} \bar{G}_t\|_2 \neq 0$. Then, the set $B_\kappa^t := Q_\kappa^t \cap B_{\rho_w,\rho_r\rho_1}^{\text{Euc}}(\boldsymbol{\theta}_0) \cap B_{\rho_2}^{\text{Euc}}(\boldsymbol{\theta}_t)$ is a non-empty set that satisfies Condition 1(a) and (b) for suitable $\rho_2$. Moreover, with probability at least $1 - \frac{2(L+2)}{m}$, at step $t$ of GD, the FNO loss $\mathcal{L}$ (6) satisfies equation (8) with*

$$\alpha_t = 2\kappa^2 \|\nabla_{\boldsymbol{\theta}} \bar{G}_t\|_2^2 - \frac{c_1}{\sqrt{m}} , \qquad (14)$$

*where $\nabla_{\boldsymbol{\theta}} \bar{G}_t = \frac{1}{n} \sum_{i=1}^n \frac{1}{R} \sum_{j=1}^R \nabla_{\boldsymbol{\theta}} G_{\boldsymbol{\theta}_t}(u^{(i)})(x_j)$, and for some constant $c_1 > 0$ which depends polynomially on the depth $L$, and the radii $\rho_w$, $\rho_r$, $\rho_1$, and $\rho_2$ whenever $\sigma_{1,w} + \sigma_{1,r} \leq 1 - \frac{\rho_w + \rho_r}{\sqrt{m}}$. Thus, the loss $\mathcal{L}(\boldsymbol{\theta})$ satisfies RSC w.r.t $(B_\kappa^t, \boldsymbol{\theta}_t)$, i.e., Condition 1(c), whenever $\|\nabla_{\boldsymbol{\theta}} \bar{G}_t\|_2^2 = \Omega(\frac{1}{\sqrt{m}})$.*

**Theorem 5** (**Smoothness for FNOs**). *Under Assumptions 5 and 6, with probability at least $1 - \frac{2(L+2)}{m}$, the FNO loss $\mathcal{L}$ (6) is $\beta$-smooth in $B_{\rho_w,\rho_r\rho_1}^{\text{Euc}}(\boldsymbol{\theta}_0)$ with $\beta$ being a positive constant which depends polynomially on the depth $L$, and the radii $\rho_w$, $\rho_r$, and $\rho_1$ whenever $\sigma_{1,w} + \sigma_{1,r} \leq 1 - \frac{\rho_w + \rho_r}{\sqrt{m}}$.*

**Remark 6** (Ensuring that $\alpha_t/\beta < 1$). Similar to our discussion in Remark 3, we prove that $\alpha_t/\beta < 1$ in Proposition 3 from Appendix E, satisfying the condition required in the statement of Theorem 1. □

**Optimization Under Gradient Descent for FNOs.** We have that Theorem 4 satisfies Condition 1 and Theorem 5 satisfies Condition 2. We also proved that $\alpha_t/\beta < 1$. Thus, when $\|\nabla_{\boldsymbol{\theta}} \bar{G}_t\|_2^2 = \Omega(\frac{1}{\sqrt{m}})$, i.e., $\alpha_t > 0$, a decrease on the loss function by GD is ensured with probability at least $1 - \frac{2(L+2)}{m}$ towards its minimum value taken within the set $B_{\rho_w,\rho_r\rho_1}^{\text{Euc}}(\boldsymbol{\theta}_0)$ due to Theorem 1.

**Remark 7** (The effects of over-parameterization for FNOs). Similar observations to Remarks 4 and 5 hold for FNOs, i.e., that over-parameterization ensures (i) a better condition for ensuring the RSC property, and (ii) a larger neighborhood around the initialization point over which our guarantees hold. Item (ii) follows from the relationship $\rho_w + \rho_r \leq \sqrt{m}$ obtained when choosing $\sigma_{1,w}$ and $\sigma_{1,r}$ to ensure a polynomial dependence as in Theorems 4 and 5.

## 7. Comparison between Neural Operators and Feedforward Neural Networks

Our presented analysis provides sufficient conditions that guarantee the optimization of DONs and FNOs under gradient descent (GD). It is non-trivial that GD should converge for neural operators in a similar way to how it converges for feedforward neural networks (FFNs), i.e., by being particular instances of the general optimization framework from Section 4. Indeed, as indicated in Table 1, there exist similarities and differences between our derivations for neural operators and the ones for FFNs.

**The challenge in the analysis of DONs.** The fact that the output of a DON is an inner product of two FFNs (1)—the branch and trunk networks—makes the mathematical analysis of the RSC and smoothness properties more involved than the analysis associated to a single FFN. Indeed, the appearance of cross-interaction terms between the two FFNs complicates the Hessian structure of the empirical loss $\mathcal{L}$ and requires a more complex definition of the $Q_\kappa^t$ set compared to the one used for FFNs or FNOs. On the other hand, since the branch and trunk networks are *individually* FFNs, their individual Hessian and gradient bounds are known.

**The challenge in the analysis of FNOs.** The fact that FNOs—unlike FFNs—include a series of learnable transformations in the Fourier domain makes the mathematical analyses of their Hessian and gradient bounds more involved than the ones for FFNs. Indeed, these Fourier transformations introduce cross-derivatives between weights in data and Fourier domains in the Hessian that need to be carefully taken into account. On the other hand, since FNOs are composed of a single network, their $Q_\kappa^t$ set is similar to FFNs, as well as their RSC and smoothness analyses.

## 8. Experiments

We present experiments on the effect of over-parameterization on the training performance of DONs and FNOs, as measured by the empirical risk over a mini-batch of the training dataset using the Adam optimizer. We consider three prototypical operator learning problems in the literature (Li et al., 2021a; Lu et al., 2021): (a) the antiderivative (or integral) operator, (b) the diffusion-reaction operator, and (c) Burgers' equation. We do not consider vector-valued problems (e.g., Navier-Stokes) because they are not covered by our theoretical framework. For definiteness, we consider the branch and trunk nets to have the same width $m$ (i.e., $m_f = m_g = m$) for the DON, and the same width $m$ for the FNO. In all experiments, we increase the width from $m = 10$ to $m = 500$. For all networks, we use the Scaled Exponential Linear Unit (SELU) (Klambauer et al., 2017) as their smooth activation function. We monitor the training process over $80,000$ training epochs and report the resulting average loss. Note that the objective of this section is to show the effect of over-parameterization on the neural operator training and not to present any kind of comparison between the two neural operators.

| | Deep Operator Network | Fourier Neural Operator |
|---|---|---|
| $Q_\kappa^t$ set | More Complex | Similar |
| **Hessian and gradient bounds of the neural operator model** | Similar* | More Complex |
| **RSC and Smoothness characterization; computing the Hessian of $\mathcal{L}$** | More Complex | Similar |

Table 1: We indicate whether a specific neural operator (DON or FNO) has a *similar* or a *more complex* derivation of specific mathematical objects or properties compared to a feedforward neural network (as in (Banerjee et al., 2023a)). *The similarity is with respect to each individual network of the DON.

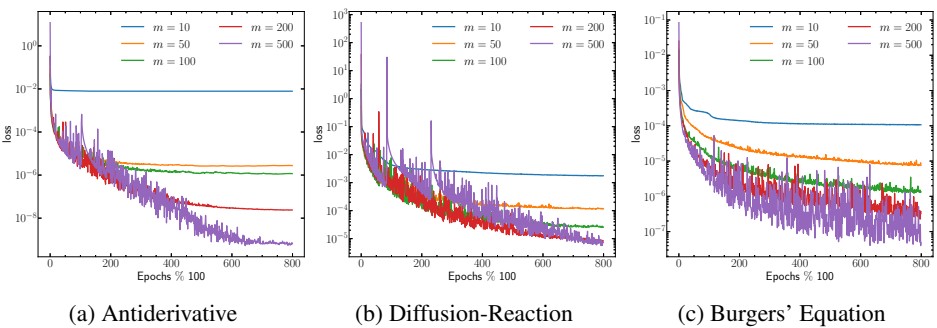

(a) Antiderivative      (b) Diffusion-Reaction      (c) Burgers' Equation

Figure 1: Training progress of DONs as measured by the empirical loss (2) over 80,000 epochs. The $y$-axis is plotted on a *log-scale* and the $x$-axis denotes the training epochs % 100 (i.e., the loss is stored at every 100$^{\text{th}}$ epoch). Wider networks typically lead to lower loss for all three problems.

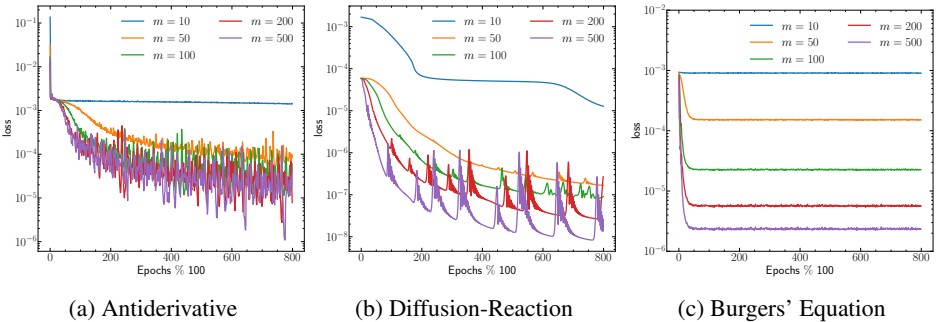

(a) Antiderivative      (b) Diffusion-Reaction      (c) Burgers' Equation

Figure 2: Training progress of FNOs as measured by the empirical loss (5) over 80,000 epochs. The setting of the plots is similar to Figure 1. Wider networks typically lead to lower loss for all three problems.

The results for DONs (Figure 1) and FNOs (Figure 2) clearly show that both neural operators benefit from an increasing width $m$ since it leads to overall lower training losses for all three learning problems and it generally leads to faster optimization convergence. The Antiderivative operator is a linear operator and therefore is learned very accurately, especially for wider DONs and FNOs where the loss is around $10^{-12}$ and $10^{-5}$ respectively. The diffusion-reaction equation demonstrates lower loss with increasing width less markedly than the antiderivative operator for DONs and more markedly for FNOs. This could be attributed in part to the fact that the operator is inherently nonlinear. Finally, regarding Burgers' equation, lower training losses and faster convergence is more markedly for FNOs than for DONs as the width increases.

Additional information on the experimental settings and additional experiments are found in Appendix F.

## 9. Conclusion

We present novel optimization guarantees for gradient descent for neural operators with smooth activations: Deep Operator Networks and Fourier Neural Operators. Our guarantees are based on the restricted strong convexity and smoothness of the loss, thus providing an encompassing framework to neural operator optimization. We argue that increasing the width of the neural operators benefits our theoretical guarantees. We also present empirical evaluations on prototypical operator learning problems to complement our theory.

## Acknowledgements

Part of the work was done when Pedro Cisneros-Velarde was affiliated with the University of Illinois Urbana-Champaign and was concluded during Pedro Cisneros-Velarde's current affiliation. The work was supported in part by the National Science Foundation (NSF) through awards IIS 21-31335, OAC 21-30835, DBI 20-21898, as well as a C3.ai research award.

## Impact Statement

This paper presents work whose goal is to advance the field of Machine Learning. Our work is a theoretical analysis of already existing algorithms, therefore, we do not believe there are potential societal consequences that must be specifically highlighted.

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

# A. Related Work

**Learning Operators.** Constructing operator networks for ordinary differential equations using learning-based approaches was first studied in (Chen & Chen, 1995), where a neural network with a single hidden layer was shown to approximate a nonlinear continuous functional. This was, in essence, akin to the Universal Approximation Theorem for classical neural networks (Cybenko, 1989; Hornik et al., 1989; Hornik, 1991; Lu et al., 2017). While this theorem only guaranteed the existence of a neural architecture, it was not practically realized until (Lu et al., 2021) provided an extension of the theorem to DONs. Since then, several works have pursued applications of DONs to different problems, e.g., (Goswami et al., 2022; Wang & Perdikaris, 2021; Diab & Al Kobaisi, 2024; Centofanti et al., 2024; Sun et al., 2023), as well as improved the DON model itself, e.g., (Wang et al., 2021b; Qiu et al., 2024). From a standpoint of generalization, Kontolati et al. (2022) studied the effects of over-parameterization on the generalization properties of DONs in the context of dynamical systems. Nonetheless, an optimization analysis of DONs is an open problem.

The operator learning paradigm has also been explored in parallel by other works seeking to directly parameterize the integral kernel in the Fourier domain using a deep network (Bhattacharya et al., 2021b;a; Li et al., 2021a; 2020b; 2021b). Several subsequent extensions explored different architectures for Fourier-based operators tailored to specific problems (Li et al., 2020a; Liu et al., 2022a; Wen et al., 2022; Pathak et al., 2022; Centofanti et al., 2024). Other notable techniques include the use of a factorized spectral representation (Tran et al., 2021) and employing FNOs in latent space in an encoder-decoder framework (Li et al., 2023). We also mention that Qin et al. (2024) noticed that FNOs have good performance for learning low-frequency information from PDE data, but are less effective on high-frequency information. Thus, they proposed using an FNO ensemble that induces the neural operator to focus on learning the latter. Recently, FNOs were used to accelerate simulations in climate science (Yang et al., 2023; Harder et al., 2023). Nevertheless, while significant progress has been made for FNOs from an applied perspective, their formal optimization analysis is an open problem.

**Optimization Analysis of Neural Networks.** Optimization of over-parameterized deep neural networks has been studied extensively, e.g., (Du et al., 2019; Arora et al., 2019b;a; Allen-Zhu et al., 2019; Liu et al., 2021a). In particular, Jacot et al. (2018) showed that the NTK of a deep network converges to an explicit kernel in the limit of infinite network width and stays constant during training. Liu et al. (2021a) showed that this constancy arises due to the scaling properties of the Hessian of the predictor as a function of network width. Banerjee et al. (2023b) showed that a deep network whose width is effectively linear on the sample size can ensure convergence under appropriate initialization. Du et al. (2019) and Allen-Zhu et al. (2019) showed that GD converges to zero training error in polynomial time for deep over-parameterized models such as ResNets and CNNs. Karimi et al. (2016) showed that the Polyak-Lojasiewicz (PL) condition, a weaker condition than strong convexity, can be used to explain the linear convergence of gradient-based methods. Banerjee et al. (2023a) showed convergence of GD for feedforward networks using RSC, which leads to a variant of the PL condition. Cisneros-Velarde et al. (2025) used RSC to prove the optimization of networks with weight normalization using GD.

# B. Additional Information on Neural Operators

## B.1. Learning Operators

We briefly outline the notion of learning for neural operators (Li et al., 2021a; 2020b; Lu et al., 2021). Consider two separable Banach spaces, the input space $\mathcal{U}$ and the output space $\mathcal{V}$, and a possibly nonlinear operator $G^{\dagger} : \mathcal{U} \to \mathcal{V}$.

The standard operator learning problem seeks to approximate $G^{\dagger}$ by a parametric operator $G_{\boldsymbol{\theta}} : \mathcal{U} \to \mathcal{V}$ that depends on the parameter vector $\boldsymbol{\theta} \in \Theta$ defined over some parameter space $\Theta$. This is done by proposing an optimization framework where we learn a vector $\boldsymbol{\theta}^{\dagger} \in \Theta$ that "best" approximates $G^{\dagger}$ in some sense. Given observations $\{\boldsymbol{u}^{(j)}\}_{j=1}^{n} \in \mathcal{U}$ and $\{G^{\dagger}(\boldsymbol{u}^{(j)})\}_{j=1}^{n} \in \mathcal{V}$ where $\boldsymbol{u}^{(j)} \sim \mu$, $j = 1, \ldots, n$, is an i.i.d sequence from the probability measure $\mu$ supported on $\mathcal{U}$, we take $\boldsymbol{\theta}^{\dagger}$ as the solution of the minimization problem

$$\boldsymbol{\theta}^{\dagger} = \operatorname{argmin}_{\boldsymbol{\theta} \in \Theta} \mathbb{E}_{\boldsymbol{u} \sim \mu} \left[ \mathcal{C} \left( G_{\boldsymbol{\theta}}(\boldsymbol{u}), G^{\dagger}(\boldsymbol{u}) \right) \right] , \tag{15}$$

where $\mathcal{C}$ is a suitable cost functional that measures the discrepancy on the approximation between the operators $G_{\boldsymbol{\theta}}(\boldsymbol{u})$ and $G^{\dagger}(\boldsymbol{u})$ for a given $\boldsymbol{u} \in \mathcal{U}$. This optimization problem is analogous to the notion of learning in finite dimensions, which is precisely the setup for which classical deep learning is used.

## B.2. DON Architecture

The schematic for the Deep Operator Network's architecture is presented in Figure 3.

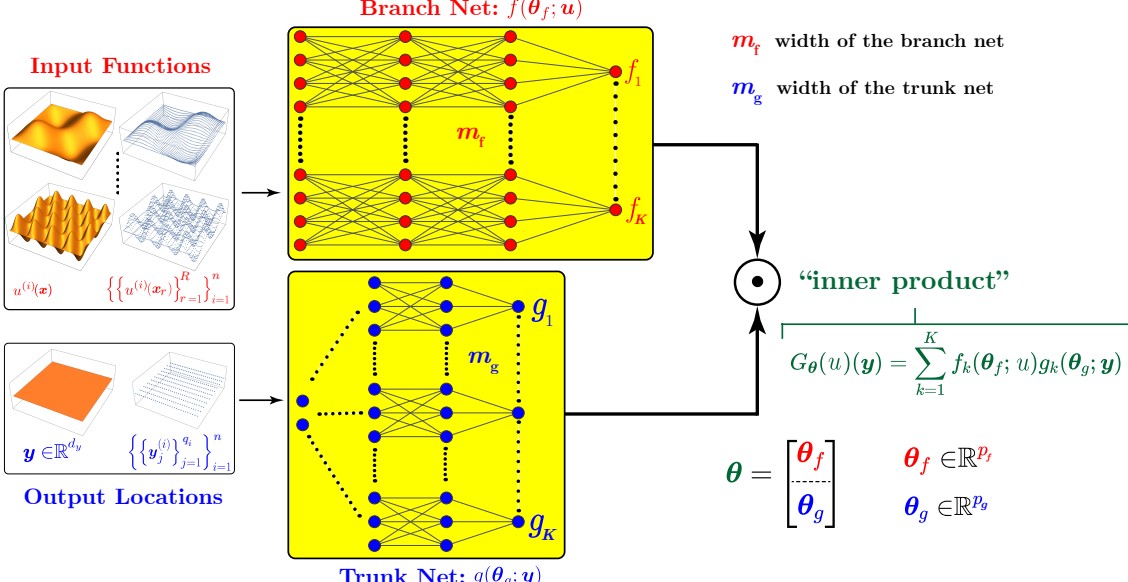

Figure 3: A schematic of the DON architecture by Lu et al. (2021) used in our study. We refer to the notation used in our paper. Note that the input functions need not be sampled on a structured grid of points.

## B.3. FNO Architecture

A schematic for the Fourier Neural Operator's architecture is presented in Figure 4.

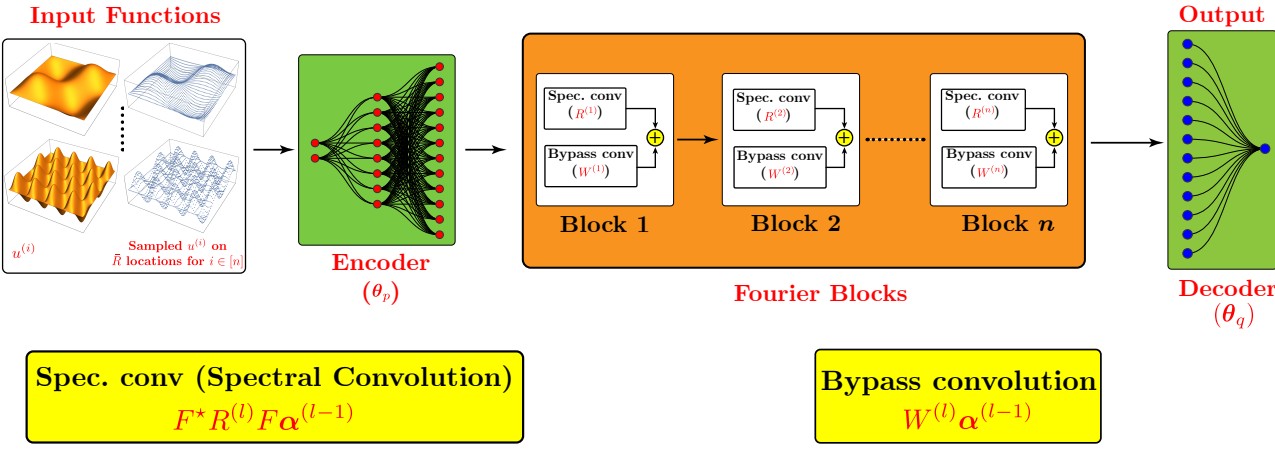

Figure 4: A schematic of the FNO architecture by Li et al. (2021a) used in our study. We refer to the notation used in our paper. "Spectral convolution" and "bypass convolution" are terms used in the FNO literature to denote the effect of the linear mappings in the spectral and spatial domain, respectively.

# C. Optimization Convergence Analysis for Section 4

We establish relevant results for Section 4. Our analysis follows very closely the recent work by Banerjee et al. (2023a) and generalizes it. We now provide all the relevant proofs.

We start with the following lemma which shows that Condition 1 implies a form of restricted PL condition.

**Lemma 1** (**Restricted PL**). *Assume Condition 1 is satisfied. Then, $\mathcal{L}$ satisfies a restricted form of the Polyak-Łojasiewicz (PL) condition w.r.t. $(\mathcal{N}_t, \boldsymbol{\theta}_t)$:*

$$\mathcal{L}(\boldsymbol{\theta}_t) - \inf_{\boldsymbol{\theta} \in \mathcal{N}_t} \mathcal{L}(\boldsymbol{\theta}) \leq \frac{1}{2\alpha_t} \|\nabla_{\boldsymbol{\theta}} \mathcal{L}(\boldsymbol{\theta}_t)\|_2^2 \,. \tag{16}$$

*Proof.* Define

$$\hat{\mathcal{L}}_{\boldsymbol{\theta}_t}(\boldsymbol{\theta}) := \mathcal{L}(\boldsymbol{\theta}_t) + \langle \boldsymbol{\theta} - \boldsymbol{\theta}_t, \nabla_{\boldsymbol{\theta}} \mathcal{L}(\boldsymbol{\theta}_t) \rangle + \frac{\alpha_t}{2} \|\boldsymbol{\theta} - \boldsymbol{\theta}_t\|_2^2 \,.$$

By the $\alpha_t$-RSC property of Condition 1(c), $\forall \boldsymbol{\theta}' \in \mathcal{N}_t$, we have

$$\mathcal{L}(\boldsymbol{\theta}') \geq \hat{\mathcal{L}}_{\boldsymbol{\theta}_t}(\boldsymbol{\theta}') \,. \tag{17}$$

Further, note that $\hat{\mathcal{L}}_{\boldsymbol{\theta}_t}(\cdot)$ is minimized at $\hat{\boldsymbol{\theta}}_{t+1} := \boldsymbol{\theta}_t - \nabla_{\boldsymbol{\theta}} \mathcal{L}(\boldsymbol{\theta}_t)/\alpha_t$ and the minimum value is:

$$\inf_{\boldsymbol{\theta} \in \mathbb{R}^p} \hat{\mathcal{L}}_{\boldsymbol{\theta}_t}(\boldsymbol{\theta}) = \hat{\mathcal{L}}_{\boldsymbol{\theta}_t}(\hat{\boldsymbol{\theta}}_{t+1}) = \mathcal{L}(\boldsymbol{\theta}_t) - \frac{1}{2\alpha_t} \|\nabla_{\boldsymbol{\theta}} \mathcal{L}(\boldsymbol{\theta}_t)\|_2^2 \,.$$

Then, we have that

$$\inf_{\boldsymbol{\theta} \in \mathcal{N}_t} \hat{\mathcal{L}}_{\boldsymbol{\theta}_t}(\boldsymbol{\theta}) \geq \inf_{\boldsymbol{\theta} \in \mathbb{R}^p} \hat{\mathcal{L}}_{\boldsymbol{\theta}_t}(\boldsymbol{\theta}) = \mathcal{L}(\boldsymbol{\theta}_t) - \frac{1}{2\alpha_t} \|\nabla_{\boldsymbol{\theta}} \mathcal{L}(\boldsymbol{\theta}_t)\|_2^2 \,. \tag{18}$$

This means that that $\hat{\mathcal{L}}_{\boldsymbol{\theta}_t}(\cdot)$ is lower bounded by the expression on the right-hand side of (18) and so we can take the infimum over $\mathcal{N}_t$ on both sides of (17) and obtain

$$\inf_{\boldsymbol{\theta} \in \mathcal{N}_t} \mathcal{L}(\boldsymbol{\theta}) \geq \inf_{\boldsymbol{\theta} \in \mathcal{N}_t} \hat{\mathcal{L}}_{\boldsymbol{\theta}_t}(\boldsymbol{\theta}) \,. \tag{19}$$

Finally, we obtain the expression in (16) by using both inequalities in (18) and (19) and rearranging terms. $\square$

Next, we show that the restricted PL condition on $\mathcal{N}_t$ in Lemma 1 along with smoothness (Condition 2) can be used to show a loss reduction on $\mathcal{N}_t$.

**Lemma 2** (**Local loss reduction**). *Assume Conditions 1 and 2 with $\alpha_t \leq \beta$ at step $t$ of the GD update as in (7) with step-size $\eta_t = \frac{\omega_t}{\beta}$ for some $\omega_t \in (0, 2)$. Then, we have*

$$\mathcal{L}(\boldsymbol{\theta}_{t+1}) - \inf_{\boldsymbol{\theta} \in \mathcal{N}_t} \mathcal{L}(\boldsymbol{\theta}) \leq \left(1 - \frac{\alpha_t \omega_t}{\beta}(2 - \omega_t)\right) \left(\mathcal{L}(\boldsymbol{\theta}_t) - \inf_{\boldsymbol{\theta} \in \mathcal{N}_t} \mathcal{L}(\boldsymbol{\theta})\right) \,. \tag{20}$$

*Proof.* Since $\mathcal{L}$ is $\beta$-smooth by Condition 2, we have

$$\begin{aligned}
\mathcal{L}(\boldsymbol{\theta}_{t+1}) &\leq \mathcal{L}(\boldsymbol{\theta}_t) + \langle \boldsymbol{\theta}_{t+1} - \boldsymbol{\theta}_t, \nabla_{\boldsymbol{\theta}} \mathcal{L}(\boldsymbol{\theta}_t) \rangle + \frac{\beta}{2} \|\boldsymbol{\theta}_{t+1} - \boldsymbol{\theta}_t\|_2^2 \\
&= \mathcal{L}(\boldsymbol{\theta}_t) - \eta_t \|\nabla_{\boldsymbol{\theta}} \mathcal{L}(\boldsymbol{\theta}_t)\|_2^2 + \frac{\beta \eta_t^2}{2} \|\nabla_{\boldsymbol{\theta}} \mathcal{L}(\boldsymbol{\theta}_t)\|_2^2 \\
&= \mathcal{L}(\boldsymbol{\theta}_t) - \eta_t \left(1 - \frac{\beta \eta_t}{2}\right) \|\nabla_{\boldsymbol{\theta}} \mathcal{L}(\boldsymbol{\theta}_t)\|_2^2 \,.
\end{aligned} \tag{21}$$

Since $\alpha_t > 0$ by assumption, from Lemma 1 we obtain

$$-\|\nabla_{\boldsymbol{\theta}} \mathcal{L}(\boldsymbol{\theta}_t)\|_2^2 \leq -2\alpha_t(\mathcal{L}(\boldsymbol{\theta}_t) - \inf_{\boldsymbol{\theta} \in \mathcal{N}_t} \mathcal{L}(\boldsymbol{\theta})) \,.$$

Hence

$$\mathcal{L}(\boldsymbol{\theta}_{t+1}) - \inf_{\boldsymbol{\theta} \in \mathcal{N}_t} \mathcal{L}(\boldsymbol{\theta}) \leq \mathcal{L}(\boldsymbol{\theta}_t) - \inf_{\boldsymbol{\theta} \in \mathcal{N}_t} \mathcal{L}(\boldsymbol{\theta}) - \eta_t \left( 1 - \frac{\beta \eta_t}{2} \right) \| \nabla_{\boldsymbol{\theta}} \mathcal{L}(\boldsymbol{\theta}_t) \|_2^2$$

$$\stackrel{(a)}{\leq} \mathcal{L}(\boldsymbol{\theta}_t) - \inf_{\boldsymbol{\theta} \in \mathcal{N}_t} \mathcal{L}(\boldsymbol{\theta}) - \eta_t \left( 1 - \frac{\beta \eta_t}{2} \right) 2 \alpha_t (\mathcal{L}(\boldsymbol{\theta}_t) - \inf_{\boldsymbol{\theta} \in \mathcal{N}_t} \mathcal{L}(\boldsymbol{\theta}))$$

$$= \left( 1 - 2 \alpha_t \eta_t \left( 1 - \frac{\beta \eta_t}{2} \right) \right) (\mathcal{L}(\boldsymbol{\theta}_t) - \inf_{\boldsymbol{\theta} \in \mathcal{N}_t} \mathcal{L}(\boldsymbol{\theta}))$$

where (a) follows for any $\eta_t \leq \frac{2}{\beta}$ because this implies $1 - \frac{\beta \eta_t}{2} \geq 0$. Choosing $\eta_t = \frac{\omega_t}{\beta}, \omega_t \in (0, 2)$,

$$\mathcal{L}(\boldsymbol{\theta}_{t+1}) - \inf_{\boldsymbol{\theta} \in \mathcal{N}_t} \mathcal{L}(\boldsymbol{\theta}) \leq \left( 1 - \frac{\alpha_t \omega_t}{\beta} (2 - \omega_t) \right) (\mathcal{L}(\boldsymbol{\theta}_t) - \inf_{\boldsymbol{\theta} \in \mathcal{N}_t} \mathcal{L}(\boldsymbol{\theta})) .$$

This completes the proof. □

Finally, we show that the local loss reduction result in $\mathcal{N}_t$ from Lemma 2 can be extended to show loss reduction in $\mathcal{B}(\boldsymbol{\theta}_0)$, which is the main optimization result.

**Theorem 1** (**Global loss reduction**). *Consider Assumption 1 and Conditions 1 and 2 with $\alpha_t \leq \beta$ at step t of the GD update (7) with step-size $\eta_t = \frac{\omega_t}{\beta}$ for some $\omega_t \in (0, 2)$. If $\mathcal{L}(\boldsymbol{\theta}_t) \neq \inf_{\boldsymbol{\theta} \in \mathcal{B}(\boldsymbol{\theta}_0)} \mathcal{L}(\boldsymbol{\theta})$, then we have $0 \leq \gamma_t :=$*

*$\frac{\inf_{\boldsymbol{\theta} \in \mathcal{N}_t} \mathcal{L}(\boldsymbol{\theta}) - \inf_{\boldsymbol{\theta} \in \mathcal{B}(\boldsymbol{\theta}_0)} \mathcal{L}(\boldsymbol{\theta})}{\mathcal{L}(\boldsymbol{\theta}_t) - \inf_{\boldsymbol{\theta} \in \mathcal{B}(\boldsymbol{\theta}_0)} \mathcal{L}(\boldsymbol{\theta})} < 1$ and*

$$\mathcal{L}(\boldsymbol{\theta}_{t+1}) - \inf_{\boldsymbol{\theta} \in \mathcal{B}(\boldsymbol{\theta}_0)} \mathcal{L}(\boldsymbol{\theta})$$

$$\leq \left( 1 - \frac{\alpha_t \omega_t (1 - \gamma_t)}{\beta} (2 - \omega_t) \right) (\mathcal{L}(\boldsymbol{\theta}_t) - \inf_{\boldsymbol{\theta} \in \mathcal{B}(\boldsymbol{\theta}_0)} \mathcal{L}(\boldsymbol{\theta})) . \tag{9}$$

*Proof.* We start by showing $\gamma_t = \frac{\inf_{\boldsymbol{\theta} \in \mathcal{N}_t} \mathcal{L}(\boldsymbol{\theta}) - \inf_{\boldsymbol{\theta} \in \mathcal{B}(\boldsymbol{\theta}_0)} \mathcal{L}(\boldsymbol{\theta})}{\mathcal{L}(\boldsymbol{\theta}_t) - \inf_{\boldsymbol{\theta} \in \mathcal{B}(\boldsymbol{\theta}_0)} \mathcal{L}(\boldsymbol{\theta})}$ satisfies $0 \leq \gamma_t < 1$. First of all, we note that this quantity is well-defined because we are assuming that $\mathcal{L}(\boldsymbol{\theta}_t) \neq \inf_{\boldsymbol{\theta} \in \mathcal{B}(\boldsymbol{\theta}_0)} \mathcal{L}(\boldsymbol{\theta})$, i.e., that the current iterate does not attain the minimum loss. The fact that $\gamma_t \geq 0$ follows immediately from $\inf_{\boldsymbol{\theta} \in \mathcal{B}(\boldsymbol{\theta}_0)} \mathcal{L}(\boldsymbol{\theta}) < \mathcal{L}(\boldsymbol{\theta}_t)$ and $\inf_{\boldsymbol{\theta} \in \mathcal{B}(\boldsymbol{\theta}_0)} \mathcal{L}(\boldsymbol{\theta}) \leq \inf_{\boldsymbol{\theta} \in \mathcal{N}_t} \mathcal{L}(\boldsymbol{\theta})$ since $\mathcal{N}_t \subseteq \mathcal{B}(\boldsymbol{\theta}_0)$ by Condition 1(a). Now, there are two ways to prove that $\gamma_t < 1$ depending on whether we consider Condition 1(b.1) or Condition 1(b.2).

We start by considering Condition 1(b.1) and prove by contradiction that $\gamma_t < 1$. Assume that $\gamma_t \geq 1$, i.e., $\inf_{\boldsymbol{\theta} \in \mathcal{N}_t} \mathcal{L}(\boldsymbol{\theta}) \geq \mathcal{L}(\boldsymbol{\theta}_t)$. Then, we note that

$$\inf_{\boldsymbol{\theta} \in \mathcal{N}_t} \mathcal{L}(\boldsymbol{\theta}) \stackrel{(a)}{\leq} \mathcal{L}(\boldsymbol{\theta}_{t+1}) \stackrel{(b)}{\leq} \mathcal{L}(\boldsymbol{\theta}_t) - \eta_t \left( 1 - \frac{\beta \eta_t}{2} \right) \| \nabla_{\boldsymbol{\theta}} \mathcal{L}(\boldsymbol{\theta}_t) \|_2^2 \stackrel{(c)}{\leq} \inf_{\boldsymbol{\theta} \in \mathcal{N}_t} \mathcal{L}(\boldsymbol{\theta}) - \eta_t \left( 1 - \frac{\beta \eta_t}{2} \right) \| \nabla_{\boldsymbol{\theta}} \mathcal{L}(\boldsymbol{\theta}_t) \|_2^2 , \tag{22}$$

where (a) follows from $\boldsymbol{\theta}_{t+1} \in \mathcal{N}_t$, (b) from (21), and (c) from $\gamma_t \geq 1$. Then, comparing the leftmost and rightmost inequalities in (22), we must have $\| \nabla_{\boldsymbol{\theta}} \mathcal{L}(\boldsymbol{\theta}_t) \|_2 = 0$ since $\eta_t \left( 1 - \frac{\beta \eta_t}{2} \right) > 0$ because of $\eta_t = \frac{\omega_t}{\beta}$ with $\omega_t \in (0, 2)$. Now, when considering Condition 1(b.1), we either assume that $\boldsymbol{\theta} \notin \mathcal{N}_t$ or that $\mathcal{L}(\boldsymbol{\theta}_t) \neq \inf_{\boldsymbol{\theta} \in \mathcal{N}_t} \mathcal{L}(\boldsymbol{\theta})$; thus, we analyze both cases.

(i) **Assuming $\boldsymbol{\theta} \notin \mathcal{N}_t$:** Note that $\| \nabla_{\boldsymbol{\theta}} \mathcal{L}(\boldsymbol{\theta}_t) \|_2 = 0$ implies $\nabla_{\boldsymbol{\theta}} \mathcal{L}(\boldsymbol{\theta}_t) = \mathbf{0}_p$ (i.e., the gradient evaluated at $\boldsymbol{\theta}_t$ is the zero vector), which then, due to GD in (7), implies $\boldsymbol{\theta}_t = \boldsymbol{\theta}_{t+1}$. Since we had $\boldsymbol{\theta}_{t+1} \in \mathcal{N}_t$, this then means that $\boldsymbol{\theta}_t \in \mathcal{N}_t$—a contradiction to our assumption.

(ii) **Assuming $\mathcal{L}(\boldsymbol{\theta}_t) \neq \inf_{\boldsymbol{\theta} \in \mathcal{N}_t} \mathcal{L}(\boldsymbol{\theta})$:** Note that $\| \nabla_{\boldsymbol{\theta}} \mathcal{L}(\boldsymbol{\theta}_t) \|_2 = 0$ implies that all the inequalities in (22) are also equalities. This then implies that $\mathcal{L}(\boldsymbol{\theta}_t) = \inf_{\boldsymbol{\theta} \in \mathcal{N}_t} \mathcal{L}(\boldsymbol{\theta})$—a contradiction to our assumption.

In either case (i) or (ii), our proof by contradiction finally shows that $\gamma_t < 1$.

We now consider Condition 1(b.2) with the element $\theta' \in \mathcal{N}_t$ as described in the condition's statement. We immediately have that $\theta'$ satisfies $\inf_{\theta \in \mathcal{N}_t} \mathcal{L}(\theta) \leq \mathcal{L}(\theta') < \mathcal{L}(\theta_t)$, which then implies $\gamma_t < 1$.

Having shown that $\gamma_t < 1$ according to Condition 1(b), we now proceed to prove equation (9). We consider two cases: (A) $\gamma_t > 0$ and (B) $\gamma_t = 0$.

We start by considering Case (A), which holds if and only if $\inf_{\theta \in \mathcal{N}_t} \mathcal{L}(\theta) > \inf_{\theta \in \mathcal{B}(\theta_0)} \mathcal{L}(\theta)$. We now define $\delta_t := \frac{\mathcal{L}(\theta_t) - \inf_{\theta \in \mathcal{N}_t} \mathcal{L}(\theta)}{\mathcal{L}(\theta_t) - \inf_{\theta \in \mathcal{B}(\theta_0)} \mathcal{L}(\theta)}$ and note that $\delta_t \in (0,1)$ since $\delta_t = 1 - \gamma_t$. Now, with $\omega_t \in (0,2)$, we have

$$
\begin{aligned}
\mathcal{L}(\theta_{t+1}) - \inf_{\theta \in \mathcal{B}(\theta_0)} \mathcal{L}(\theta) &= \mathcal{L}(\theta_{t+1}) - \inf_{\theta \in \mathcal{N}_t} \mathcal{L}(\theta) + \inf_{\theta \in \mathcal{N}_t} \mathcal{L}(\theta) - \inf_{\theta \in \mathcal{B}(\theta_0)} \mathcal{L}(\theta) \\
&\overset{(a)}{\leq} \left(1 - \frac{\alpha_t \omega_t}{\beta}(2 - \omega_t)\right)\left(\mathcal{L}(\theta_t) - \inf_{\theta \in \mathcal{N}_t} \mathcal{L}(\theta)\right) + \left(\inf_{\theta \in \mathcal{N}_t} \mathcal{L}(\theta) - \inf_{\theta \in \mathcal{B}(\theta_0)} \mathcal{L}(\theta)\right) \\
&= \left(1 - \frac{\alpha_t \omega_t}{\beta}(2 - \omega_t)\right)\delta_t\left(\mathcal{L}(\theta_t) - \inf_{\theta \in \mathcal{B}(\theta_0)} \mathcal{L}(\theta)\right) + (1 - \delta_t)\left(\mathcal{L}(\theta_t) - \inf_{\theta \in \mathcal{B}(\theta_0)} \mathcal{L}(\theta)\right) \\
&\overset{(b)}{=} \left(1 - \frac{\alpha_t \omega_t}{\beta}(2 - \omega_t)(1 - \gamma_t)\right)\left(\mathcal{L}(\theta_t) - \inf_{\theta \in \mathcal{B}(\theta_0)} \mathcal{L}(\theta)\right),
\end{aligned}
$$

which is (9), and where (a) follows from Lemma 2 and (b) follows from

$$
\left(1 - \frac{\alpha_t \omega_t}{\beta}(2 - \omega_t)\right)\delta_t + (1 - \delta_t) = 1 - \frac{\alpha_t \omega_t}{\beta}(2 - \omega_t)\delta_t = 1 - \frac{\alpha_t \omega_t}{\beta}(2 - \omega_t)(1 - \gamma_t).
$$

We now consider Case (B), i.e., $\gamma_t = 0$, which holds if and only if $\inf_{\theta \in \mathcal{N}_t} \mathcal{L}(\theta) = \inf_{\theta \in \mathcal{B}(\theta_0)} \mathcal{L}(\theta)$. Then, we have

$$
\begin{aligned}
\mathcal{L}(\theta_{t+1}) - \inf_{\theta \in \mathcal{B}(\theta_0)} \mathcal{L}(\theta) &= \mathcal{L}(\theta_{t+1}) - \inf_{\theta \in \mathcal{N}_t} \mathcal{L}(\theta) \\
&\overset{(a)}{\leq} \left(1 - \frac{\alpha_t \omega_t}{\beta}(2 - \omega_t)\right)\left(\mathcal{L}(\theta_t) - \inf_{\theta \in \mathcal{N}_t} \mathcal{L}(\theta)\right) \\
&= \left(1 - \frac{\alpha_t \omega_t}{\beta}(2 - \omega_t)\right)\left(\mathcal{L}(\theta_t) - \inf_{\theta \in \mathcal{B}(\theta_0)} \mathcal{L}(\theta)\right),
\end{aligned}
$$

which is (9) when $\gamma_t = 0$, and where (a) follows from Lemma 2. This completes the proof. $\square$

## D. Analysis for Deep Operator Networks

### D.1. Bounds on the Hessian, Gradients and the Predictor

The convergence analysis makes use of the gradients and Hessians of the empirical loss with respect to the parameters $\theta$, namely,

$$
\nabla_\theta \mathcal{L}(\theta) = \left[\nabla_{\theta_f} \mathcal{L}^\top \ \nabla_{\theta_g} \mathcal{L}^\top\right]^\top, \quad \text{and} \quad \nabla_\theta^2 \mathcal{L}(\theta) = H(\theta) = \begin{bmatrix} H_{ff} & H_{fg} \\ H_{gf} & H_{gg} \end{bmatrix}, \tag{23}
$$

where $\nabla_{\theta_f} \mathcal{L}(\theta) = \partial \mathcal{L}(\theta)/\partial \theta_f \in \mathbb{R}^{p_f}$ and $\nabla_{\theta_g} \mathcal{L}(\theta) = \partial \mathcal{L}(\theta)/\partial \theta_g \in \mathbb{R}^{p_g}$. Note that we make use of the notation $\nabla_{\theta_f}(\cdot)$ to denote the derivative with respect to the parameters $\theta_f$ and this *is not* a functional gradient. Similarly, the individual blocks in the $2 \times 2$ block Hessian $H(\theta)$ are given by

$$
H_{ff} = \nabla_{\theta_f}^2 \mathcal{L} = \frac{\partial^2 \mathcal{L}}{\partial \theta_f{}^2}, \quad H_{fg} = \frac{\partial^2 \mathcal{L}}{\partial \theta_f \partial \theta_g}, \quad H_{gf} = H_{fg}^\top = \frac{\partial^2 \mathcal{L}}{\partial \theta_g \partial \theta_f}, \quad H_{gg} = \nabla_{\theta_g}^2 \mathcal{L} = \frac{\partial^2 \mathcal{L}}{\partial \theta_g{}^2}, \tag{24}
$$

where $H_{ff} \in \mathbb{R}^{p_f \times p_f}$, $H_{gg} \in \mathbb{R}^{p_g \times p_g}$, $H_{fg} \in \mathbb{R}^{p_f \times p_g}$, $H_{gf} \in \mathbb{R}^{p_g \times p_f}$ and the argument $\theta$ is ignored for clarity of exposition. Using (3) and rewriting the derivatives in (23) and (24), recalling that $\ell_{i,j} = (G_\theta(u^{(i)})(y_j^{(i)}) - G^\dagger(u^{(i)})(y_j^{(i)}))^2$,

we get

$$\frac{\partial \mathcal{L}}{\partial \boldsymbol{\theta}_f} = \frac{1}{n} \sum_{i=1}^{n} \frac{1}{q_i} \sum_{j=1}^{q_i} \ell'_{i,j} \sum_{k=1}^{K} g_{k,j}^{(i)} \nabla_{\boldsymbol{\theta}_f} f_k^{(i)} \quad \text{and} \quad \frac{\partial \mathcal{L}}{\partial \boldsymbol{\theta}_g} = \frac{1}{n} \sum_{i=1}^{n} \frac{1}{q_i} \sum_{j=1}^{q_i} \ell'_{ij} \sum_{k=1}^{K} f_k^{(i)} \nabla_{\boldsymbol{\theta}_g} g_{k,j}^{(i)}, \tag{25}$$

$$\frac{\partial^2 \mathcal{L}}{\partial \boldsymbol{\theta}_f{}^2} = \frac{1}{n} \sum_{i=1}^{n} \frac{1}{q_i} \sum_{j=1}^{q_i} \ell'_{i,j} \sum_{k=1}^{K} g_{k,j}^{(i)} \nabla_{\boldsymbol{\theta}_f}^2 f_k^{(i)} + \frac{1}{n} \sum_{i=1}^{n} \frac{1}{q_i} \sum_{j=1}^{q_i} \ell''_{i,j} \left( \sum_{k,k'=1}^{K} g_{k,j}^{(i)} g_{k',j}^{(i)} \nabla_{\boldsymbol{\theta}_f} f_k^{(i)} \nabla_{\boldsymbol{\theta}_f} f_{k'}^{(i)\top} \right),$$

$$\frac{\partial^2 \mathcal{L}}{\partial \boldsymbol{\theta}_g{}^2} = \frac{1}{n} \sum_{i=1}^{n} \frac{1}{q_i} \sum_{j=1}^{q_i} \ell'_{i,j} \sum_{k=1}^{k} f_k^{(i)} \nabla_{\boldsymbol{\theta}_g}^2 g_{k,j}^{(i)} + \frac{1}{n} \sum_{i=1}^{n} \frac{1}{q_i} \sum_{j=1}^{q_i} \ell''_{i,j} \left( \sum_{k,k'=1}^{K} f_k^{(i)} f_{k'}^{(i)} \nabla_{\boldsymbol{\theta}_g} g_{k,j}^{(i)} \nabla_{\boldsymbol{\theta}_g} g_{k',j}^{(i)\top} \right), \tag{26}$$

$$\frac{\partial^2 \mathcal{L}}{\partial \boldsymbol{\theta}_f \partial \boldsymbol{\theta}_g} = \frac{1}{n} \sum_{i=1}^{n} \frac{1}{q_i} \sum_{j=1}^{q_i} \ell'_{i,j} \sum_{k=1}^{K} \nabla_{\boldsymbol{\theta}_f} f_k^{(i)} \nabla_{\boldsymbol{\theta}_g} g_{k,j}^{(i)\top} + \frac{1}{n} \sum_{i=1}^{n} \frac{1}{q_i} \sum_{j=1}^{q_i} \ell''_{i,j} \left( \sum_{k,k'=1}^{K} g_{k,j}^{(i)} f_{k'}^{(i)} \nabla_{\boldsymbol{\theta}_f} f_k^{(i)} \nabla_{\boldsymbol{\theta}_g} g_{k',j}^{(i)\top} \right),$$

for the individual blocks of the Hessian (23) where we make use of the notation $f_k^{(i)} = f_k(\boldsymbol{\theta}_f; u^{(i)})$ and $g_{k,j}^{(i)} = g_k(\boldsymbol{\theta}_g; \boldsymbol{y}_j^{(i)})$. In the rest of the paper, with some abuse of notation, we also make use of the implicit notation $f_k^{(i)}(\boldsymbol{\theta}_f) = f_k(\boldsymbol{\theta}_f; u^{(i)})$ and $g_{k,j}^{(i)}(\boldsymbol{\theta}_g) = g_k(\boldsymbol{\theta}_g; \boldsymbol{y}_j^{(i)})$.

In order to prove the RSC and smoothness properties of the empirical loss $\mathcal{L}$, we need to upper bound the spectral norm of its Hessian. As can be seen above, the gradient and Hessian of the predictors (i.e., the branch $f_k^{(i)}$ and trunk $g_{k,j}^{(i)}$ networks, $k \in [K]$, $j \in [q_i]$, $i \in [n]$) appear in the Hessian of $\mathcal{L}$, and thus, we will eventually need the upper bound of their norms. For this, we will make use of the next lemma.

**Lemma 3** (**Bounds on the predictors**). *Under Assumptions 2 and 3, and for $\boldsymbol{\theta} \in B_{\rho,\rho_1}^{\mathrm{Euc}}(\boldsymbol{\theta}_0)$, with probability at least $1 - 2KL\left(\frac{1}{m_f} + \frac{1}{m_g}\right)$, we have for every $k \in [K]$, $i \in [n]$, $j \in [q_i]$,*

$$\left\| \nabla_{\boldsymbol{\theta}_f}^2 f_k^{(i)} \right\| \leq \frac{c^{(f)}}{\sqrt{m_f}} \quad \text{and} \quad \left\| \nabla_{\boldsymbol{\theta}_g}^2 g_{k,j}^{(i)} \right\| \leq \frac{c^{(g)}}{\sqrt{m_g}} \,,$$
$$\left\| \nabla_{\boldsymbol{\theta}_f} f_k^{(i)} \right\|_2 \leq \varrho^{(f)} \quad \text{and} \quad \left\| \nabla_{\boldsymbol{\theta}_g} g_{k,j}^{(i)} \right\|_2 \leq \varrho^{(g)} \,, \tag{27}$$
$$|f_k^{(i)}| \leq \lambda_1, \quad \text{and} \quad |g_{k,j}^{(i)}| \leq \lambda_2 \,,$$

*where $c^{(f)}$, $c^{(g)}$, $\varrho^{(f)}$, $\varrho^{(g)}$, $\lambda_1$, and $\lambda_2$ are suitable constants that depend on $\sigma_0$, the depth $L$ and the radii $\rho$, $\rho_1$. The dependence of the constants reduces to the depth and the radii and becomes polynomial whenever $\sigma_0 \leq 1 - \rho \max\{\frac{1}{\sqrt{m_f}}, \frac{1}{\sqrt{m_g}}\}$.*

*Proof.* The proof follows from a direct adaptation of Theorem 4.1 and of both the statement and proof of Lemma 4.1 in (Banerjee et al., 2023a) to our setting. Indeed, these results show that $c^{(f)}$, $\varrho^{(f)}$, and $\lambda_1$ depend on $\sigma_0$, the depth $L$ and the radii $\rho$, $\rho_1$; and that such dependence reduces to the depth and the radii and becomes polynomial whenever $\sigma_0 \leq 1 - \frac{\rho}{\sqrt{m_f}}$. A similar dependence is obtained for the constants $c^{(g)}$, $\varrho^{(g)}$, and $\lambda_2$ whenever $\sigma_0 \leq 1 - \frac{\rho}{\sqrt{m_g}}$. The last statement in Lemma 27 follows immediately. Finally, since the bound for a single branch network output holds with probability at least $1 - \frac{2L}{m_f}$ and for a single trunk network output holds with probability at least $1 - \frac{2L}{m_g}$, then in order for these bounds to hold for the $K$ outputs of all predictors, we obtain the overall probability using De Morgan's law and a union bound. $\square$

### D.2. RSC and Smoothness Results

Using the results from the previous section, we derive the RSC and smoothness results.

**Theorem 2** (**RSC for DONs**). *Consider Assumptions 2, 3, and 4, and $Q_\kappa^t$ as in Definition 2 with $\|\nabla_{\boldsymbol{\theta}} \bar{G}_t\|_2 \neq 0$. Then, the set $B_\kappa^t := Q_\kappa^t \cap B_{\rho,\rho_1}^{\mathrm{Euc}}(\boldsymbol{\theta}_0) \cap B_{\rho_2}^{\mathrm{Euc}}(\boldsymbol{\theta}_t)$ is a non-empty set that satisfies Condition 1(a) and (b) for suitable $\rho_2$ and*

*sufficiently small $\kappa \in (0,1)$. Moreover, with probability at least $1 - 2KL(\frac{1}{m_f} + \frac{1}{m_g})$, at step $t$ of GD, the DON loss $\mathcal{L}$ (3) satisfies equation (8) with*

$$\alpha_t = 2\kappa^2 \|\nabla_{\boldsymbol{\theta}} \bar{G}_t\|_2^2 - c_1 K^2 \left( \frac{1}{\sqrt{m_f}} + \frac{1}{\sqrt{m_g}} \right) \tag{13}$$

*where $\nabla_{\boldsymbol{\theta}} \bar{G}_t = \frac{1}{n} \sum_{i=1}^n \frac{1}{q_i} \sum_{j=1}^{q_i} \nabla_{\boldsymbol{\theta}} G_{\boldsymbol{\theta}_t}(u^{(i)})(\boldsymbol{y}_j^{(i)})$, and for some constant $c_1 > 0$ which depends polynomially on the depth $L$, and the radii $\rho$, $\rho_1$, and $\rho_2$ whenever $\sigma_0 \leq 1 - \rho \max\{\frac{1}{\sqrt{m_f}}, \frac{1}{\sqrt{m_g}}\}$. Thus, the loss $\mathcal{L}$ satisfies RSC w.r.t $(B_\kappa^t, \boldsymbol{\theta}_t)$, i.e., Condition 1(c), whenever $\|\nabla_{\boldsymbol{\theta}} \bar{G}_t\|_2^2 = \Omega(\frac{1}{\sqrt{m_f}} + \frac{1}{\sqrt{m_g}})$.*

*Proof.* We start by proving the first part of the theorem's statement. Since $B_\kappa^t \subset B_{\rho,\rho_1}^{\text{Euc}}(\boldsymbol{\theta}_0)$, we satisfy Condition 1(a). We now need to satisfy Condition 1(b). We particularly focus on Condition 1(b.2). For this, we first show the existence of an element $\boldsymbol{\theta}' \in B_\kappa^t$. For such $\boldsymbol{\theta}'$, it must be true that $\boldsymbol{\theta}' \in Q_\kappa^t$. From Definition 2, $\boldsymbol{\theta}'$ needs to satisfy three conditions:

$$|\cos(\boldsymbol{\theta}' - \boldsymbol{\theta}_t, \nabla_{\boldsymbol{\theta}} \bar{G}_t)| \geq \kappa \quad \text{(cosine similarity condition)},$$

$$(\boldsymbol{\theta}_f' - \boldsymbol{\theta}_{f,t})^\top \left( \frac{1}{n} \sum_{i=1}^n \frac{1}{q_i} \sum_{j=1}^{q_i} \ell_{i,j}' \sum_{k=1}^K \nabla_{\boldsymbol{\theta}_f} f_k^{(i)} \nabla_{\boldsymbol{\theta}_g} g_{k,j}^{(i)\top} \right) (\boldsymbol{\theta}_g' - \boldsymbol{\theta}_{g,t}) \geq 0 \quad \text{(average condition)},$$

$$(\boldsymbol{\theta}_f' - \boldsymbol{\theta}_{f,t})^\top \left( \sum_{k=1}^K \nabla_{\boldsymbol{\theta}_f} f_k^{(i)} \nabla_{\boldsymbol{\theta}_g} g_{k,j}^{(i)\top} \right) (\boldsymbol{\theta}_g' - \boldsymbol{\theta}_{g,t}) \leq 0, \forall i \in [n], \forall j \in [q_i] \quad \text{(output condition)}.$$

Let $\nabla_{\boldsymbol{\theta}_f} \bar{G}_t$ be the first $p_f$ components of $\nabla_{\boldsymbol{\theta}} \bar{G}_{\boldsymbol{\theta}_t}$, and let $\nabla_{\boldsymbol{\theta}_g} \bar{G}_t$ be the last $p_g$ components of $\nabla_{\boldsymbol{\theta}} \bar{G}_{\boldsymbol{\theta}_t}$.

We start by assuming that $\|\nabla_{\boldsymbol{\theta}_f} \bar{G}_t\|_2 \neq 0$. Then, let us consider $\boldsymbol{\theta}' = [\boldsymbol{\theta}_f'^\top \ \boldsymbol{\theta}_g'^\top]^\top$, where $\boldsymbol{\theta}_f' \in \mathbb{R}^{p_f}$ will be specified later and $\boldsymbol{\theta}_g' = \boldsymbol{\theta}_{g,t}$. Then, belonging to the $Q_\kappa^t$ set conveniently reduces to the feasibility of the cosine similarity condition as follows:

$$|\cos(\boldsymbol{\theta}' - \boldsymbol{\theta}_t, \nabla_{\boldsymbol{\theta}} \bar{G}_t)| \geq \kappa. \tag{28}$$

Now, let $c_t$ be defined as $\|\nabla_{\boldsymbol{\theta}} \bar{G}_t\|_2 = c_t \|\nabla_{\boldsymbol{\theta}_f} \bar{G}_t\|_2$. We immediately notice that $c_t > 0$. Moreover,

$$|\cos(\boldsymbol{\theta}' - \boldsymbol{\theta}_t, \nabla_{\boldsymbol{\theta}} \bar{G}_t)| = \left| \left( \frac{\boldsymbol{\theta}_f' - \boldsymbol{\theta}_{f,t}}{\|\boldsymbol{\theta}_f' - \boldsymbol{\theta}_{f,t}\|_2} \right)^\top \left( \frac{\nabla_{\boldsymbol{\theta}_f} \bar{G}_t}{\|\nabla_{\boldsymbol{\theta}} \bar{G}_t\|_2} \right) \right| = \frac{1}{c_t} \left| \left( \frac{\boldsymbol{\theta}_f' - \boldsymbol{\theta}_{f,t}}{\|\boldsymbol{\theta}_f' - \boldsymbol{\theta}_{f,t}\|_2} \right)^\top \left( \frac{\nabla_{\boldsymbol{\theta}_f} \bar{G}_t}{\|\nabla_{\boldsymbol{\theta}_f} \bar{G}_t\|_2} \right) \right|, \tag{29}$$

and thus, we immediately have that (28) can be expressed as

$$|\cos(\boldsymbol{\theta}_f' - \boldsymbol{\theta}_{f,t}, \nabla_{\boldsymbol{\theta}_f} \bar{G}_t)| \geq c_t \kappa. \tag{30}$$

We now choose $\kappa > 0$ sufficiently small so that $c_t \kappa < 1$.

With all of this in mind, we proceed to show the existence of an element $\boldsymbol{\theta}' \in B_\kappa^t$ of the form $\boldsymbol{\theta}' = [\boldsymbol{\theta}_f'^\top \ \boldsymbol{\theta}_{g,t}^\top]^\top$ satisfying condition (30) and the following two conditions:

Condition (A): $\|\boldsymbol{\theta}_f' - \boldsymbol{\theta}_{f,t}\|_2 = \epsilon$ for some $\epsilon < \frac{2\|\nabla_{\boldsymbol{\theta}_f} \mathcal{L}(\boldsymbol{\theta}_t)\|_2 \sqrt{1-(c_t\kappa)^2}}{\beta}$; and

Condition (B): the angle $\nu'$ between $(\boldsymbol{\theta}_f' - \boldsymbol{\theta}_{f,t})$ and $-\nabla_{\boldsymbol{\theta}_f} \mathcal{L}(\boldsymbol{\theta}_t)$ is acute, so that $\cos(\nu') > 0$.

To show the existence of such element $\boldsymbol{\theta}' \in B_\kappa^t$, we propose two possible constructions:

Choice (A): If the points $\boldsymbol{\theta}_{f,t+1}$, $\nabla_{\boldsymbol{\theta}_f} \bar{G}_t + \boldsymbol{\theta}_{f,t}$, and $\boldsymbol{\theta}_{f,t}$ are not collinear, then they define a hyperplane $\mathcal{P}$ that contains the vectors $\nabla_{\boldsymbol{\theta}_f} \bar{G}_t$ and $-\nabla_{\boldsymbol{\theta}_f} \mathcal{L}(\boldsymbol{\theta}_t)$ (recall that $\boldsymbol{\theta}_{f,t+1} - \boldsymbol{\theta}_{f,t} = -\nabla_{\boldsymbol{\theta}_f} \mathcal{L}(\boldsymbol{\theta}_t)$ by gradient descent). We choose $\boldsymbol{\theta}_f'$ such that the vector $\boldsymbol{\theta}_f' - \boldsymbol{\theta}_{f,t}$ lies in $\mathcal{P}$ with $\cos(\boldsymbol{\theta}_f' - \boldsymbol{\theta}_{f,t}, \nabla_{\boldsymbol{\theta}_f} \bar{G}_t) = c_t \kappa$ (i.e., it satisfies condition (30) with equality) while simultaneously satisfying Condition (B). If the points $\boldsymbol{\theta}_{f,t+1}$, $\nabla_{\boldsymbol{\theta}_f} \bar{G}_t + \boldsymbol{\theta}_{f,t}$, and $\boldsymbol{\theta}_{f,t}$ are

collinear, we choose $\boldsymbol{\theta}'_f$ such that it is not collinear with these points, thus defining a hyperplane $\mathcal{P}$ with these other three points, and such that $\boldsymbol{\theta}'_f$ is also taken so that $\cos(\boldsymbol{\theta}'_f - \boldsymbol{\theta}_{f,t}, \nabla_{\boldsymbol{\theta}_f}\bar{G}_t) = c_t\kappa$ while simultaneously satisfying Condition (B).

Thus far we have only defined *angle* (or *direction*) conditions on the vector $\boldsymbol{\theta}'_f - \boldsymbol{\theta}_{f,t}$, and so there could be an infinite number of values for $\boldsymbol{\theta}'_f$ satisfying such angle conditions without $\boldsymbol{\theta}'$ belonging to the set $B^{\text{Euc}}_{\rho,\rho_1}(\boldsymbol{\theta}_0)$ nor $\boldsymbol{\theta}'_f$ satisfying Condition (A). To determine the feasible values for $\boldsymbol{\theta}'_f$, we observe that $\boldsymbol{\theta}_t$ is *strictly inside* the set $B^{\text{Euc}}_{\rho,\rho_1}(\boldsymbol{\theta}_0)$ by Assumption 4, and so $\boldsymbol{\theta}'_f$ can be taken arbitrarily close to $\boldsymbol{\theta}_{f,t}$ so that $\boldsymbol{\theta}' \in B^{\text{Euc}}_{\rho,\rho_1}(\boldsymbol{\theta}_0)$ and Condition (A) is satisfied.

We remark that, regardless of the collinearity of the points $\boldsymbol{\theta}_{f,t+1}$, $\nabla_{\boldsymbol{\theta}_f}\bar{G}_t + \boldsymbol{\theta}_{f,t}$, and $\boldsymbol{\theta}_{f,t}$, hyperplane $\mathcal{P}$ contains the vectors $\boldsymbol{\theta}'_f - \boldsymbol{\theta}_{f,t}$, $\nabla_{\boldsymbol{\theta}_f}\bar{G}_t$, and $-\nabla_{\boldsymbol{\theta}_f}\mathcal{L}(\boldsymbol{\theta}_t)$, all sharing its origin at $\boldsymbol{\theta}_{f,t} \in \mathcal{P}$.

Choice (B): We choose $\boldsymbol{\theta}'$ as in Choice (A) but with $\nabla_{\boldsymbol{\theta}_f}\bar{G}_t$ replaced by $-\nabla_{\boldsymbol{\theta}_f}\bar{G}_t$.

We immediately notice that $\boldsymbol{\theta}'$ defined by either Choice (A) or Choice (B) satisfies $\boldsymbol{\theta}' \in Q^t_\kappa \cap B^{\text{Euc}}_{\rho,\rho_1}(\boldsymbol{\theta}_0)$. To make $\boldsymbol{\theta}'$ belong to the set $B^t_\kappa$, we need to find a radius $\rho_2$ such that $\boldsymbol{\theta}' \in B^{\text{Euc}}_{\rho_2}(\boldsymbol{\theta}_t)$, or, equivalently, such that $\boldsymbol{\theta}'_f \in B^{\text{Euc}}_{\rho_2}(\boldsymbol{\theta}_{f,t})$ due to our construction of $\boldsymbol{\theta}'$. Such $\rho_2$ is found by taking $\rho_2 > \epsilon$ with $\epsilon$ as in Condition (A). Finally, it is straightforward to verify that such $\boldsymbol{\theta}' \in B^t_\kappa$ defined by either Choice (A) or Choice (B) will always exist, by considering the following cases for the angle $\nu$ between $\nabla_{\boldsymbol{\theta}_f}\bar{G}_t$ and $-\nabla_{\boldsymbol{\theta}_f}\mathcal{L}(\boldsymbol{\theta}_t)$:

(i) If $\nu \in [0, \pi/2]$ or $\nu \in [3\pi/2, 2\pi]$, then Choice (A) will be true, since $-\nabla_{\boldsymbol{\theta}_f}\mathcal{L}(\boldsymbol{\theta}_t)$ is in the positive half space[1] of $\nabla_{\boldsymbol{\theta}_f}\bar{G}_t$; and

(ii) if $\nu \in [\pi/2, \pi]$ or $\nu \in [\pi, 3\pi/2]$, then Choice (B) will be true, since $-\nabla_{\boldsymbol{\theta}_f}\mathcal{L}(\boldsymbol{\theta}_t)$ is in the positive half space of $-\nabla_{\boldsymbol{\theta}_f}\bar{G}_t$.

Now, let us assume we are in the case of item (i) above, so that $\boldsymbol{\theta}'$ is constructed according to Choice (A) (the rest of the proof can be adapted to the case of Choice (B) and to item (ii) by using a symmetrical argument and so it is omitted). Let $\nu_1$ be the angle between $\boldsymbol{\theta}'_f - \boldsymbol{\theta}_{f,t}$ and $\nabla_{\boldsymbol{\theta}_f}\bar{G}_t$, so that $\cos(\nu_1) = c_t\kappa$ according to Choice (A). Then, we have that

$$|\cos(\nu')| = |\cos(\nu - \nu_1)| \geq |\cos(\pi/2 - \nu_1)| = |\sin(\nu_1)| = \sqrt{1 - \cos^2(\nu_1)} = \sqrt{1 - (c_t\kappa)^2} .$$

Further, by the construction in Condition (B), $\cos(\nu') > 0$, which implies $\cos(\nu') \geq \sqrt{1 - (c_t\kappa)^2} > 0$. Now, by the smoothness property of the empirical loss $\mathcal{L}$ we have

$$\begin{aligned}
\mathcal{L}(\boldsymbol{\theta}') &\leq \mathcal{L}(\boldsymbol{\theta}_t) - \langle \boldsymbol{\theta}' - \boldsymbol{\theta}_t, -\nabla_{\boldsymbol{\theta}}\mathcal{L}(\boldsymbol{\theta}_t)\rangle + \frac{\beta}{2}\|\boldsymbol{\theta}' - \boldsymbol{\theta}_t\|^2_2 \\
&= \mathcal{L}(\boldsymbol{\theta}_t) - \|\boldsymbol{\theta}'_f - \boldsymbol{\theta}_{f,t}\|_2 \|\nabla_{\boldsymbol{\theta}_f}\mathcal{L}(\boldsymbol{\theta}_t)\|_2 \cos(\nu') + \frac{\beta}{2}\|\boldsymbol{\theta}'_f - \boldsymbol{\theta}_{f,t}\|^2_2 \\
&= \mathcal{L}(\boldsymbol{\theta}_t) - \epsilon\|\nabla_{\boldsymbol{\theta}_f}\mathcal{L}(\boldsymbol{\theta}_t)\|_2 \cos(\nu') + \frac{\beta}{2}\epsilon^2 \\
&\leq \mathcal{L}(\boldsymbol{\theta}_t) - \epsilon\|\nabla_{\boldsymbol{\theta}_f}\mathcal{L}(\boldsymbol{\theta}_t)\|_2\sqrt{1 - (c_t\kappa)^2} + \frac{\beta}{2}\epsilon^2 \\
&= \mathcal{L}(\boldsymbol{\theta}_t) - \frac{\beta\epsilon}{2}\left(\frac{2\|\nabla_{\boldsymbol{\theta}_f}\mathcal{L}(\boldsymbol{\theta}_t)\|_2\sqrt{1 - (c_t\kappa)^2}}{\beta} - \epsilon\right) \\
&< \mathcal{L}(\boldsymbol{\theta}_t) .
\end{aligned}$$

where the last inequality follows by the construction of $\epsilon$ in Condition (A). Thus, we have $\mathcal{L}(\boldsymbol{\theta}') < \mathcal{L}(\boldsymbol{\theta}_t)$, which implies that the constructed $\boldsymbol{\theta}'$ is as described in Condition 1(b.2). This finishes the proof for the first part of the theorem's statement.

So far, we have assumed that $\|\nabla_{\boldsymbol{\theta}_f}\bar{G}_t\|_2 \neq 0$ in order to prove Condition 1(b.2). If, instead, we assume that $\|\nabla_{\boldsymbol{\theta}_g}\bar{G}_t\|_2 \neq 0$, then the whole proof for Condition 1(b.2) is done symmetrically: we start by taking $\boldsymbol{\theta}' = [\boldsymbol{\theta}_{f,t}^\top \ \boldsymbol{\theta}'^\top_g]^\top$ and then characterize $\boldsymbol{\theta}'_g \in \mathbb{R}^{p_g}$.

---

[1]We say vector $\mathbf{a}$ is in the positive half-space of vector $\mathbf{b}$ if $\mathbf{a}^\top\mathbf{b} \geq 0$.

We now proceed to prove the second part of the proof. For any $\boldsymbol{\theta}' \in B_\kappa^t$, by the second order Taylor expansion of the DON loss with respect to iterate $\boldsymbol{\theta}_t$, we have

$$\mathcal{L}(\boldsymbol{\theta}') = \mathcal{L}(\boldsymbol{\theta}_t) + \langle \boldsymbol{\theta}' - \boldsymbol{\theta}_t, \nabla_{\boldsymbol{\theta}} \mathcal{L}(\boldsymbol{\theta}_t) \rangle + \frac{1}{2}(\boldsymbol{\theta}' - \boldsymbol{\theta}_t)^\top \frac{\partial^2 \mathcal{L}(\tilde{\boldsymbol{\theta}})}{\partial \boldsymbol{\theta}^2}(\boldsymbol{\theta}' - \boldsymbol{\theta}_t) \,,$$

where $\tilde{\boldsymbol{\theta}} = \xi \boldsymbol{\theta}' + (1 - \xi)\boldsymbol{\theta}_t$ for some $\xi \in [0, 1]$. To establish $\alpha_t$-RSC of the loss with $\alpha_t$ as in (13), it suffices to focus on the quadratic form of the Hessian and show

$$(\boldsymbol{\theta}' - \boldsymbol{\theta}_t)^\top \frac{\partial^2 \mathcal{L}(\tilde{\boldsymbol{\theta}})}{\partial \boldsymbol{\theta}^2}(\boldsymbol{\theta} - \boldsymbol{\theta}_t) \geq \alpha_t \|\boldsymbol{\theta}' - \boldsymbol{\theta}_t\|_2^2 \,. \tag{31}$$

Note that the Hessian, by chain rule, is given by

$$\boldsymbol{H}(\tilde{\boldsymbol{\theta}}) := \frac{\partial^2 \mathcal{L}(\tilde{\boldsymbol{\theta}})}{\partial \boldsymbol{\theta}^2} = \frac{1}{n} \sum_{i=1}^n \frac{1}{q_i} \sum_{j=1}^{q_i} \left( \ell_{i,j}'' \nabla_{\boldsymbol{\theta}} G_{\tilde{\boldsymbol{\theta}}}(u^{(i)})(\boldsymbol{y}_j^{(i)}) \nabla_{\boldsymbol{\theta}} G_{\tilde{\boldsymbol{\theta}}}(u^{(i)})(\boldsymbol{y}_j^{(i)})^\top + \ell_{i,j}' \nabla^2 G_{\tilde{\boldsymbol{\theta}}}(u^{(i)})(\boldsymbol{y}_j^{(i)}) \right) \,.$$

where $\ell_{i,j} = (G_{\tilde{\boldsymbol{\theta}}}(u^{(i)})(\boldsymbol{y}_j^{(i)}) - G^\dagger(u^{(i)})(\boldsymbol{y}_j^{(i)}))^2$. Given the $2 \times 2$ block structure of the Hessian as in (23), denoting $\delta\boldsymbol{\theta} := \boldsymbol{\theta}' - \boldsymbol{\theta}_t$ for compactness, the quadratic form on the Hessian is given by

$$\delta\boldsymbol{\theta}^\top \boldsymbol{H}(\tilde{\boldsymbol{\theta}}) \delta\boldsymbol{\theta} = \underbrace{\delta\boldsymbol{\theta}_f^\top H_{ff}(\tilde{\boldsymbol{\theta}}) \delta\boldsymbol{\theta}_f}_{T_1} + \underbrace{2\delta\boldsymbol{\theta}_f^\top H_{fg}(\tilde{\boldsymbol{\theta}}) \delta\boldsymbol{\theta}_g}_{T_2} + \underbrace{\delta\boldsymbol{\theta}_g^\top H_{gg}(\tilde{\boldsymbol{\theta}}) \delta\boldsymbol{\theta}_g}_{T_3} \,. \tag{32}$$

Focusing on $T_1$ and using the exact form of $H_{ff}(\tilde{\boldsymbol{\theta}})$ as in (26), we have

$$T_1 = \frac{1}{n} \sum_{i=1}^n \frac{1}{q_i} \sum_{j=1}^{q_i} \ell_{i,j}'' \left\langle \delta\boldsymbol{\theta}_f, \sum_{k=1}^K g_{k,j}^{(i)}(\tilde{\boldsymbol{\theta}}_g) \nabla_{\boldsymbol{\theta}_f} f_k^{(i)}(\tilde{\boldsymbol{\theta}}_f) \right\rangle^2 + \frac{1}{n} \sum_{i=1}^n \frac{1}{q_i} \sum_{j=1}^{q_i} \ell_{ij}' \sum_{k=1}^K g_{k,j}^{(i)}(\tilde{\boldsymbol{\theta}}_g) \delta\boldsymbol{\theta}_f^\top \nabla_{\boldsymbol{\theta}_f}^2 f_k^{(i)}(\tilde{\boldsymbol{\theta}}_f) \delta\boldsymbol{\theta}_f$$

$$\overset{(a)}{\geq} \frac{2}{n} \sum_{i=1}^n \frac{1}{q_i} \sum_{j=1}^{q_i} \left\langle \delta\boldsymbol{\theta}_f, \nabla_{\boldsymbol{\theta}_f} G_{\tilde{\boldsymbol{\theta}}}(u^{(i)})(\boldsymbol{y}_j^{(i)}) \right\rangle^2 - \frac{(2K\lambda_1\lambda_2 + \tilde{c})\lambda_2 c^{(f)}}{\sqrt{m_f}} \|\delta\boldsymbol{\theta}_f\|_2^2 \,,$$

where (a) follows from $\ell_{ij}'' = 2$ and the different bounds in Lemma 3 since $\tilde{\boldsymbol{\theta}} \in B_{\rho,\rho_1}^{\text{Euc}}(\boldsymbol{\theta}_0)$, so that $|\ell_{ij}'| \leq 2K\lambda_1\lambda_2 + \tilde{c}$ with $\tilde{c} = \max_{i \in [n], j \in [q_i]} |G^\dagger(u^{(i)})(\boldsymbol{y}_j^{(i)})|$. Similarly, for $T_3$ we get

$$T_3 \geq \frac{2}{n} \sum_{i=1}^n \frac{1}{q_i} \sum_{j=1}^{q_i} \left\langle \delta\boldsymbol{\theta}_g, \nabla_{\boldsymbol{\theta}_g} G_{\tilde{\boldsymbol{\theta}}}(u^{(i)})(\boldsymbol{y}_j^{(i)}) \right\rangle^2 - \frac{(2K\lambda_1\lambda_2 + \tilde{c})\lambda_1 c^{(g)}}{\sqrt{m_g}} \|\delta\boldsymbol{\theta}_g\|_2^2 \,.$$

Then,

$$T_1 + T_3 \overset{(a)}{\geq} \frac{2}{n} \sum_{i=1}^n \frac{1}{q_i} \sum_{j=1}^{q_i} \left( \left\langle \delta\boldsymbol{\theta}_g, \nabla_{\boldsymbol{\theta}_g} G_{\tilde{\boldsymbol{\theta}}}(u^{(i)})(\boldsymbol{y}_j^{(i)}) \right\rangle^2 + \left\langle \delta\boldsymbol{\theta}_f, \nabla_{\boldsymbol{\theta}_f} G_{\tilde{\boldsymbol{\theta}}}(u^{(i)})(\boldsymbol{y}_j^{(i)}) \right\rangle^2 \right)$$

$$- (2K\lambda_1\lambda_2 + \tilde{c}) \left( \frac{\lambda_1 c^{(g)}}{\sqrt{m_g}} + \frac{\lambda_2 c^{(f)}}{\sqrt{m_f}} \right) \|\delta\boldsymbol{\theta}\|_2^2$$

$$= \frac{2}{n} \sum_{i=1}^n \frac{1}{q_i} \sum_{j=1}^{q_i} \left( \left\langle \delta\boldsymbol{\theta}_g, \nabla_{\boldsymbol{\theta}_g} G_{\tilde{\boldsymbol{\theta}}}(u^{(i)})(\boldsymbol{y}_j^{(i)}) \right\rangle + \left\langle \delta\boldsymbol{\theta}_f, \nabla_{\boldsymbol{\theta}_f} G_{\tilde{\boldsymbol{\theta}}}(u^{(i)})(\boldsymbol{y}_j^{(i)}) \right\rangle \right)^2$$

$$- \frac{4}{n} \sum_{i=1}^n \frac{1}{q_i} \sum_{j=1}^{q_i} \left\langle \delta\boldsymbol{\theta}_g, \nabla_{\boldsymbol{\theta}_g} G_{\tilde{\boldsymbol{\theta}}}(u^{(i)})(\boldsymbol{y}_j^{(i)}) \right\rangle \left\langle \delta\boldsymbol{\theta}_f, \nabla_{\boldsymbol{\theta}_f} G_{\tilde{\boldsymbol{\theta}}}(u^{(i)})(\boldsymbol{y}_j^{(i)}) \right\rangle$$

$$- (2K\lambda_1\lambda_2 + \tilde{c}) \left( \frac{\lambda_1 c^{(g)}}{\sqrt{m_g}} + \frac{\lambda_2 c^{(f)}}{\sqrt{m_f}} \right) \|\delta\boldsymbol{\theta}\|_2^2 \,,$$

where (a) follows from $\|\boldsymbol{\theta}_f\|_2, \|\boldsymbol{\theta}_g\|_2 \leq \|\boldsymbol{\theta}\|_2$.

Focusing on $T_2$ and using the exact form as in (26), we have

$$T_2 = 2\delta\boldsymbol{\theta}_f^\top \left( \frac{1}{n}\sum_{i=1}^{n}\frac{1}{q_i}\sum_{j=1}^{q_i}\ell'_{ij}\sum_{k=1}^{K}\nabla_{\boldsymbol{\theta}_f}f_k^{(i)}(\tilde{\boldsymbol{\theta}}_f)\nabla_{\boldsymbol{\theta}_g}g_{k,j}^{(i)}(\tilde{\boldsymbol{\theta}}_g)^\top \right)\delta\boldsymbol{\theta}_g$$

$$+ 2\delta\boldsymbol{\theta}_f^\top \left( \frac{1}{n}\sum_{i=1}^{n}\frac{1}{q_i}\sum_{j=1}^{q_i}\ell''_{i,j}\left(\sum_{k=1}^{K}g_{k,j}^{(i)}\nabla_{\boldsymbol{\theta}_f}f_k^{(i)}(\tilde{\boldsymbol{\theta}}_f)\right)\left(\sum_{k'=1}^{K}f_{k'}^{(i)}\nabla_{\boldsymbol{\theta}_g}g_{k',j}^{(i)}(\tilde{\boldsymbol{\theta}}_g)^\top\right)\right)\delta\boldsymbol{\theta}_g$$

$$\overset{(a)}{=} 2\delta\boldsymbol{\theta}_f^\top \underbrace{\left( \frac{1}{n}\sum_{i=1}^{n}\frac{1}{q_i}\sum_{j=1}^{q_i}\ell'_{ij}\sum_{k=1}^{K}\nabla_{\boldsymbol{\theta}_f}f_k^{(i)}(\tilde{\boldsymbol{\theta}}_f)\nabla_{\boldsymbol{\theta}_g}g_{k,j}^{(i)}(\tilde{\boldsymbol{\theta}}_g)^\top \right)\delta\boldsymbol{\theta}_g}_{I_1}$$

$$+ \left( \frac{4}{n}\sum_{i=1}^{n}\frac{1}{q_i}\sum_{j=1}^{q_i}\left\langle\delta\boldsymbol{\theta}_g,\nabla_{\boldsymbol{\theta}_g}G_{\boldsymbol{\theta}_t}(u^{(i)})(\boldsymbol{y}_j^{(i)})\right\rangle\left\langle\delta\boldsymbol{\theta}_f,\nabla_{\boldsymbol{\theta}_f}G_{\boldsymbol{\theta}_t}(u^{(i)})(\boldsymbol{y}_j^{(i)})\right\rangle\right),$$

where (a) follows from $\ell''_{i,j} = 2$.

For $I_1$ our goal is to first transfer the dependence of the gradient terms on $\tilde{\boldsymbol{\theta}}$ to $\boldsymbol{\theta}_t$, so that we can use properties of the restricted set $Q_\kappa^t$ which is based on $\boldsymbol{\theta}_t$ to simplify the analysis. Towards that end, note that

$$\frac{1}{2}I_1 = \delta\boldsymbol{\theta}_f^\top \left( \frac{1}{n}\sum_{i=1}^{n}\frac{1}{q_i}\sum_{j=1}^{q_i}\ell'_{ij}\sum_{k=1}^{K}\nabla_{\boldsymbol{\theta}_f}f_k^{(i)}(\boldsymbol{\theta}_{t,f})\nabla_{\boldsymbol{\theta}_g}g_{k,j}^{(i)}(\boldsymbol{\theta}_{t,g})^\top \right)\delta\boldsymbol{\theta}_g$$

$$+ \delta\boldsymbol{\theta}_f^\top \left( \frac{1}{n}\sum_{i=1}^{n}\frac{1}{q_i}\sum_{j=1}^{q_i}\ell'_{ij}\sum_{k=1}^{K}\left(\nabla_{\boldsymbol{\theta}_f}f_k^{(i)}(\tilde{\boldsymbol{\theta}}_f) - \nabla_{\boldsymbol{\theta}_f}f_k^{(i)}(\boldsymbol{\theta}_{t,f})\right)\nabla_{\boldsymbol{\theta}_g}g_{k,j}^{(i)}(\tilde{\boldsymbol{\theta}}_g)^\top \right)\delta\boldsymbol{\theta}_g$$

$$+ \delta\boldsymbol{\theta}_f^\top \left( \frac{1}{n}\sum_{i=1}^{n}\frac{1}{q_i}\sum_{j=1}^{q_i}\ell'_{ij}\sum_{k=1}^{K}\nabla_{\boldsymbol{\theta}_f}f_k^{(i)}(\boldsymbol{\theta}_{t,f})\left(\nabla_{\boldsymbol{\theta}_g}g_{k,j}^{(i)}(\tilde{\boldsymbol{\theta}}_g) - \nabla_{\boldsymbol{\theta}_g}g_{k,j}^{(i)}(\boldsymbol{\theta}_{t,g})\right)^\top \right)\delta\boldsymbol{\theta}_g$$

$$\overset{(a)}{\geq} -\frac{(2K\lambda_1\lambda_2+\tilde{c})}{n}\sum_{i=1}^{n}\frac{1}{q_i}\sum_{j=1}^{q_i}\left\|\nabla_{\boldsymbol{\theta}_f}f_k^{(i)}(\tilde{\boldsymbol{\theta}}_f)-\nabla_{\boldsymbol{\theta}_f}f_k^{(i)}(\boldsymbol{\theta}_{t,f})\right\|_2\left\|\nabla_{\boldsymbol{\theta}_g}g_{k,j}^{(i)}(\tilde{\boldsymbol{\theta}}_g)\right\|_2\|\delta\boldsymbol{\theta}_f\|_2\|\delta\boldsymbol{\theta}_g\|_2$$

$$-\frac{(2K\lambda_1\lambda_2+\tilde{c})}{n}\sum_{i=1}^{n}\frac{1}{q_i}\sum_{j=1}^{q_i}\sum_{k=1}^{K}\left\|\nabla_{\boldsymbol{\theta}_f}f_k^{(i)}(\boldsymbol{\theta}_{t,f})\right\|_2\left\|\nabla_{\boldsymbol{\theta}_g}g_{k,j}^{(i)}(\tilde{\boldsymbol{\theta}}_g)-\nabla_{\boldsymbol{\theta}_g}g_{k,j}^{(i)}(\boldsymbol{\theta}_{t,g})\right\|_2\|\delta\boldsymbol{\theta}_f\|_2\|\delta\boldsymbol{\theta}_g\|_2$$

$$\overset{(b)}{=} -\frac{(2K\lambda_1\lambda_2+\tilde{c})}{n}\sum_{i=1}^{n}\frac{1}{q_i}\sum_{j=1}^{q_i}\left\|\nabla_{\bar{\boldsymbol{\theta}}_f}^2 f_k^{(i)}(\bar{\boldsymbol{\theta}}_f)\right\|_2\|\tilde{\boldsymbol{\theta}}_f-\boldsymbol{\theta}_{t,f}\|_2\left\|\nabla_{\boldsymbol{\theta}_g}g_{k,j}^{(i)}(\tilde{\boldsymbol{\theta}}_g)\right\|_2\|\delta\boldsymbol{\theta}_f\|_2\|\delta\boldsymbol{\theta}_g\|_2$$

$$-\frac{(2K\lambda_1\lambda_2+\tilde{c})}{n}\sum_{i=1}^{n}\frac{1}{q_i}\sum_{j=1}^{q_i}\sum_{k=1}^{K}\left\|\nabla_{\boldsymbol{\theta}_f}f_k^{(i)}(\boldsymbol{\theta}_{t,f})\right\|_2\left\|\nabla_{\bar{\boldsymbol{\theta}}_g}^2 g_{k,j}^{(i)}(\bar{\boldsymbol{\theta}}_g)\right\|_2\|\tilde{\boldsymbol{\theta}}_g-\boldsymbol{\theta}_{t,g}\|_2\|\delta\boldsymbol{\theta}_f\|_2\|\delta\boldsymbol{\theta}_g\|_2$$

$$\overset{(c)}{\geq} -(2K\lambda_1\lambda_2+\tilde{c})\left(\frac{c^{(f)}\varrho^{(g)}}{\sqrt{m_f}}\right)\|\delta\boldsymbol{\theta}_f\|_2^2\|\delta\boldsymbol{\theta}_g\|_2 - (2K\lambda_1\lambda_2+\tilde{c})\left(\frac{c^{(g)}\varrho^{(f)}}{\sqrt{m_g}}\right)\|\delta\boldsymbol{\theta}_f\|_2\|\delta\boldsymbol{\theta}_g\|_2^2$$

$$\overset{(d)}{\geq} -(2K\lambda_1\lambda_2+\tilde{c})\left(\frac{c^{(g)}\varrho^{(f)}}{\sqrt{m_f}}+\frac{c^{(g)}\varrho^{(f)}}{\sqrt{m_f}}\right)\|\delta\boldsymbol{\theta}\|_2^3$$

$$\geq -(2K\lambda_1\lambda_2+\tilde{c})\rho_2\left(\frac{c^{(g)}\varrho^{(f)}}{\sqrt{m_f}}+\frac{c^{(g)}\varrho^{(f)}}{\sqrt{m_f}}\right)\|\delta\boldsymbol{\theta}\|_2^2,$$

where (a) follows from the definition of $Q_\kappa^t$ set (Definition 12) since $\boldsymbol{\theta}' \in B_\kappa^t \subset Q_{\kappa,\cdot}^t$ (b) follows from the generalized mean value theorem with $\bar{\boldsymbol{\theta}}_f = \xi_1\tilde{\boldsymbol{\theta}}_f + (1-\xi_1)\boldsymbol{\theta}_{t,f}$ for some $\xi_1 \in [0,1]$ and $\bar{\boldsymbol{\theta}}_g = \xi_2\tilde{\boldsymbol{\theta}}_g + (1-\xi_2)\boldsymbol{\theta}_{t,g}$ for some $\xi_2 \in [0,1]$; (c) follows from the results in Lemma 3 since $[\bar{\boldsymbol{\theta}}_f^\top\ \bar{\boldsymbol{\theta}}_g^\top]^\top \in B_{\rho,\rho_1}^{\text{Euc}}(\boldsymbol{\theta}_0)$, and the fact that $\|\tilde{\boldsymbol{\theta}}_f - \boldsymbol{\theta}_{f,t}\|_2 \leq \|\delta\boldsymbol{\theta}_f\|_2$ and $\|\tilde{\boldsymbol{\theta}}_g - \boldsymbol{\theta}_{g,t}\|_2 \leq \|\delta\boldsymbol{\theta}_g\|_2$; and (d) follows from $\|\delta\boldsymbol{\theta}_f\|_2, \|\delta\boldsymbol{\theta}_g\|_2 \leq \|\delta\boldsymbol{\theta}\|_2$.

Replacing $I_1$ back in $T_2$ and then combining the bounds on $T_1 + T_3$ and $T_2$, we have

$$
\begin{aligned}
\delta\boldsymbol{\theta}^\top \boldsymbol{H}(\tilde{\boldsymbol{\theta}})\delta\boldsymbol{\theta} &\geq \frac{2}{n}\sum_{i=1}^{n}\frac{1}{q_i}\sum_{j=1}^{q_i}\left(\left\langle \delta\boldsymbol{\theta}_g, \nabla_{\boldsymbol{\theta}_g}G_{\tilde{\boldsymbol{\theta}}}(u^{(i)})(\boldsymbol{y}_j^{(i)})\right\rangle + \left\langle \delta\boldsymbol{\theta}_f, \nabla_{\boldsymbol{\theta}_f}G_{\tilde{\boldsymbol{\theta}}}(u^{(i)})(\boldsymbol{y}_j^{(i)})\right\rangle\right)^2 \\
&\quad - (2K\lambda_1\lambda_2 + \tilde{c})\left(\frac{\lambda_1 c^{(g)}}{\sqrt{m_g}} + \frac{\lambda_2 c^{(f)}}{\sqrt{m_f}}\right)\|\delta\boldsymbol{\theta}\|_2^2 \\
&\quad - 2(2K\lambda_1\lambda_2 + \tilde{c})\rho_2\left(\frac{c^{(g)}\varrho^{(f)}}{\sqrt{m_f}} + \frac{c^{(g)}\varrho^{(f)}}{\sqrt{m_f}}\right)\|\delta\boldsymbol{\theta}\|_2^2 \\
&= \frac{2}{n}\sum_{i=1}^{n}\frac{1}{q_i}\sum_{j=1}^{q_i}\left\langle \delta\boldsymbol{\theta}, \nabla_{\boldsymbol{\theta}}G_{\tilde{\boldsymbol{\theta}}}(u^{(i)})(\boldsymbol{y}_j^{(i)})\right\rangle^2 \\
&\quad - (2K\lambda_1\lambda_2 + \tilde{c})\left(\frac{\lambda_1 c^{(g)}}{\sqrt{m_g}} + \frac{\lambda_2 c^{(f)}}{\sqrt{m_f}}\right)\|\delta\boldsymbol{\theta}\|_2^2 \\
&\quad - 2(2K\lambda_1\lambda_2 + \tilde{c})\rho_2\left(\frac{c^{(g)}\varrho^{(f)}}{\sqrt{m_f}} + \frac{c^{(g)}\varrho^{(f)}}{\sqrt{m_f}}\right)\|\delta\boldsymbol{\theta}\|_2^2 \\
&= \underbrace{\frac{2}{n}\sum_{i=1}^{n}\frac{1}{q_i}\sum_{j=1}^{q_i}\left(\left\langle \delta\boldsymbol{\theta}, \nabla_{\boldsymbol{\theta}}G_{\boldsymbol{\theta}_t}(u^{(i)})(\boldsymbol{y}_j^{(i)})\right\rangle + \left(\left\langle \delta\boldsymbol{\theta}, \nabla_{\boldsymbol{\theta}}G_{\tilde{\boldsymbol{\theta}}}(u^{(i)})(\boldsymbol{y}_j^{(i)})\right\rangle - \left\langle \delta\boldsymbol{\theta}, \nabla_{\boldsymbol{\theta}}G_{\boldsymbol{\theta}_t}(u^{(i)})(\boldsymbol{y}_j^{(i)})\right\rangle\right)\right)^2}_{I_2} \\
&\quad - (2K\lambda_1\lambda_2 + \tilde{c})\left(\frac{\lambda_1 c^{(g)}}{\sqrt{m_g}} + \frac{\lambda_2 c^{(f)}}{\sqrt{m_f}}\right)\|\delta\boldsymbol{\theta}\|_2^2 \\
&\quad - 2(2K\lambda_1\lambda_2 + \tilde{c})\rho_2\left(\frac{c^{(g)}\varrho^{(f)}}{\sqrt{m_f}} + \frac{c^{(g)}\varrho^{(f)}}{\sqrt{m_f}}\right)\|\delta\boldsymbol{\theta}\|_2^2 .
\end{aligned}
\tag{33}
$$

Then,

$$
\begin{aligned}
I_2 &= \frac{2}{n}\sum_{i=1}^{n}\frac{1}{q_i}\sum_{j=1}^{q_i}\left\langle \delta\boldsymbol{\theta}, \nabla_{\boldsymbol{\theta}}G_{\boldsymbol{\theta}_t}(u^{(i)})(\boldsymbol{y}_j^{(i)})\right\rangle^2 + \sum_{i=1}^{n}\frac{2}{q_i}\sum_{j=1}^{q_i}\left\langle \delta\boldsymbol{\theta}, \nabla_{\boldsymbol{\theta}}G_{\tilde{\boldsymbol{\theta}}}(u^{(i)})(\boldsymbol{y}_j^{(i)}) - \nabla_{\boldsymbol{\theta}}G_{\boldsymbol{\theta}_t}(u^{(i)})(\boldsymbol{y}_j^{(i)})\right\rangle^2 \\
&\quad + \frac{4}{n}\sum_{i=1}^{n}\frac{1}{q_i}\sum_{j=1}^{q_i}\left\langle \delta\boldsymbol{\theta}_f, \nabla_{\boldsymbol{\theta}}G_{\boldsymbol{\theta}_t}(u^{(i)})(\boldsymbol{y}_j^{(i)})\right\rangle\left\langle \delta\boldsymbol{\theta}, \nabla_{\boldsymbol{\theta}}G_{\tilde{\boldsymbol{\theta}}}(u^{(i)})(\boldsymbol{y}_j^{(i)}) - \nabla_{\boldsymbol{\theta}}G_{\boldsymbol{\theta}_t}(u^{(i)})(\boldsymbol{y}_j^{(i)})\right\rangle \\
&\overset{(a)}{=} \frac{2}{n}\sum_{i=1}^{n}\frac{1}{q_i}\sum_{j=1}^{q_i}\left\langle \delta\boldsymbol{\theta}, \nabla_{\boldsymbol{\theta}}G_{\boldsymbol{\theta}_t}(u^{(i)})(\boldsymbol{y}_j^{(i)})\right\rangle^2 + \sum_{i=1}^{n}\frac{2}{q_i}\sum_{j=1}^{q_i}\left\langle \delta\boldsymbol{\theta}, \nabla_{\boldsymbol{\theta}}G_{\tilde{\boldsymbol{\theta}}}(u^{(i)})(\boldsymbol{y}_j^{(i)}) - \nabla_{\boldsymbol{\theta}}G_{\boldsymbol{\theta}_t}(u^{(i)})(\boldsymbol{y}_j^{(i)})\right\rangle^2 \\
&\quad + \frac{4}{n}\sum_{i=1}^{n}\frac{1}{q_i}\sum_{j=1}^{q_i}\left\langle \delta\boldsymbol{\theta}, \nabla_{\boldsymbol{\theta}}G_{\boldsymbol{\theta}_t}(u^{(i)})(\boldsymbol{y}_j^{(i)})\right\rangle\left((\delta\boldsymbol{\theta})^\top\nabla_{\boldsymbol{\theta}}^2 G_{\tilde{\tilde{\boldsymbol{\theta}}}}(u^{(i)})(\boldsymbol{y}_j^{(i)})(\tilde{\boldsymbol{\theta}} - \boldsymbol{\theta}_t)\right) \\
&\geq \frac{2}{n}\sum_{i=1}^{n}\frac{1}{q_i}\sum_{j=1}^{q_i}\left\langle \delta\boldsymbol{\theta}, \nabla_{\boldsymbol{\theta}}G_{\boldsymbol{\theta}_t}(u^{(i)})(\boldsymbol{y}_j^{(i)})\right\rangle^2 \\
&\quad + \frac{4}{n}\sum_{i=1}^{n}\frac{1}{q_i}\sum_{j=1}^{q_i}\left\langle \delta\boldsymbol{\theta}, \nabla_{\boldsymbol{\theta}}G_{\boldsymbol{\theta}_t}(u^{(i)})(\boldsymbol{y}_j^{(i)})\right\rangle\left((\delta\boldsymbol{\theta})^\top\nabla_{\boldsymbol{\theta}}^2 G_{\tilde{\tilde{\boldsymbol{\theta}}}}(u^{(i)})(\boldsymbol{y}_j^{(i)})(\tilde{\boldsymbol{\theta}} - \boldsymbol{\theta}_t)\right) \\
&\overset{(b)}{\geq} \frac{2}{n}\sum_{i=1}^{n}\frac{1}{q_i}\sum_{j=1}^{q_i}\left\langle \delta\boldsymbol{\theta}, \nabla_{\boldsymbol{\theta}}G_{\boldsymbol{\theta}_t}(u^{(i)})(\boldsymbol{y}_j^{(i)})\right\rangle^2 \\
&\quad - 4\xi_3 K(\lambda_2\varrho^{(f)} + \lambda_1\varrho^{(g)})\|\delta\boldsymbol{\theta}\|_2\frac{1}{n}\sum_{i=1}^{n}\frac{1}{q_i}\sum_{j=1}^{q_i}\left|(\delta\boldsymbol{\theta})^\top\nabla_{\boldsymbol{\theta}}^2 G_{\tilde{\tilde{\boldsymbol{\theta}}}}(u^{(i)})(\boldsymbol{y}_j^{(i)})(\delta\boldsymbol{\theta})\right|
\end{aligned}
\tag{34}
$$

where (a) follows from the generalized mean value theorem and has $\tilde{\tilde{\boldsymbol{\theta}}} \in \xi_3\tilde{\boldsymbol{\theta}} + (1 - \xi_3)\boldsymbol{\theta}_t$ for some $\xi_3 \in [0, 1]$; and (b)

follows from the fact that $\tilde{\boldsymbol{\theta}} - \boldsymbol{\theta}_t = \xi_3(\theta' - \boldsymbol{\theta}')$ and

$$
\begin{aligned}
\left\| \nabla_{\boldsymbol{\theta}} G_{\boldsymbol{\theta}_t}(u^{(i)})(\boldsymbol{y}_j^{(i)}) \right\|_2 &\leq \left\| \nabla_{\boldsymbol{\theta}_f} G_{\boldsymbol{\theta}_t}(u^{(i)})(\boldsymbol{y}_j^{(i)}) \right\|_2 + \left\| \nabla_{\boldsymbol{\theta}_g} G_{\boldsymbol{\theta}_t}(u^{(i)})(\boldsymbol{y}_j^{(i)}) \right\|_2 \\
&\leq \left\| \sum_{k=1}^{K} g_{k,j}^{(i)}(\boldsymbol{\theta}_t) \nabla_{\boldsymbol{\theta}_f} f_k^{(i)}(\boldsymbol{\theta}_t) \right\|_2 + \left\| \sum_{k=1}^{K} \nabla_{\boldsymbol{\theta}_g} g_{k,j}^{(i)}(\boldsymbol{\theta}_t)(f_k^{(i)}(\boldsymbol{\theta}_t)) \right\|_2 \\
&\leq \sum_{k=1}^{K} |g_{k,j}^{(i)}(\boldsymbol{\theta}_t)| \| \nabla_{\boldsymbol{\theta}_f} f_k^{(i)}(\boldsymbol{\theta}_t) \|_2 + \sum_{k=1}^{K} |f_k^{(i)}(\boldsymbol{\theta}_t)| \| \nabla_{\boldsymbol{\theta}_g} g_{k,j}^{(i)}(\boldsymbol{\theta}_t) \|_2 \\
&\leq K \lambda_2 \varrho^{(f)} + K \lambda_1 \varrho^{(g)} \, ,
\end{aligned}
$$

where the last inequality follows from Lemma 3.

Now, we have that

$$
\begin{aligned}
(\delta\boldsymbol{\theta})^\top \nabla_{\boldsymbol{\theta}}^2 G_{\tilde{\boldsymbol{\theta}}}(u^{(i)})(\boldsymbol{y}_j^{(i)})(\delta\boldsymbol{\theta}) &= \sum_{k=1}^{K} (\delta\boldsymbol{\theta}_f)^\top (g_{k,j}^{(i)}(\tilde{\tilde{\boldsymbol{\theta}}}_g) \nabla_{\boldsymbol{\theta}_f}^2 f_k^{(i)}(\tilde{\tilde{\boldsymbol{\theta}}}_f)(\delta\boldsymbol{\theta}_f) + \sum_{k=1}^{K} (\delta\boldsymbol{\theta}_g)^\top (f_k^{(i)}(\tilde{\tilde{\boldsymbol{\theta}}}_f) \nabla_{\boldsymbol{\theta}_g}^2 g_{k,j}^{(i)}(\tilde{\tilde{\boldsymbol{\theta}}}_g)(\delta\boldsymbol{\theta}_g) \\
&\quad + 2 \sum_{k=1}^{K} (\delta\boldsymbol{\theta}_g)^\top (\nabla_{\boldsymbol{\theta}_g} g_{k,j}^{(i)}(\tilde{\tilde{\boldsymbol{\theta}}}_g)(\nabla_{\boldsymbol{\theta}_f} f_k^{(i)}(\tilde{\tilde{\boldsymbol{\theta}}}_f))^\top (\delta\boldsymbol{\theta}_f) \\
&\overset{(a)}{\leq} K \left( \frac{\lambda_2 c^{(f)}}{\sqrt{m_f}} + \frac{\lambda_1 c^{(g)}}{\sqrt{m_g}} \right) \|\delta\boldsymbol{\theta}\|_2^2 + \underbrace{2 \sum_{k=1}^{K} (\delta\boldsymbol{\theta}_g)^\top (\nabla_{\boldsymbol{\theta}_g} g_{k,j}^{(i)}(\tilde{\tilde{\boldsymbol{\theta}}}_g)(\nabla_{\boldsymbol{\theta}_f} f_k^{(i)}(\tilde{\tilde{\boldsymbol{\theta}}}_f))^\top (\delta\boldsymbol{\theta}_f)}_{I_3}
\end{aligned}
$$
(35)

where (a) follows from Lemma 3 since $\tilde{\tilde{\boldsymbol{\theta}}} \in B_{\rho,\rho_1}^{\mathrm{Euc}}(\boldsymbol{\theta}_0)$.

Now, for $I_3$,

$$\frac{1}{2}I_3 = \delta\boldsymbol{\theta}_f^\top \left(\sum_{k=1}^K \nabla_{\boldsymbol{\theta}_f} f_k^{(i)}(\boldsymbol{\theta}_{t,f})\nabla_{\boldsymbol{\theta}_g} g_{k,j}^{(i)}(\boldsymbol{\theta}_{t,g})^\top\right)\delta\boldsymbol{\theta}_g$$

$$+ \delta\boldsymbol{\theta}_f^\top \left(\sum_{k=1}^K \left(\nabla_{\boldsymbol{\theta}_f} f_k^{(i)}(\tilde{\tilde{\boldsymbol{\theta}}}_f) - \nabla_{\boldsymbol{\theta}_f} f_k^{(i)}(\boldsymbol{\theta}_{t,f})\right)\nabla_{\boldsymbol{\theta}_g} g_{k,j}^{(i)}(\tilde{\tilde{\boldsymbol{\theta}}}_g)^\top\right)\delta\boldsymbol{\theta}_g$$

$$+ \delta\boldsymbol{\theta}_f^\top \left(\sum_{k=1}^K \nabla_{\boldsymbol{\theta}_f} f_k^{(i)}(\boldsymbol{\theta}_{t,f})\left(\nabla_{\boldsymbol{\theta}_g} g_{k,j}^{(i)}(\tilde{\tilde{\boldsymbol{\theta}}}_g) - \nabla_{\boldsymbol{\theta}_g} g_{k,j}^{(i)}(\boldsymbol{\theta}_{t,g})\right)^\top\right)\delta\boldsymbol{\theta}_g$$

$$\leq \delta\boldsymbol{\theta}_f^\top \left(\sum_{k=1}^K \nabla_{\boldsymbol{\theta}_f} f_k^{(i)}(\boldsymbol{\theta}_{t,f})\nabla_{\boldsymbol{\theta}_g} g_{k,j}^{(i)}(\boldsymbol{\theta}_{t,g})^\top\right)\delta\boldsymbol{\theta}_g$$

$$+ \sum_{k=1}^K \left\|\nabla_{\boldsymbol{\theta}_f} f_k^{(i)}(\tilde{\tilde{\boldsymbol{\theta}}}_f) - \nabla_{\boldsymbol{\theta}_f} f_k^{(i)}(\boldsymbol{\theta}_{t,f})\right\|_2 \left\|\nabla_{\boldsymbol{\theta}_g} g_{k,j}^{(i)}(\tilde{\tilde{\boldsymbol{\theta}}}_g)\right\|_2 \|\delta\boldsymbol{\theta}_f\|_2\|\delta\boldsymbol{\theta}_g\|_2$$

$$+ \sum_{k=1}^K \left\|\nabla_{\boldsymbol{\theta}_f} f_k^{(i)}(\boldsymbol{\theta}_{t,f})\right\|_2 \left\|\nabla_{\boldsymbol{\theta}_g} g_{k,j}^{(i)}(\tilde{\tilde{\boldsymbol{\theta}}}_g) - \nabla_{\boldsymbol{\theta}_g} g_{k,j}^{(i)}(\boldsymbol{\theta}_{t,g})\right\|_2 \|\delta\boldsymbol{\theta}_f\|_2\|\delta\boldsymbol{\theta}_g\|_2$$

$$\overset{(a)}{\leq} \delta\boldsymbol{\theta}_f^\top \left(\sum_{k=1}^K \nabla_{\boldsymbol{\theta}_f} f_k^{(i)}(\boldsymbol{\theta}_{t,f})\nabla_{\boldsymbol{\theta}_g} g_{k,j}^{(i)}(\boldsymbol{\theta}_{t,g})^\top\right)\delta\boldsymbol{\theta}_g$$

$$+ \xi_3\xi K \left(\frac{c^{(f)}\varrho^{(g)}}{\sqrt{m_f}}\right)\|\delta\boldsymbol{\theta}_f\|_2^2\|\delta\boldsymbol{\theta}_g\|_2 + \xi_3\xi K\left(\frac{c^{(g)}\varrho^{(f)}}{\sqrt{m_g}}\right)\|\delta\boldsymbol{\theta}_f\|_2\|\delta\boldsymbol{\theta}_g\|_2^2$$

$$\overset{(b)}{\leq} \delta\boldsymbol{\theta}_f^\top \left(\sum_{k=1}^K \nabla_{\boldsymbol{\theta}_f} f_k^{(i)}(\boldsymbol{\theta}_{t,f})\nabla_{\boldsymbol{\theta}_g} g_{k,j}^{(i)}(\boldsymbol{\theta}_{t,g})^\top\right)\delta\boldsymbol{\theta}_g$$

$$+ K\left(\frac{c^{(g)}\varrho^{(f)}}{\sqrt{m_f}} + \frac{c^{(f)}\varrho^{(g)}}{\sqrt{m_g}}\right)\|\delta\boldsymbol{\theta}\|_2^3$$

$$\leq \delta\boldsymbol{\theta}_f^\top \left(\sum_{k=1}^K \nabla_{\boldsymbol{\theta}_f} f_k^{(i)}(\boldsymbol{\theta}_{t,f})\nabla_{\boldsymbol{\theta}_g} g_{k,j}^{(i)}(\boldsymbol{\theta}_{t,g})^\top\right)\delta\boldsymbol{\theta}_g$$

$$+ K\rho_2\left(\frac{c^{(g)}\varrho^{(f)}}{\sqrt{m_g}} + \frac{c^{(f)}\varrho^{(g)}}{\sqrt{m_f}}\right)\|\delta\boldsymbol{\theta}\|_2^2\,,$$

where (a) follows from the generalized mean value theorem, from $\|\tilde{\tilde{\boldsymbol{\theta}}}_f - \boldsymbol{\theta}_{t,f}\|_2 = \|\xi_3\tilde{\boldsymbol{\theta}}_f + (1-\xi_3)\boldsymbol{\theta}_{t,f} - \boldsymbol{\theta}_{t,f}\|_2 = \xi_3\xi\|\boldsymbol{\theta}' - \boldsymbol{\theta}_{t,f}\|_2 = \xi_3\xi\|\delta\boldsymbol{\theta}_f\|_2$, and from the results in Lemma 3 since $\tilde{\tilde{\boldsymbol{\theta}}} \in B_{\rho,\rho_1}^{\mathrm{Euc}}(\boldsymbol{\theta}_0)$; and (b) follows from $\|\boldsymbol{\theta}_f\|_2, \|\boldsymbol{\theta}_g\|_2 \leq \|\boldsymbol{\theta}\|_2$ and $\xi_3\xi \leq 1$.

Replacing the bound on $I_3$ back to (35), we obtain

$$(\delta\boldsymbol{\theta})^\top \nabla_{\tilde{\boldsymbol{\theta}}}^2 G_{\tilde{\boldsymbol{\theta}}}(u^{(i)})(\boldsymbol{y}_j^{(i)})(\delta\boldsymbol{\theta}) \leq \delta\boldsymbol{\theta}_f^\top \left(\sum_{k=1}^K \nabla_{\boldsymbol{\theta}_f} f_k^{(i)}(\boldsymbol{\theta}_{t,f})\nabla_{\boldsymbol{\theta}_g} g_{k,j}^{(i)}(\boldsymbol{\theta}_{t,g})^\top\right)\delta\boldsymbol{\theta}_g$$

$$+ K(1+\rho_2)\left(\frac{c^{(g)}(\lambda_1 + 2\varrho^{(f)})}{\sqrt{m_g}} + \frac{c^{(f)}(\lambda_2 + 2\varrho^{(g)})}{\sqrt{m_f}}\right)\|\delta\boldsymbol{\theta}\|_2^2 \qquad (36)$$

$$\leq K(1+\rho_2)\left(\frac{c^{(g)}(\lambda_1 + 2\varrho^{(f)})}{\sqrt{m_g}} + \frac{c^{(f)}(\lambda_2 + 2\varrho^{(g)})}{\sqrt{m_f}}\right)\|\delta\boldsymbol{\theta}\|_2^2\,,$$

where the last inequality follows from the fact that $\boldsymbol{\theta}' \in Q_\kappa^t$, using the properties of the restricted set $Q_\kappa^t$ in Definition 12.

Replacing (36) back to $I_2$ in (34), we obtain

$$
\begin{aligned}
I_2 &\geq \frac{2}{n}\sum_{i=1}^{n}\frac{1}{q_i}\sum_{j=1}^{q_i}\left\langle\delta\boldsymbol{\theta},\nabla_{\boldsymbol{\theta}}G_{\boldsymbol{\theta}_t}(u^{(i)})(\boldsymbol{y}_j^{(i)})\right\rangle^2\\
&\quad -4\xi_3 K(\lambda_2\varrho^{(f)}+\lambda_1\varrho^{(g)})\|\delta\boldsymbol{\theta}\|_2\times K(1+\rho_2)\left(\frac{c^{(g)}(\lambda_1+2\varrho^{(f)})}{\sqrt{m_g}}+\frac{c^{(f)}(\lambda_2+2\varrho^{(g)})}{\sqrt{m_f}}\right)\|\delta\boldsymbol{\theta}\|_2^2\\
&=\frac{2}{n}\sum_{i=1}^{n}\frac{1}{q_i}\sum_{j=1}^{q_i}\left\langle\delta\boldsymbol{\theta},\nabla_{\boldsymbol{\theta}}G_{\boldsymbol{\theta}_t}(u^{(i)})(\boldsymbol{y}_j^{(i)})\right\rangle^2\\
&\quad -4\xi_3 K^2(1+\rho_2)\rho_2(\lambda_2\varrho^{(f)}+\lambda_1\varrho^{(g)})\left(\frac{c^{(g)}(\lambda_1+2\varrho^{(f)})}{\sqrt{m_g}}+\frac{c^{(f)}(\lambda_2+2\varrho^{(g)})}{\sqrt{m_f}}\right)\|\delta\boldsymbol{\theta}\|_2^2\,.
\end{aligned}
\tag{37}
$$

Replacing this lower bound (37) back to the Hessian expression in (38),

$$
\begin{aligned}
\delta\boldsymbol{\theta}^{\top}\boldsymbol{H}(\tilde{\boldsymbol{\theta}})\delta\boldsymbol{\theta} &\geq \frac{2}{n}\sum_{i=1}^{n}\frac{1}{q_i}\sum_{j=1}^{q_i}\left\langle\delta\boldsymbol{\theta},\nabla_{\boldsymbol{\theta}}G_{\boldsymbol{\theta}_t}(u^{(i)})(\boldsymbol{y}_j^{(i)})\right\rangle^2\\
&\quad -4\xi_3 K^2(1+\rho_2)\rho_2(\lambda_2\varrho^{(f)}+\lambda_1\varrho^{(g)})\left(\frac{c^{(g)}(\lambda_1+2\varrho^{(f)})}{\sqrt{m_g}}+\frac{c^{(f)}(\lambda_2+2\varrho^{(g)})}{\sqrt{m_f}}\right)\|\delta\boldsymbol{\theta}\|_2^2\\
&\quad -(2K\lambda_1\lambda_2+\tilde{c})\left(\frac{\lambda_1 c^{(g)}}{\sqrt{m_g}}+\frac{\lambda_2 c^{(f)}}{\sqrt{m_f}}\right)\|\delta\boldsymbol{\theta}\|_2^2\\
&\quad -2(2K\lambda_1\lambda_2+\tilde{c})\rho_2\left(\frac{c^{(g)}\varrho^{(f)}}{\sqrt{m_f}}+\frac{c^{(g)}\varrho^{(f)}}{\sqrt{m_f}}\right)\|\delta\boldsymbol{\theta}\|_2^2\\
&\overset{(a)}{\geq}2\left\langle\delta\boldsymbol{\theta},\nabla_{\boldsymbol{\theta}}\bar{G}_{\boldsymbol{\theta}_t}\right\rangle^2-c_1 K^2\left(\frac{1}{\sqrt{m_f}}+\frac{1}{\sqrt{m_g}}\right)\|\delta\boldsymbol{\theta}\|_2^2\\
&\overset{(b)}{\geq}2\kappa^2\|\nabla_{\boldsymbol{\theta}}\bar{G}_{\boldsymbol{\theta}_t}\|_2^2\|\delta\boldsymbol{\theta}\|_2^2-c_1 K^2\left(\frac{1}{\sqrt{m_f}}+\frac{1}{\sqrt{m_g}}\right)\|\delta\boldsymbol{\theta}\|_2^2\\
&=\alpha_t\|\delta\boldsymbol{\theta}\|_2^2\,,
\end{aligned}
\tag{38}
$$

where (a) follows from Jensen's inequality with $\bar{G}_{\boldsymbol{\theta}}=\frac{1}{n}\sum_{i=1}^{n}\frac{1}{q_i}\sum_{j=1}^{q_i}G_{\boldsymbol{\theta}}(u^{(i)})(\boldsymbol{y}_j^{(i)})$; where (b) follows from the fact that $\boldsymbol{\theta}'\in Q_\kappa^t$ and using the properties of the restricted set $Q_\kappa^t$ in Definition 12; and where $\alpha_t=2\kappa^2\|\nabla_{\boldsymbol{\theta}}\bar{G}_{\boldsymbol{\theta}}\|_2^2-c_1 K^2\left(\frac{1}{m_f}+\frac{1}{m_g}\right)$. Note that adding all the constants from the second to the fourth line in (38) define the constant $c_1$, and so $c_1$ depends on $\sigma_1$, the depth $L$, and the radii $\rho$, $\rho_1$, and $\rho_2$ due to Lemma 3. As in the statement of Lemma 3, this dependence reduces to the depth and the radii and becomes polynomial whenever $\sigma_0\leq 1-\rho\max\{\frac{1}{\sqrt{m_f}},\frac{1}{\sqrt{m_g}}\}$. This completes the proof. $\qquad\square$

**Theorem 3 (Smoothness for DONs).** *Under Assumptions 2 and 3, with probability at least $1-2KL(\frac{1}{m_f}+\frac{1}{m_g})$, the DON loss $\mathcal{L}$ (3) is $\beta$-smooth in $B_{\rho,\rho_1}^{\mathrm{Euc}}(\boldsymbol{\theta}_0)$ with $\beta=c_2 K^2$, where $c_2>0$ is a constant which depends polynomially on the depth $L$, and the radii $\rho$, $\rho_1$, and $\rho_2$ whenever $\sigma_0\leq 1-\rho\max\{\frac{1}{\sqrt{m_f}},\frac{1}{\sqrt{m_g}}\}$.*

*Proof.* By the second order Taylor expansion of $\mathcal{L}(\boldsymbol{\theta}')$ about the point $\bar{\boldsymbol{\theta}}$ with $\boldsymbol{\theta}',\bar{\boldsymbol{\theta}}\in B_{\rho,\rho_1}^{\mathrm{Euc}}(\boldsymbol{\theta}_0)$, we have $\mathcal{L}(\boldsymbol{\theta}')=$

$\mathcal{L}(\bar{\boldsymbol{\theta}}) + \langle \boldsymbol{\theta}' - \bar{\boldsymbol{\theta}}, \nabla_{\boldsymbol{\theta}}\mathcal{L}(\bar{\boldsymbol{\theta}}) \rangle + \frac{1}{2}(\boldsymbol{\theta}' - \bar{\boldsymbol{\theta}})^{\top}\frac{\partial^2 \mathcal{L}(\tilde{\boldsymbol{\theta}})}{\partial \boldsymbol{\theta}^2}(\boldsymbol{\theta}' - \bar{\boldsymbol{\theta}})$, where $\tilde{\boldsymbol{\theta}} = \xi\boldsymbol{\theta}' + (1-\xi)\bar{\boldsymbol{\theta}}$ for some $\xi \in [0,1]$. Then,

$$(\boldsymbol{\theta}' - \bar{\boldsymbol{\theta}})^{\top}\frac{\partial^2 \mathcal{L}(\tilde{\boldsymbol{\theta}})}{\partial \boldsymbol{\theta}^2}(\boldsymbol{\theta}' - \bar{\boldsymbol{\theta}}) = (\boldsymbol{\theta}' - \bar{\boldsymbol{\theta}})^{\top}\left(\frac{1}{n}\sum_{i=1}^{n}\frac{1}{q_i}\sum_{j=1}^{q_i}\ell_{i,j}''\nabla_{\boldsymbol{\theta}}G_{\tilde{\boldsymbol{\theta}}}(u^{(i)})(\boldsymbol{y}_j^{(i)})\nabla_{\boldsymbol{\theta}}G_{\tilde{\boldsymbol{\theta}}}(u^{(i)})(\boldsymbol{y}_j^{(i)})^{\top}\right.$$

$$\left. + \ell_{i,j}'\nabla_{\boldsymbol{\theta}}^2 G_{\tilde{\boldsymbol{\theta}}}(u^{(i)})(\boldsymbol{y}_j^{(i)})\right)(\boldsymbol{\theta}' - \bar{\boldsymbol{\theta}})$$

$$= \underbrace{\frac{1}{n}\sum_{i=1}^{n}\frac{1}{q_i}\sum_{j=1}^{q_i}\ell_{i,j}''\left\langle \boldsymbol{\theta}' - \bar{\boldsymbol{\theta}}, \nabla_{\boldsymbol{\theta}}G_{\tilde{\boldsymbol{\theta}}}(u^{(i)})(\boldsymbol{y}_j^{(i)})\right\rangle^2}_{I_1}$$

$$+ \underbrace{\frac{1}{n}\sum_{i=1}^{n}\frac{1}{q_i}\sum_{j=1}^{q_i}\ell_{i,j}'(\boldsymbol{\theta}' - \bar{\boldsymbol{\theta}})^{\top}\nabla_{\boldsymbol{\theta}}^2 G_{\tilde{\boldsymbol{\theta}}}(u^{(i)})(\boldsymbol{y}_j^{(i)})(\boldsymbol{\theta}' - \bar{\boldsymbol{\theta}})}_{I_2},$$

where $\ell_{i,j} = (G_{\tilde{\boldsymbol{\theta}}}(u^{(i)})(\boldsymbol{y}_j^{(i)}) - G^{\dagger}(u^{(i)})(\boldsymbol{y}_j^{(i)}))^2$.

Now, note that

$$I_1 = \frac{1}{n}\sum_{i=1}^{n}\frac{1}{q_i}\sum_{j=1}^{q_i}\ell_{i,j}''\left\langle \boldsymbol{\theta}' - \bar{\boldsymbol{\theta}}, \nabla_{\boldsymbol{\theta}}G_{\tilde{\boldsymbol{\theta}}}(u^{(i)})(\boldsymbol{y}_j^{(i)})\right\rangle^2$$

$$\overset{(a)}{\leq} \frac{2}{n}\sum_{i=1}^{n}\frac{1}{q_i}\sum_{j=1}^{q_i}\left\|\nabla_{\boldsymbol{\theta}}G_{\tilde{\boldsymbol{\theta}}}(u^{(i)})(\boldsymbol{y}_j^{(i)})\right\|_2^2\|\boldsymbol{\theta}' - \bar{\boldsymbol{\theta}}\|_2^2$$

$$\overset{(b)}{\leq} 4K^2(\lambda_2\varrho^{(f)} + \lambda_1\varrho^{(g)})^2\|\boldsymbol{\theta}' - \bar{\boldsymbol{\theta}}\|_2^2,$$

where (a) follows by the Cauchy-Schwartz inequality and (b) from Lemma 3 as follows

$$\|\nabla_{\boldsymbol{\theta}}G_{\tilde{\boldsymbol{\theta}}}(u^{(i)})(\boldsymbol{y}_j^{(i)})\|_2 \leq \sum_{k=1}^{K}(\|g_{k,j}^{(i)}(\tilde{\boldsymbol{\theta}}_g)\nabla_{\boldsymbol{\theta}_f}f_k^{(i)}(\tilde{\boldsymbol{\theta}}_f)\|_2 + \|f_k^{(i)}(\tilde{\boldsymbol{\theta}}_f)\nabla_{\boldsymbol{\theta}_g}g_{k,j}^{(i)}(\tilde{\boldsymbol{\theta}}_g)\|_2) \leq K(\lambda_2\varrho^{(f)} + \lambda_1\varrho^{(g)}),$$

since $\tilde{\boldsymbol{\theta}} \in B_{\rho,\rho_1}^{\text{Euc}}(\boldsymbol{\theta}_0)$.

Now, for $I_2$,

$$I_2 \leq \frac{1}{n}\sum_{i=1}^{n}\frac{1}{q_i}\sum_{j=1}^{q_i}|\ell_{i,j}'||(\boldsymbol{\theta}' - \bar{\boldsymbol{\theta}})^{\top}\nabla^2 G_{\tilde{\boldsymbol{\theta}}}(u^{(i)})(\boldsymbol{y}_j^{(i)})(\boldsymbol{\theta}' - \bar{\boldsymbol{\theta}})|$$

$$\overset{(a)}{\leq} (2K\lambda_1\lambda_2 + \tilde{c})\left(K\varrho^{(f)}\varrho^{(g)} + K(1+\rho_2)\left(\frac{c^{(g)}(\lambda_1 + \varrho^{(f)})}{\sqrt{m_g}} + \frac{c^{(f)}(\lambda_2 + \varrho^{(g)})}{\sqrt{m_f}}\right)\right)\|\boldsymbol{\theta}' - \bar{\boldsymbol{\theta}}\|_2^2,$$

with $\tilde{c} = \max_{i\in[n],j\in[q_i]}|G^{\dagger}(u^{(i)})(\boldsymbol{y}_j^{(i)})|$, and where (a) follows from modifying the result in equation (36) from Theorem 2 according to our setting.

Putting the upper bounds on $I_1$ and $I_2$ back, we have

$$(\boldsymbol{\theta}' - \bar{\boldsymbol{\theta}})^{\top}\frac{\partial^2 \mathcal{L}(\tilde{\boldsymbol{\theta}})}{\partial \boldsymbol{\theta}^2}(\boldsymbol{\theta}' - \bar{\boldsymbol{\theta}}) \leq \left[4K^2(\lambda_2\varrho^{(f)} + \lambda_1\varrho^{(g)})^2\right.$$

$$\left. + (2K\lambda_1\lambda_2 + \tilde{c})\left(K\varrho^{(f)}\varrho^{(g)} + K(1+\rho_2)\left(\frac{c^{(g)}(\lambda_1 + \varrho^{(f)})}{\sqrt{m_g}} + \frac{c^{(f)}(\lambda_2 + \varrho^{(g)})}{\sqrt{m_f}}\right)\right)\right]$$

$$\times \|\boldsymbol{\theta}' - \bar{\boldsymbol{\theta}}\|_2^2.$$

Note that all the constants on the right-hand side of the inequality above form an expression that depends on $K$ and on $\sigma_1$, the depth $L$, and the radii $\rho$, $\rho_1$, and $\rho_2$ due to Lemma 3. As in the statement of Lemma 3, the dependence of such expression

reduces to the depth and the radii and becomes polynomial whenever $\sigma_0 \leq 1 - \rho \max\{\frac{1}{\sqrt{m_f}}, \frac{1}{\sqrt{m_g}}\}$. This completes the proof. $\qquad\square$

**Proposition 1** (**RSC to smoothness ratio**)**.** *Under the same conditions as in Theorems 2 and 3, we have that $\alpha_t/\beta < 1$ with probability at least $1 - 2LK(\frac{1}{m_f} + \frac{1}{m_g})$.*

*Proof.* From the proofs of both Theorems 2 and 3, $\alpha_t < 2\kappa^2 \|\nabla_{\boldsymbol{\theta}_t} \bar{G}_t\|_2^2 \leq 2\kappa^2 K^2 (\lambda_2 \varrho^{(f)} + \lambda_1 \varrho^{(g)})^2 \leq 4K^2 (\lambda_2 \varrho^{(f)} + \lambda_1 \varrho^{(g)})^2 < \beta$, and so $\frac{\alpha_t}{\beta} < 1$. $\qquad\square$

# E. Analysis for Fourier Neural Operators

We recall the FNO model

$$
\begin{aligned}
\boldsymbol{\alpha}^{(0)} &= P(u)(\boldsymbol{x}) \\
\boldsymbol{\alpha}^{(1)} &= \phi\left(\frac{1}{\sqrt{m}} W^{(1)} \boldsymbol{\alpha}^{(0)}\right) \\
\boldsymbol{\alpha}^{(l)} &= \phi\left(\frac{1}{\sqrt{m}} W^{(l)} \boldsymbol{\alpha}^{(l-1)} + \frac{1}{\sqrt{m}} F^* R^{(l)} F \boldsymbol{\alpha}^{(l-1)}\right), \quad l \in \{2, \dots, L+1\} \\
f(\boldsymbol{\theta}; \mathbf{x}) = \boldsymbol{\alpha}^{(L+2)} &:= \frac{1}{\sqrt{m}} \mathbf{v}^\top \boldsymbol{\alpha}^{(L+1)},
\end{aligned}
\tag{39}
$$

where $W^{(l)}, R^{(l)} \in \mathbb{R}^{m \times m}$ for $l \in \{2, \dots, L+1\}$, $W^{(1)} \in \mathbb{R}^{m \times d}$.

## E.1. Bounds on the Hessian, Gradients and the Predictor

**Lemma 4** (**Bounds on the Predictor**)**.** *Under Assumptions 5 and 6 and for $\boldsymbol{\theta} \in B_{\rho_w, \rho_r \rho_1}^{\mathrm{Euc}}(\boldsymbol{\theta}_0)$ we have with probability at least $1 - \frac{2(L+2)}{m}$, that for any input function $u$ and evaluation point $\boldsymbol{x}$ as in Section 6,*

$$
\left\|\nabla_{\boldsymbol{\theta}}^2 f\right\| \leq \frac{c}{\sqrt{m}}, \tag{40}
$$

$$
\|\nabla_{\boldsymbol{\theta}} f\|_2 \leq \varrho, \tag{41}
$$

$$
|f| \leq \lambda, \tag{42}
$$

*where $c$, $\varrho$, $\lambda$ are suitable constants that depend on $\sigma_{1,w}$, $\sigma_{1,r}$, the depth $L$, and the radii $\rho_w$, $\rho_r$, and $\rho_1$. The dependence of the constants reduces to depth and the radii and becomes polynomial whenever $\sigma_{1,w} + \sigma_{1,r} \leq 1 - \frac{\rho_w + \rho_r}{\sqrt{m}}$.*

In this section we will prove all the bounds in Lemma 4.

**Lemma 5** (**Initialization of the Parameters**)**.** *Under Assumption 6, with probability at least $1 - \frac{2}{m}$ we have*

$$
\|W_0^{(l)}\|_2 \leq \sigma_{1,w} \sqrt{m}, \quad \text{and} \quad \|R_0^{(l)}\|_2 \leq \sigma_{1,r} \sqrt{m}. \tag{43}
$$

*Proof.* The proof follows directly from Lemma A.1 in (Banerjee et al., 2023a).

**Proposition 2** (**Layer-wise matrices**)**.** *Under Assumption 6, for $\boldsymbol{\theta} \in B_{\rho_w, \rho_r \rho_1}^{\mathrm{Euc}}(\boldsymbol{\theta}_0)$, with probability at least $1 - \frac{2}{m}$ we have*

$$
\left\|W^{(l)}\right\|_2 \leq \left(\sigma_{1,w} + \frac{\rho_w}{\sqrt{m}}\right)\sqrt{m}, \ l \in [L+1] \quad \text{and} \quad \left\|R^{(l)}\right\|_2 \leq \left(\sigma_{1,r} + \frac{\rho_r}{\sqrt{m}}\right)\sqrt{m}, \ l \in \{2, \dots, L+1\} \tag{44}
$$

*Proof.* By the triangle inequality and Lemma 5,

$$
\begin{aligned}
\|W^{(l)}\|_2 &\leq \|W_0^{(l)}\|_2 + \|W^{(l)} - W_0^{(l)}\|_2 \leq \sigma_{1,w} \sqrt{m} + \rho_w, \\
\|R^{(l)}\|_2 &\leq \|R_0^{(l)}\|_2 + \|R^{(l)} - R_0^{(l)}\|_2 \leq \sigma_{1,r} \sqrt{m} + \rho_r.
\end{aligned}
$$

$\qquad\square$

We now bound the norm of the output $\boldsymbol{\alpha}^{(l)}$ at the layer $l \in [L+1]$.

**Lemma 6** (**Norm of the $l$-th layer output**). *For $l \in [L+1]$, under Assumptions 5 and 6 for $\theta \in B^{\mathrm{Euc}}_{\rho_w, \rho_r \rho_1}(\boldsymbol{\theta}_0)$, with probability at least $1 - \frac{2l}{m}$, we have*

$$\left\| \boldsymbol{\alpha}^{(l)} \right\|_2 \leq \sqrt{m} \left( \sigma_1 + \frac{\rho}{\sqrt{m}} \right)^l + \sqrt{m} \sum_{i=1}^{l} \left( \sigma_1 + \frac{\rho}{\sqrt{m}} \right)^{i-1} |\phi(0)| = \left( \gamma^l + |\phi(0)| \sum_{i=1}^{l} \gamma^{i-1} \right) \sqrt{m}, \qquad (45)$$

*where,*

$$\sigma_1 = \sigma_{1,w} + \sigma_{1,r}, \quad \rho = \rho_w + \rho_r, \quad and \quad \gamma = \sigma_1 + \frac{\rho}{\sqrt{m}}.$$

*Proof.* We prove the result using induction (e.g., see Lemma A.2 in (Banerjee et al., 2023a)). First, note that for the first hidden layer, using the fact that $\phi$ is 1-Lipschitz,

$$\left\| \phi\left( \frac{1}{\sqrt{d}} W^{(1)} \boldsymbol{\alpha}^{(0)} \right) \right\|_2 - \|\phi(\mathbf{0})\|_2 \leq \left\| \phi\left( \frac{1}{\sqrt{d}} W^{(1)} \boldsymbol{\alpha}^{(0)} \right) - \phi(\mathbf{0}) \right\|_2 \leq \left\| \frac{1}{\sqrt{d}} W^{(1)} \boldsymbol{\alpha}^{(0)} \right\|_2, \qquad (46)$$

where $\mathbf{0}$ denotes the zero vector of appropriate size. This in turn gives, using $\|\boldsymbol{\alpha}^{(0)}\|_2 = \sqrt{d}$,

$$\begin{aligned}
\left\| \boldsymbol{\alpha}^{(1)} \right\|_2 &= \left\| \phi\left( \frac{1}{\sqrt{d}} W^{(1)} \boldsymbol{\alpha}^{(0)} \right) \right\|_2 \leq \left\| \frac{1}{\sqrt{d}} W^{(1)} \boldsymbol{\alpha}^{(0)} \right\|_2 + \|\phi(\mathbf{0})\|_2 \\
&\leq \frac{1}{\sqrt{d}} \left\| W^{(1)} \right\|_2 \left\| \boldsymbol{\alpha}^{(0)} \right\|_2 + |\phi(0)|\sqrt{m} \\
&\leq \left( \sigma_{1,w} + \frac{\rho_w}{\sqrt{m}} \right) \sqrt{m} + |\phi(0)|\sqrt{m} \\
&\leq \left( \sigma_{1,w} + \sigma_{1,r} + \frac{\rho_w + \rho_r}{\sqrt{m}} \right) \sqrt{m} + |\phi(0)|\sqrt{m} \ .
\end{aligned}$$

Now, consider also the output at layer 2, namely,

$$\|\boldsymbol{\alpha}^{(2)}\|_2 = \left\| \phi\left( \frac{1}{\sqrt{m}} W^{(2)} \boldsymbol{\alpha}^{(1)} + \frac{1}{\sqrt{m}} F^* R^{(2)} F \boldsymbol{\alpha}^{(1)} \right) \right\|_2,$$

which gives,

$$\begin{aligned}
&\left\| \phi\left( \frac{1}{\sqrt{m}} W^{(2)} \boldsymbol{\alpha}^{(1)} + \frac{1}{\sqrt{m}} F^* R^{(2)} F \boldsymbol{\alpha}^{(1)} \right) \right\|_2 - \|\phi(\mathbf{0})\|_2 \\
&\leq \left\| \phi\left( \frac{1}{\sqrt{m}} W^{(2)} \boldsymbol{\alpha}^{(1)} + \frac{1}{\sqrt{m}} F^* R^{(2)} F \boldsymbol{\alpha}^{(1)} \right) - \phi(\mathbf{0}) \right\|_2 \leq \left\| \frac{1}{\sqrt{m}} W^{(2)} \boldsymbol{\alpha}^{(1)} + \frac{1}{\sqrt{m}} F^* R^{(2)} F \boldsymbol{\alpha}^{(1)} \right\|_2,
\end{aligned}$$

and, in turn,

$$\begin{aligned}
\|\boldsymbol{\alpha}^{(2)}\|_2 &\leq \left\| \frac{1}{\sqrt{m}} W^{(2)} \boldsymbol{\alpha}^{(1)} + \frac{1}{\sqrt{m}} F^* R^{(2)} F \boldsymbol{\alpha}^{(1)} \right\|_2 + \|\phi(\mathbf{0})\|_2 \\
&\leq \left\| \frac{1}{\sqrt{m}} W^{(2)} \boldsymbol{\alpha}^{(1)} \right\|_2 + \left\| \frac{1}{\sqrt{m}} F^* R^{(2)} F \boldsymbol{\alpha}^{(1)} \right\|_2 + |\phi(0)|\sqrt{m} \\
&\overset{(a)}{\leq} \frac{1}{\sqrt{m}} \|W^{(2)}\|_2 \|\boldsymbol{\alpha}^{(1)}\|_2 + \frac{1}{\sqrt{m}} \|R^{(2)}\|_2 \|\boldsymbol{\alpha}^{(1)}\|_2 + \sqrt{m}|\phi(0)| \\
&\leq \left( \sigma_{1,w} + \frac{\rho_w}{\sqrt{m}} + \sigma_{1,r} + \frac{\rho_r}{\sqrt{m}} \right) \|\boldsymbol{\alpha}^{(1)}\|_2 + \sqrt{m}|\phi(0)| \\
&\leq \sqrt{m} \left( \sigma_1 + \frac{\rho}{\sqrt{m}} \right)^2 + \left( 1 + \left( \sigma_1 + \frac{\rho}{\sqrt{m}} \right) \right) \sqrt{m}|\phi(0)|,
\end{aligned}$$

where (a) follows from the fact that the operator $F$ is a unitary matrix. Now, for the inductive step, consider that the output at layer $l-1$ satisfies

$$\left\|\boldsymbol{\alpha}^{(l-1)}\right\|_2 \leq \sqrt{m}\left(\sigma_1 + \frac{\rho}{\sqrt{m}}\right)^{l-1} + \sqrt{m}\sum_{i=1}^{l-1}\left(\sigma_1 + \frac{\rho}{\sqrt{m}}\right)^{i-1}|\phi(0)|.$$

Finally, at layer $l$, we have

$$\left\|\boldsymbol{\alpha}^{(l)}\right\|_2 \leq \frac{1}{\sqrt{m}}\left(\underbrace{\left\|W^{(l)}\right\|_2 + \left\|F^*R^{(l)}F\right\|_2}\right)\|\boldsymbol{\alpha}^{(l-1)}\|_2 + \sqrt{m}|\phi(0)| \tag{47}$$

$$\leq \left(\sigma_{1,w} + \sigma_{1,r} + \frac{\rho_w + \rho_r}{\sqrt{m}}\right)\|\boldsymbol{\alpha}^{(l-1)}\|_2 + \sqrt{m}|\phi(0)| \tag{48}$$

$$\leq \sqrt{m}\left(\sigma_1 + \frac{\rho}{\sqrt{m}}\right)^l + \sqrt{m}\sum_{i=1}^l\left(\sigma_1 + \frac{\rho}{\sqrt{m}}\right)^{i-1}|\phi(0)|. \tag{49}$$

Introducing $\gamma = \sigma_1 + \frac{\rho}{\sqrt{m}}$, we can write

$$\|\boldsymbol{\alpha}^{(l)}\|_2 \leq \sqrt{m}\left(\gamma^l + |\phi(0)|\sum_{i=1}^l\gamma^{i-1}\right). \tag{50}$$

This completes the proof. $\qquad\square$

From now on, we will use the notation $\rho$, $\sigma_1$, and $\gamma$ as defined in Lemma 6.

**Lemma 7.** *For $l \in \{2, \ldots, L+1\}$, under Assumptions 5 and 6 for $\theta \in B_{\rho_w, \rho_r \rho_1}^{\text{Euc}}(\theta_0)$, with probability at least $1 - \frac{2}{m}$, we have*

$$\left\|\frac{\partial \boldsymbol{\alpha}^{(l)}}{\partial \boldsymbol{\alpha}^{(l-1)}}\right\|_2 \leq \gamma. \tag{51}$$

*Proof.* We first note that

$$\left[\frac{\partial \boldsymbol{\alpha}^{(l)}}{\partial \boldsymbol{\alpha}^{(l-1)}}\right]_{ij} = \frac{1}{\sqrt{m}}\phi'(\widetilde{\boldsymbol{\alpha}}^{(l-1)})\left[W_{ij}^{(l)} + [F^*R^{(l)}F]_{ij}\right].$$

Now, from the definition $\|A\|_2 = \sup_{\|\boldsymbol{v}\|_2=1}\|A\boldsymbol{v}\|_2$ we have,

$$\left\|\frac{\partial \boldsymbol{\alpha}^{(l)}}{\partial \boldsymbol{\alpha}^{(l-1)}}\right\|_2 = \sup_{\|\boldsymbol{v}\|_2=1}\frac{1}{\sqrt{m}}\left(\phi'\left\|\left(W^{(l)} + F^*R^{(l)}F\right)\boldsymbol{v}\right\|_2\right)$$

$$\overset{(a)}{\leq} \sup_{\|\boldsymbol{v}\|_2=1}\frac{1}{\sqrt{m}}\left(\|W^{(l)}\boldsymbol{v}\|_2 + \|F^*R^{(l)}F\boldsymbol{v}\|_2\right)$$

$$\overset{(b)}{=} \sup_{\|\boldsymbol{v}\|_2=1}\frac{1}{\sqrt{m}}\left(\|W^{(l)}\boldsymbol{v}\|_2 + \|R^{(l)}F\boldsymbol{v}\|_2\right) \tag{52}$$

$$\overset{(c)}{\leq} \sup_{\|\boldsymbol{v}\|_2=1}\frac{1}{\sqrt{m}}\left(\|W^{(l)}\|_2\|\boldsymbol{v}\|_2 + \|R^{(l)}\|_2\|\boldsymbol{v}\|_2\right)$$

$$= \frac{1}{\sqrt{m}}\left(\|W^{(l)}\|_2 + \|R^{(l)}\|_2\right),$$

where $(a)$ follows from the fact that $\phi$ is 1-Lipchitz and by using the triangle inequality, and $(b)$ and $(c)$ follow from the fact that $F^*$ and $F$ are isometries with respect to the $L_2$-norm, i.e. $\|F\boldsymbol{v}\|_2 = \|\boldsymbol{v}\|_2$ and $\|F^*\boldsymbol{v}\|_2 = \|\boldsymbol{v}\|_2$ for $\boldsymbol{v} \in \mathbb{R}^m$. This

finally gives

$$\left\| \frac{\partial \boldsymbol{\alpha}^{(l)}}{\partial \boldsymbol{\alpha}^{(l-1)}} \right\|_2 \leq \frac{1}{\sqrt{m}} \left( \|W^{(l)}\|_2 + \|R^{(l)}\|_2 \right) \leq \left( \sigma_{1,w} + \frac{\rho_w}{\sqrt{m}} \right) + \left( \sigma_{1,r} + \frac{\rho_r}{\sqrt{m}} \right)$$
$$= \gamma \,,$$

where we used Proposition 2. This completes the proof. $\qquad\square$

We make use of the Einstein summation convention, i.e. repeated indices imply summation, unless explicitly stated. We also use the notation $vec(\cdot)$ to denote the vectorization of the matrix argument according to some fixed manner (e.g., row-wise vectorization).

**Lemma 8.** *Under Assumptions 5 and 6 and for $\theta \in B^{\text{Euc}}_{\rho_w, \rho_r \rho_1}(\boldsymbol{\theta}_0)$, with probability at least $1 - \frac{2l}{m}$,*

$$\left\| \frac{\partial \boldsymbol{\alpha}^{(l)}}{\partial \mathbf{w}^{(l)}} \right\|_2, \left\| \frac{\partial \boldsymbol{\alpha}^{(l)}}{\partial \mathbf{r}^{(l)}} \right\|_2 \leq \left( \gamma^{l-1} + |\phi(0)| \sum_{i=1}^{l-1} \gamma^{i-1} \right)$$

*where, $\mathbf{w}^{(l)} = vec(W^{(l)})$ for $l \in [L+1]$, and $\mathbf{r}^{(l)} = vec(R^{(l)})$ for $l \in \{2, \ldots, L+1\}$.*

*Proof.* We can index the vectors $\boldsymbol{w}^{(l)}$ and $\mathbf{r}^{(l)}$ according to their matrix form $W^{(l)}_{jj'}$ and $R^{(l)}_{jj'}$, respectively, with the indices $j \in [m]$, and $j' \in [d]$ when $l = 1$ or $j' \in [m]$ when $l \in \{2, \ldots, L+1\}$. Therefore,

$$\left[ \frac{\partial \boldsymbol{\alpha}^{(l)}}{\partial \mathbf{w}^{(l)}} \right]_{i,jj'} = \frac{1}{\sqrt{m}} \phi'(\widetilde{\boldsymbol{\alpha}}^{(l)}_i) \delta_{ij} \boldsymbol{\alpha}^{(l-1)}_{j'}, \quad \delta_{ij} = \begin{cases} 1 & i = j \\ 0 & \text{otherwise} \end{cases}.$$

Now, for $l \in \{2, \ldots, L+1\}$, we can write the $L_2$-norm of the matrices as follows

$$\left\| \frac{\partial \boldsymbol{\alpha}^{(l)}}{\partial \mathbf{w}^{(l)}} \right\|_2^2 = \sup_{\|V\|_F = 1} \frac{1}{m} \sum_{i=1}^{m} \left( \phi'\left(\widetilde{\boldsymbol{\alpha}}^{(l)}_i\right) \sum_{j,j'=1}^{m} \boldsymbol{\alpha}^{(l-1)}_{j'} \delta_{ij} V_{jj'} \right)^2$$
$$\leq \sup_{\|V\|_F = 1} \frac{1}{m} \|V \boldsymbol{\alpha}^{(l-1)}\|_2^2$$
$$\leq \sup_{\|V\|_F = 1} \frac{1}{m} \|V\|_2^2 \|\boldsymbol{\alpha}^{(l-1)}\|_2^2$$
$$\overset{(a)}{\leq} \sup_{\|V\|_F = 1} \frac{1}{m} \|V\|_F^2 \|\boldsymbol{\alpha}^{(l-1)}\|_2^2$$
$$= \frac{1}{m} \|\boldsymbol{\alpha}^{(l-1)}\|_2^2$$
$$\overset{(b)}{\leq} \frac{1}{m} \left[ \sqrt{m} \left( \gamma^{l-1} + |\phi(0)| \sum_{i=1}^{l-1} \gamma^{i-1} \right) \right]^2 = \left( \gamma^{l-1} + |\phi(0)| \sum_{i=1}^{l-1} \gamma^{i-1} \right)^2,$$

where $(a)$ follows from the fact that $\|V\|_2 \leq \|V\|_F$ and $(b)$ from Lemma 6. The $l = 1$ case follows in a similar fashion:

$$\left\| \frac{\partial \boldsymbol{\alpha}^{(1)}}{\partial \mathbf{w}^{(1)}} \right\|_2^2 \leq \frac{1}{d} \|\boldsymbol{\alpha}^{(0)}\|_2^2 = 1.$$

Similarly, for $l \in \{2, \ldots, L+1\}$,

$$
\begin{aligned}
\left\| \frac{\partial \boldsymbol{\alpha}^{(l)}}{\partial \mathbf{r}^{(l)}} \right\|_2^2 &= \sup_{\|V\|_F = 1} \frac{1}{m} \sum_{i=1}^m \left( \phi' \left( \widetilde{\boldsymbol{\alpha}}_i^{(l)} \right) F_{ij}^* F_{j'p} \boldsymbol{\alpha}_p^{(l-1)} V_{jj'} \right)^2 \\
&\leq \sup_{\|V\|_F = 1} \frac{1}{m} \| (F^* V F) \boldsymbol{\alpha}^{(l-1)} \|_2^2 \\
&\leq \sup_{\|V\|_F = 1} \frac{1}{m} \| F^* V F \|_2^2 \| \boldsymbol{\alpha}^{(l-1)} \|_2^2 \\
&\leq \sup_{\|V\|_F = 1} \frac{1}{m} \| F^* \|_2^2 \| V \|_2^2 \| F \|_2^2 \| \boldsymbol{\alpha}^{(l-1)} \|_2^2 \\
&\overset{(a)}{\leq} \sup_{\|V\|_F = 1} \frac{1}{m} \| V \|_F^2 \| \boldsymbol{\alpha}^{(l-1)} \|_2^2 \\
&= \frac{1}{m} \| \boldsymbol{\alpha}^{(l-1)} \|_2^2 \\
&\overset{(b)}{\leq} \frac{1}{m} \left[ \sqrt{m} \left( \gamma^{l-1} + |\phi(0)| \sum_{i=1}^{l-1} \gamma^{i-1} \right) \right]^2 = \left( \gamma^{l-1} + |\phi(0)| \sum_{i=1}^{l-1} \gamma^{i-1} \right)^2,
\end{aligned}
$$

where $(a)$ follows again by $\|V\|_2 \leq \|V\|_F$ and the fact that $F^*$ and $F$ are unitary matrices, and $(b)$ from Lemma 6. This completes the proof. $\qquad \square$

**Hessians.** We now focus on bounding the Hessian of the predictor $f$ in equation (39). Note that the FNO model can be considered as having $L+1$ layers, with Layer 1 being a feedforward single layer encoder on top of the encoder $P$, the $L$ layers from Layer 2 to Layer $L+1$ being FNO hidden layers, and Layer $L+2$ being the output of the linear decoder. Likewise, we decompose the Hessian matrix $\boldsymbol{H}$ of the FNO in three different blocks corresponding to the aforementioned encoder, FNO hidden layers, and decoder, respectively.

Firstly, the Hessian blocks associated to the hidden FNO layers are:

- the $L \times L$ sub-blocks corresponding to $H_w^{(l_1, l_2)} := \frac{\partial^2 f}{\partial \mathbf{w}^{(l_1)} \partial \mathbf{w}^{(l_2)}}$ for $l_1, l_2 \in \{2, \ldots, L+1\}$,

- the $L \times L$ sub-blocks corresponding to $H_r^{(l_1, l_2)} := \frac{\partial^2 f}{\partial \mathbf{r}^{(l_1)} \partial \mathbf{r}^{(l_2)}}$ for $l_1, l_2 \in \{2, \ldots, L+1\}$, and

- the cross blocks have terms of the form $H_{w,r}^{(l_1, l_2)} := \frac{\partial^2 f}{\partial \mathbf{w}^{(l_1)} \partial \mathbf{r}^{(l_2)}}$ for $l_1, l_2 \in \{2, \ldots, L+1\}$.

Secondly, the Hessian blocks corresponding to the single layer encoder, i.e., with respect to weight $W^{(1)}$:

- diagonal block $H_w^{(1,1)} := \frac{\partial^2 f}{\partial \mathbf{w}^{(1)2}}$,

- off-diagonal blocks $H_w^{(1,l_1)} := \frac{\partial^2 f}{\partial \mathbf{w}^{(1)} \partial \mathbf{w}^{(l_1)}}$ and $H_w^{(l_1, 1)}$ for $l_1 \in \{2, \ldots, L+1\}$, and

- off-diagonal blocks $H_{w,r}^{(1, l_2)} := \frac{\partial^2 f}{\partial \mathbf{w}^{(1)} \partial \mathbf{r}^{(l_2)}}$ ans $H_{r,w}^{(l_2, 1)}$ for $l_2 \in \{2, \ldots, L+1\}$.

Finally, the Hessian blocks corresponding to the decoder, i.e., with respect to weight $\mathbf{v}$:

- diagonal block $H_v := \frac{\partial^2 f}{\partial \mathbf{v}^2}$, which is the zero matrix $\mathbf{0}_{m \times m}$,

- off-diagonal block $H_{w,v}^{(l_1)} := \frac{\partial^2 f}{\partial \mathbf{w}^{(l_1)} \partial \mathbf{v}}$ and $H_{v,w}^{(l_1)}$ for $l_1 \in \{1, \ldots, L+1\}$, and

- off-diagonal block $H_{r,v}^{(l_2)} := \frac{\partial^2 f}{\partial \mathbf{r}^{(l_2)} \partial \mathbf{v}}$ and $H_{v,r}^{(l_2)}$ for $l_2 \in \{2, \ldots, L+1\}$.

First, we note that due to the symmetry of the Hessian matrix of the FNO model $\boldsymbol{H}$:

$$\|\boldsymbol{H}\|_2 \leq \sum_{l_1,l_2=1}^{L+1} \|H_w^{(l_1,l_2)}\|_2 + \sum_{l_1,l_2=2}^{L+1} \|H_r^{(l_1,l_2)}\|_2 + 2\sum_{l_1=1}^{L+1}\sum_{l_2=2}^{L+1} \|H_{w,r}^{(l_1,l_2)}\|_2 + 2\sum_{l_1=1}^{L+1} \|H_{w,v}^{(l_1)}\|_2 + 2\sum_{l_2=2}^{L+1} \|H_{r,v}^{(l_2)}\|_2 . \quad (53)$$

We define

$$\mathcal{Q}_\infty(f) := \max_{l\in[L+1]} \left\|\frac{\partial f}{\partial \boldsymbol{\alpha}^{(l)}}\right\|_\infty ,$$

$$\mathcal{Q}_2^{(w,r)}(f) := \max_{l\in[L+1]} \left\{\left\|\frac{\partial \boldsymbol{\alpha}^{(l)}}{\partial \mathbf{w}^{(l)}}\right\|_2, \left\|\frac{\partial \boldsymbol{\alpha}^{(l)}}{\partial \mathbf{r}^{(l)}}\right\|_2\right\} ,$$

$$\mathcal{Q}_{2,2,1}^{(w,r)}(f) := \max_{\substack{1\leq l_1\leq L+1 \\ 2\leq l_2\leq L+1 \\ 3\leq l_3\leq L+1}} \left\{\left\|\frac{\partial^2 \boldsymbol{\alpha}^{(l_2)}}{\partial \mathbf{w}^{(l_2)}\partial \mathbf{r}^{(l_2)}}\right\|_{2,2,1}, \left\|\frac{\partial \boldsymbol{\alpha}^{(l_1)}}{\partial \mathbf{w}^{(l_1)}}\right\|_2 \left\|\frac{\partial^2 \boldsymbol{\alpha}^{(l_2)}}{\partial \boldsymbol{\alpha}^{(l_2-1)}\partial \mathbf{r}^{(l_2)}}\right\|_{2,2,1}, \right.$$

$$\left. \left\|\frac{\partial \boldsymbol{\alpha}^{(l_1)}}{\partial \mathbf{r}^{(l_1)}}\right\|_2 \left\|\frac{\partial^2 \boldsymbol{\alpha}^{(l_2)}}{\partial \boldsymbol{\alpha}^{(l_2-1)}\partial \mathbf{w}^{(l_2)}}\right\|_{2,2,1}, \left\|\frac{\partial \boldsymbol{\alpha}^{(l_1)}}{\partial \mathbf{w}^{(l_1)}}\right\|_2 \left\|\frac{\partial \boldsymbol{\alpha}^{(l_2)}}{\partial \mathbf{r}^{(l_2)}}\right\|_2 \left\|\frac{\partial^2 \boldsymbol{\alpha}^{(l_3)}}{(\partial \boldsymbol{\alpha}^{(l_3-1)})^2}\right\|_{2,2,1}\right\} ,$$

$$\mathcal{Q}_{2,2,1}^{(w)}(f) := \max_{\substack{1\leq l_1\leq L+1 \\ 2\leq l_2\leq L+1 \\ 3\leq l_3\leq L+1}} \left\{\left\|\frac{\partial^2 \boldsymbol{\alpha}^{(l_1)}}{(\partial \mathbf{w}^{(l_1)})^2}\right\|_{2,2,1}, \left\|\frac{\partial \boldsymbol{\alpha}^{(l_1)}}{\partial \mathbf{w}^{(l_1)}}\right\|_2 \left\|\frac{\partial^2 \boldsymbol{\alpha}^{(l_2)}}{\partial \boldsymbol{\alpha}^{(l_2-1)}\partial \mathbf{w}^{(l_2)}}\right\|_{2,2,1}, \right.$$

$$\left. \left\|\frac{\partial \boldsymbol{\alpha}^{(l_1)}}{\partial \mathbf{w}^{(l_1)}}\right\|_2 \left\|\frac{\partial \boldsymbol{\alpha}^{(l_2)}}{\partial \mathbf{w}^{(l_2)}}\right\|_2 \left\|\frac{\partial^2 \boldsymbol{\alpha}^{(l_3)}}{(\partial \boldsymbol{\alpha}^{(l_3-1)})^2}\right\|_{2,2,1}\right\} ,$$

$$\mathcal{Q}_{2,2,1}^{(r)}(f) := \max_{\substack{2\leq l_1\leq L+1 \\ 3\leq l_2\leq L+1 \\ 4\leq l_3\leq L+1}} \left\{\left\|\frac{\partial^2 \boldsymbol{\alpha}^{(l_1)}}{(\partial \mathbf{r}^{(l_1)})^2}\right\|_{2,2,1}, \left\|\frac{\partial \boldsymbol{\alpha}^{(l_1)}}{\partial \mathbf{r}^{(l_1)}}\right\|_2 \left\|\frac{\partial^2 \boldsymbol{\alpha}^{(l_2)}}{\partial \boldsymbol{\alpha}^{(l_2-1)}\partial \mathbf{r}^{(l_2)}}\right\|_{2,2,1}, \right.$$

$$\left. \left\|\frac{\partial \boldsymbol{\alpha}^{(l_1)}}{\partial \mathbf{r}^{(l_1)}}\right\|_2 \left\|\frac{\partial \boldsymbol{\alpha}^{(l_2)}}{\partial \mathbf{r}^{(l_2)}}\right\|_2 \left\|\frac{\partial^2 \boldsymbol{\alpha}^{(l_3)}}{(\partial \boldsymbol{\alpha}^{(l_3-1)})^2}\right\|_{2,2,1}\right\} ,$$

$$(54)$$

where, for an order-3 tensor $T \in \mathbb{R}^{d_1\times d_2\times d_3}$ we define the operator $\|\cdot\|_{2,2,1}$ as follows,

$$\|T\|_{2,2,1} := \sup_{\|\mathbf{a}\|_2=\|\mathbf{b}\|_2=1} \sum_{k=1}^{d_3} \left|\sum_{i=1}^{d_1}\sum_{j=1}^{d_2} T_{ijk} a_i b_j\right| , \quad \mathbf{a}\in\mathbb{R}^{d_1}, \mathbf{b}\in\mathbb{R}^{d_2} . \quad (55)$$

Note that it seems from (54) that we need the depth $L$ of the FNO to be $L \geq 3$. However, the bounds presented in Lemma 4 also hold for FNOS with depth $L < 3$: indeed, the upper bounds we derive in this section for an FNO with depth $L$ will trivially hold for FNOS with depths $L-1,\ldots,1$.

**Lemma 9.** *Under Assumptions 5 and 6 for $\boldsymbol{\theta} \in B_{\rho_w,\rho_r\rho_1}^{\mathrm{Euc}}(\boldsymbol{\theta}_0)$, the following inequalities hold with probability at least $1 - \frac{2(L+2)}{m}$, for $l_1 \in [L+1]$,*

$$\left\|\frac{\partial^2 \boldsymbol{\alpha}^{(l_1)}}{(\partial \mathbf{w}^{(l_1)})^2}\right\|_{2,2,1} \leq \beta_\phi(1+\gamma^L)^2(1+L|\phi(0)|)^2 , \quad (56)$$

*and for $l_2 \in \{2,\ldots,L+1\}$,*

$$\left\|\frac{\partial^2 \boldsymbol{\alpha}^{(l_2)}}{\partial \mathbf{w}^{(l_2)}\partial \mathbf{r}^{(l_2)}}\right\|_{2,2,1} \leq \beta_\phi(1+\gamma^L)^2(1+L|\phi(0)|)^2 , \quad (57)$$

$$\left\|\frac{\partial^2 \boldsymbol{\alpha}^{(l_2)}}{(\partial \boldsymbol{\alpha}^{(l_2-1)})^2}\right\|_{2,2,1} \leq 2\beta_\phi\gamma^2 , \quad (58)$$

$$\left\|\frac{\partial^2 \boldsymbol{\alpha}^{(l_2)}}{\partial \boldsymbol{\alpha}^{(l_2-1)} \partial \mathbf{w}^{(l_2)}}\right\|_{2,2,1} \le \beta_\phi (1+\gamma^L)^2 (1+(1+L|\phi(0)|)^2) + 1 \,, \tag{59}$$

$$\left\|\frac{\partial^2 \boldsymbol{\alpha}^{(l_2)}}{\partial \boldsymbol{\alpha}^{(l_2-1)} \partial \mathbf{r}^{(l_2)}}\right\|_{2,2,1} \le \beta_\phi (1+\gamma^L)^2 (1+(1+L|\phi(0)|)^2) + 1 \,, \; and \tag{60}$$

$$\left\|\frac{\partial^2 \boldsymbol{\alpha}^{(l_2)}}{(\partial \mathbf{r}^{(l_2)})^2}\right\|_{2,2,1} \le \beta_\phi (1+\gamma^L)^2 (1+L|\phi(0)|)^2 \,. \tag{61}$$

*Proof.* We first begin by proving (57). Note that from (39) we have

$$\frac{\partial^2 \boldsymbol{\alpha}_i^{(l_2)}}{\partial \mathbf{w}_{jj'}^{(l_2)} \partial \mathbf{r}_{kk'}^{(l_2)}} = \frac{1}{m} \phi'' \left(\tilde{\boldsymbol{\alpha}}^{(l_2)}\right) \cdot \boldsymbol{\alpha}_{j'}^{(l_2-1)} \delta_{ij} F_{ik}^* F_{k'q} \boldsymbol{\alpha}_q^{(l_2-1)},$$

where we make use of the Einstein notation. Now,

$$
\begin{aligned}
&\left\|\frac{\partial^2 \boldsymbol{\alpha}_i^{(l_2)}}{\partial \mathbf{w}^{(l_2)} \partial \mathbf{r}^{(l_2)}}\right\|_{2,2,1} \\
&= \sup_{\|V_1\|_F=1, \|V_2\|_F=1} \sum_{i=1}^m \left|\frac{1}{m} \phi''(\tilde{\boldsymbol{\alpha}}_i^{(l_2)}) \boldsymbol{\alpha}_{j'}^{(l_2-1)} \delta_{ij} F_{ik}^* F_{k'q} \boldsymbol{\alpha}_q^{(l_2-1)} V_{1_{jj'}} V_{2_{kk'}}\right| \\
&= \sup_{\|V_1\|_F=1, \|V_2\|_F=1} \sum_{i=1}^m \left|\frac{\phi''(\tilde{\boldsymbol{\alpha}}_i^{(l_2)})}{m} \left(V_{1_{ij'}}, \boldsymbol{\alpha}_{j'}^{(l_2-1)}\right) \left(F_{ik}^* V_{2_{kk'}} F_{k'q} \boldsymbol{\alpha}_q^{(l_2-1)}\right)\right| \\
&\le \sup_{\|V_1\|_F=1, \|V_2\|_F=1} \frac{\beta_\phi}{m} \sum_{i=1}^m \left|(V_1 \boldsymbol{\alpha}^{(l_2-1)})_i ((F^* V_2 F) \boldsymbol{\alpha}^{(l_2-1)})_i\right| \\
&\stackrel{(a)}{\le} \sup_{\|V_1\|_F=1, \|V_2\|_F=1} \frac{\beta_\phi}{2m} \left(\left\|V_1 \boldsymbol{\alpha}^{(l_2-1)}\right\|_2^2 + \left\|F^* V_2 F \boldsymbol{\alpha}^{(l_2-1)}\right\|_2^2\right) \\
&\stackrel{(b)}{\le} \frac{\beta_\phi}{2m} \left(\left\|\boldsymbol{\alpha}^{(l_2-1)}\right\|_2^2 + \left\|\boldsymbol{\alpha}^{(l_2-1)}\right\|_2^2\right) \le \beta_\phi \left(\gamma^{l_2-1} + |\phi(0)| \sum_{i=1}^{l_2-1} \gamma^{i-1}\right)^2,
\end{aligned}
\tag{62}
$$

where $(a)$ follows from the quadratic expression; where $(b)$ follows from $\|V_1 \boldsymbol{\alpha}^{(l_2-1)}\|_2 \le \|V_1\|_2 \|\boldsymbol{\alpha}^{(l_2-1)}\|_2$, $\|V_1\|_2 \le \|V_1\|_F$, $\|V_2\|_2 \le \|V_2\|_F$, $\|F^* V_2 F \boldsymbol{\alpha}^{(l_2-1)}\|_2 = \|V_2 F \boldsymbol{\alpha}^{(l_2-1)}\|_2 \le \|V_2\|_2 \|F \boldsymbol{\alpha}^{(l_2-1)}\|_2 = \|V_2\|_2 \|\boldsymbol{\alpha}^{(l_2-1)}\|_2$ due to $F$ being a unitary operator; and where the last inequality follows from (50). Finally, we can upper bound the last quantity above as in (57) and complete the proof.

For proving (58), again note from (39) that

$$
\begin{aligned}
\left[\frac{\partial^2 \boldsymbol{\alpha}^{(l_2)}}{\partial \boldsymbol{\alpha}^{(l_2-1)2}}\right]_{i,j,k} &= \frac{1}{m} \phi''(\tilde{\boldsymbol{\alpha}}^{(l_2)}) \left(W_{ij}^{(l_2)} + F_{ip}^* R_{pq}^{(l_2)} F_{qj}\right) \cdot \left(W_{ik}^{(l_2)} + F_{iu}^* R_{uv}^{(l_2)} F_{vk}\right) \\
&= \frac{\phi''}{m} \left[\underbrace{W_{ij}^{(l_2)} W_{ik}^{(l_2)}}_{T_1} + \underbrace{W_{ij}^{(l_2)} F_{iu}^* R_{uv}^{(l_2)} F_{vk}}_{T_2} + \underbrace{F_{ip}^* R_{pq}^{(l_2)} F_{qj} W_{ik}^{(l_2)}}_{T_3} + \underbrace{F_{ip}^* R_{pq}^{(l_2)} F_{qj} F_{iu}^* R_{uv}^{(l_2)} F_{vk}}_{T_4}\right].
\end{aligned}
\tag{63}
$$

Then, we can write

$$\left\|\frac{\partial^2 \boldsymbol{\alpha}^{(l_2)}}{\partial \boldsymbol{\alpha}^{(l_2-1)2}}\right\|_{2,2,1} = \sup_{\|\boldsymbol{v}_1\|_2=1, \|\boldsymbol{v}_2\|_2=1} \sum_{i=1}^m \left|\left[\frac{\partial^2 \boldsymbol{\alpha}^{(l_1)}}{\partial \boldsymbol{\alpha}^{(l_1-1)2}}\right]_{i,j,k} v_{1_j} v_{2_k}\right|.$$

Let us consider the notation $\gamma_w = \sigma_{1,w} + \frac{\rho_w}{\sqrt{m}}$ and $\gamma_r = \sigma_{1,r} + \frac{\rho_r}{\sqrt{m}}$. Now, we handle each of the terms separately:

$$
\begin{aligned}
\sup_{\|\boldsymbol{v}_1\|_2=1,\|\boldsymbol{v}_2\|_2=1} \sum_{i=1}^m \left| \frac{\phi''}{m} T_{1_{i,j,k}} v_{1_j} v_{2_k} \right| &= \frac{|\phi''|}{m} \sup_{\|\boldsymbol{v}_1\|_2=1,\|\boldsymbol{v}_2\|_2=1} \sum_{i=1}^m \left| \left( W_{ij}^{(l_2)} v_{1_j} \right) \cdot \left( W_{ik}^{(l_2)} v_{2_k} \right) \right| \\
&\leq \frac{\beta_\phi}{2m} \sup_{\|\boldsymbol{v}_1\|_2=1,\|\boldsymbol{v}_2\|_2=1} \left( \|W^{(l_2)}\|_2^2 \|\boldsymbol{v}_1\|_2^2 + \|W^{(l_2)}\|_2^2 \|\boldsymbol{v}_2\|_2^2 \right) \\
&= \beta_\phi \left( \sigma_{1,w} + \frac{\rho_w}{\sqrt{m}} \right)^2 = \beta_\phi \gamma_w^2.
\end{aligned}
\tag{64}
$$

$$
\begin{aligned}
\sup_{\|\boldsymbol{v}_1\|_2=1,\|\boldsymbol{v}_2\|_2=1} \sum_{i=1}^m \left| \frac{\phi''}{m} T_{4_{i,j,k}} v_{1_j} v_{2_k} \right| &= \frac{|\phi''|}{m} \sup_{\|\boldsymbol{v}_1\|_2=1,\|\boldsymbol{v}_2\|_2=1} \sum_{i=1}^m \left| \left( (F^* R^{(l_2)} F)_{ij} v_{1_j} \right) \cdot \left( (F^* R^{(l_2)} F)_{ik} v_{2_k} \right) \right| \\
&\leq \frac{\beta_\phi}{2m} \sup_{\|\boldsymbol{v}_1\|_2=1,\|\boldsymbol{v}_2\|_2=1} \left( \|F^* R^{(l_2)} F\|_2^2 \|\boldsymbol{v}_1\|_2^2 + \|F^* R^{(l_2)} F\|_2^2 \|\boldsymbol{v}_2\|_2^2 \right) \\
&= \frac{\beta_\phi}{m} \|F^* R^{(l_2)} F\|_2^2 \\
&\leq \frac{\beta_\phi}{m} \|R^{(l_2)}\|_2^2 \leq \beta_\phi \gamma_r^2.
\end{aligned}
\tag{65}
$$

$$
\begin{aligned}
\sup_{\|\boldsymbol{v}_1\|_2=1,\|\boldsymbol{v}_2\|_2=1} \sum_{i=1}^m \left| \frac{\phi''}{m} T_{2_{i,j,k}} v_{1_j} v_{2_k} \right| &= \frac{|\phi''|}{m} \sup_{\|\boldsymbol{v}_1\|_2=1,\|\boldsymbol{v}_2\|_2=1} \sum_{i=1}^m \left| (W_{ij}^{(l_2)} v_{1_j}) \cdot (F^* R^{(l_2)} F)_{ik} v_{2_k} \right| \\
&\leq \frac{\beta_\phi}{2m} \sup_{\|\boldsymbol{v}_1\|_2=1,\|\boldsymbol{v}_2\|_2=1} \left( \|W^{(l_2)}\|_2^2 \|\boldsymbol{v}_1\|_2^2 + \|F^* R^{(l_2)} F\|_2^2 \|\boldsymbol{v}_2\|_2^2 \right) \\
&\leq \frac{\beta_\phi}{2m} \left( \|W^{(l_2)}\|_2^2 + \|R^{(l_2)}\|_2^2 \right) \leq \frac{\beta_\phi}{2} \left( \gamma_w^2 + \gamma_r^2 \right).
\end{aligned}
\tag{66}
$$

Similarly, for the term corresponding to $T_3$ we obtain

$$
\sup_{\|\boldsymbol{v}_1\|_2=1,\|\boldsymbol{v}_2\|_2=1} \sum_{i=1}^m \left| \frac{\phi''}{m} T_{3_{i,j,k}} v_{1_j} v_{2_k} \right| \leq \frac{\beta_\phi}{2} \left( \gamma_w^2 + \gamma_r^2 \right).
\tag{67}
$$

Putting together (64), (65), (66) and (67), we get

$$
\left\| \frac{\partial^2 \boldsymbol{\alpha}^{(l_2)}}{\partial \boldsymbol{\alpha}^{(l_2-1)^2}} \right\|_{2,2,1}^2 \leq 2\beta_\phi(\gamma_w^2 + \gamma_r^2) \leq 2\beta_\phi(\gamma_w^2 + \gamma_r^2 + 2\gamma_w \gamma_r) = 2\beta_\phi \gamma^2.
\tag{68}
$$

This completes the proof for (58).

We now look at the proof for (60). First note that

$$
\begin{aligned}
\frac{\partial^2 \boldsymbol{\alpha}_i^{(l_2)}}{\partial \boldsymbol{\alpha}_k^{(l_2-1)} \partial \mathbf{r}_{jj'}^{(l_2)}} &= \frac{1}{m} \phi''(\tilde{\boldsymbol{\alpha}}_i) \left( W_{ik}^{(l_2)} + F_{ip}^* R_{pq}^{(l_2)} F_{qk} \right) F_{ij}^* F_{j'q} \boldsymbol{\alpha}_q^{(l_2-1)} + \frac{1}{\sqrt{m}} \phi'(\tilde{\boldsymbol{\alpha}}_i^{(l_2)}) F_{ij}^* F_{j'k} \\
&= \underbrace{\frac{\phi''}{m} \left( W_{ik}^{(l_2)} F_{ij}^* F_{j'q} \boldsymbol{\alpha}_q^{(l_2-1)} \right)}_{T_1} + \underbrace{\frac{\phi''}{m} \left( F_{ip}^* R_{pq}^{(l_2)} F_{qk} F_{ij}^* F_{j'q} \boldsymbol{\alpha}_q^{(l_2-1)} \right)}_{T_2} + \underbrace{\frac{1}{\sqrt{m}} \phi'(\tilde{\boldsymbol{\alpha}}_i^{(l_2)}) F_{ij}^* F_{j'k}}_{T_3}.
\end{aligned}
$$

Again, we analyze each of the terms separately

$$\left\|T_{1_{i,jj',k}}\right\|_{2,2,1} = \sup_{\|\boldsymbol{v}_1\|_2=1,\|V_2\|_F=1} \sum_{i=1}^{m} \left|\frac{\phi''}{m}\left(W_{ik}v_{1_k}^{(l_2)}F_{ij}^*V_{2_{jj'}},F_{j'q}\boldsymbol{\alpha}_q^{(l_2-1)}\right)\right|$$

$$\leq \frac{\beta_\phi}{2m}\left(\|W^{(l_2)}\boldsymbol{v}_1\|_2^2 + \|F^*V_2 F\boldsymbol{\alpha}^{(l_2-1)}\|_2^2\right) \tag{69}$$

$$\leq \frac{\beta_\phi}{2}\left(\gamma_w^2 + \left(\gamma^{l_2-1} + |\phi(0)|\sum_{i=1}^{l_2-1}\gamma^{i-1}\right)^2\right)$$

$$\left\|T_{2_{i,jj',k}}\right\|_{2,2,1} = \sup_{\|\boldsymbol{v}_1\|_2=1,\|V_2\|_F=1} \sum_{i=1}^{m} \left|\frac{\phi''}{m}\left(F_{ip}^*R_{pq}^{(l_2)}F_{qk}v_{1_k}F_{ij}^*V_{2_{jj'}},F_{j'q}\boldsymbol{\alpha}_q^{(l_2-1)}\right)\right|$$

$$\leq \sup_{\|\boldsymbol{v}_1\|_2=1,\|V_2\|_F=1}\frac{\beta_\phi}{2m}\left(\|F^*R^{(l_2)}F\boldsymbol{v}_1\|_2^2 + \|F^*V_2 F\boldsymbol{\alpha}^{(l_2-1)}\|_2^2\right)$$

$$\leq \sup_{\|\boldsymbol{v}_1\|_2=1,\|V_2\|_F=1}\frac{\beta_\phi}{2m}\left(\|F^*R^{(l_2)}F\|_2^2\|\boldsymbol{v}_1\|_2^2 + \|F^*V_2 F\|_2^2\|\boldsymbol{\alpha}^{(l_2-1)}\|_2^2\right) \tag{70}$$

$$\overset{(a)}{\leq} \frac{\beta_\phi}{2m}\left(\|R^{(l_2)}\|_2^2 + \|\boldsymbol{\alpha}^{(l_2-1)}\|_2^2\right) \leq \frac{\beta_\phi}{2}\left(\gamma_r^2 + \left(\gamma^{l_2-1} + |\phi(0)|\sum_{i=1}^{l_2-1}\gamma^{i-1}\right)^2\right)$$

where $(a)$ follows, again, by exploiting the isometry of $F^*$ and $F$ with respect to the $L_2$ norm, and using $\|V_2\|_2 \leq \|V_2\|_F$. Finally,

$$\left\|T_{3_{i,jj',k}}\right\|_{2,2,1} = \sup_{\|\boldsymbol{v}_1\|_2=1,\|V_2\|_F=1} \sum_{i=1}^{m} \left|\frac{\phi'}{\sqrt{m}}F_{ij}^*V_{2_{jj'}}F_{j'k}v_{1_k}\right|$$

$$\leq \frac{1}{\sqrt{m}}\sup_{\|\boldsymbol{v}_1\|_2=1,\|V_2\|_F=1}\sum_{i=1}^{m}|(F^*V_2 F\boldsymbol{v}_1)_i|$$

$$\leq \sup_{\|\boldsymbol{v}_1\|_2=1,\|V_2\|_F=1}\|F^*V_2 F\boldsymbol{v}_1\|_2 \tag{71}$$

$$\leq \sup_{\|\boldsymbol{v}_1\|_2=1,\|V_2\|_F=1}\|V_2\|_2\|\boldsymbol{v}_1\|_2 = 1 .$$

Combining (69), (70) and (71), we get

$$\left\|\frac{\partial^2\boldsymbol{\alpha}^{(l_2)}}{\partial\boldsymbol{\alpha}^{(l_2-1)}\partial\mathbf{r}^{(l_2)}}\right\|_{2,2,1} \leq \frac{\beta_\phi}{2}\left(\gamma_w^2 + \gamma_r^2\right) + \beta_\phi\left(\gamma^{l_2-1} + |\phi(0)|\sum_{i=1}^{l_2-1}\gamma^{i-1}\right)^2 + 1$$

$$\leq \beta_\phi\left(\gamma^2 + \left(\gamma^{l_2-1} + |\phi(0)|\sum_{i=1}^{l_2-1}\gamma^{i-1}\right)^2\right) + 1 , \tag{72}$$

and finally we can upper bound the last quantity above as in (60) and complete the proof.

For proving (59) consider the following

$$\left[\frac{\partial^2\boldsymbol{\alpha}^{(l_2)}}{\partial\boldsymbol{\alpha}^{(l_2-1)}\partial\boldsymbol{w}^{(l_2)}}\right]_{i,jj',k} = \left(\underbrace{\frac{\phi''(\widetilde{\boldsymbol{\alpha}}_i^{(l_2)})}{m}W_{ik}^{(l_2)}\boldsymbol{\alpha}_{j'}^{(l_2-1)}\delta_{ij}}_{T_1} + \underbrace{\frac{\phi''(\widetilde{\boldsymbol{\alpha}}_i^{(l_2)})}{m}F_{ip}^*R_{pq}^{(l_2)}F_{qk}\boldsymbol{\alpha}_{j'}^{(l_2-1)}\delta_{ij}}_{T_2}\right) + \underbrace{\frac{1}{\sqrt{m}}\phi'(\widetilde{\boldsymbol{\alpha}}^{(l_2)})\delta_{ij}\delta_{kj'}}_{T_3}$$

Then analyzing each term separately, we get

$$
\begin{aligned}
\left\| T_{1_{i,jj',k}} \right\|_{2,2,1} &= \sup_{\|\boldsymbol{v}_1\|_2=1, \|V_2\|_F=1} \sum_{i=1}^m \left| \frac{\phi''}{m} W_{ik}^{(l_2)} v_{1_k} V_{2_{ij'}} \boldsymbol{\alpha}_{j'}^{(l_2-1)} \right| \\
&\leq \sup_{\|\boldsymbol{v}_1\|_2=1, \|V_2\|_F=1} \frac{\beta_\phi}{2m} \left( \|W^{(l_2)}\|_2^2 \|\boldsymbol{v}_1\|_2^2 + \|V_2\|_2^2 \|\boldsymbol{\alpha}^{(l_2-1)}\|_2^2 \right) \\
&= \frac{\beta_\phi}{2m} \left( \|W^{(l_2)}\|_2^2 + \|\boldsymbol{\alpha}^{(l_2-1)}\|_2^2 \right) \leq \frac{\beta_\phi}{2} \left( \gamma_w^2 + \left( \gamma^{l_1-1} + |\phi(0)| \sum_{i=1}^{l_1-1} \gamma^{i-1} \right)^2 \right),
\end{aligned}
\tag{73}
$$

$$
\begin{aligned}
\left\| T_{2_{i,jj',k}} \right\|_{2,2,1} &= \sup_{\|\boldsymbol{v}_1\|_2=1, \|V_2\|_F=1} \sum_{i=1}^m \left| \frac{\phi''}{m} F_{ip}^* R_{pq}^{(l_2)} F_{qk} v_{1_k} V_{2_{ij'}} \boldsymbol{\alpha}_{j'}^{(l_2-1)} \right| \\
&\leq \sup_{\|\boldsymbol{v}_1\|_2=1, \|V_2\|_F=1} \frac{\beta_\phi}{2m} \left( \|F^* R^{(l_2)} F \boldsymbol{v}_1\|_2^2 + \|V_2 \boldsymbol{\alpha}^{(l_2-1)}\|_2^2 \right) \\
&\leq \sup_{\|\boldsymbol{v}_1\|_2=1, \|V_2\|_F=1} \frac{\beta_\phi}{2m} \left( \|F^* R^{(l_2)} F\|_2^2 \|\boldsymbol{v}_1\|_2^2 + \|V_2\|_2^2 \|\boldsymbol{\alpha}^{(l_2-1)}\|_2^2 \right) \\
&= \frac{\beta_\phi}{2m} \left( \|F^* R^{(l_2)} F\|_2^2 + \|\boldsymbol{\alpha}^{(l_2-1)}\|_2^2 \right) \\
&\leq \frac{\beta_\phi}{2} \left( \gamma_r^2 + \left( \gamma^{l_1-1} + |\phi(0)| \sum_{i=1}^{l_1-1} \gamma^{i-1} \right)^2 \right),
\end{aligned}
\tag{74}
$$

and, finally,

$$
\begin{aligned}
\left\| T_{3_{i,jj'k}} \right\|_{2,2,1} &= \sup_{\|\boldsymbol{v}_1\|_2=1, \|V_2\|_F=1} \sum_{i=1}^m \left| \frac{\phi'}{\sqrt{m}} V_{2ik} v_{1_k} \right| \\
&\leq \sup_{\|\boldsymbol{v}_1\|_2=1, \|V_2\|_F=1} \sum_{i=1}^m \frac{1}{\sqrt{m}} \|\boldsymbol{v}_1\|_2 \|V_{2,i,:}\|_2 \\
&\leq \sup_{\|V_2\|_F=1} \sqrt{\sum_{i=1}^m \|V_{2,i,:}\|_2^2} \\
&= 1.
\end{aligned}
\tag{75}
$$

Hence, we have

$$
\begin{aligned}
\left\| \frac{\partial^2 \boldsymbol{\alpha}^{(l_2)}}{\partial \boldsymbol{\alpha}^{(l_2-1)} \partial \mathbf{w}^{(l_2)}} \right\|_{2,2,1}^2 &\leq \frac{\beta_\phi}{2} \left( \gamma_w^2 + \gamma_r^2 \right) + \beta_\phi \left( \gamma^{l_1-1} + |\phi(0)| \sum_{i=1}^{l_1-1} \gamma^{i-1} \right)^2 + 1 \\
&\leq \beta_\phi \left( \gamma^2 + \left( \gamma^{l_1-1} + |\phi(0)| \sum_{i=1}^{l_1-1} \gamma^{i-1} \right)^2 \right) + 1,
\end{aligned}
\tag{76}
$$

and finally we can upper bound the last quantity above as in (59).

We now focus on proving (56). Note that from (39) we have

$$
\left[ \frac{\partial^2 \boldsymbol{\alpha}^{(l_1)}}{(\partial \mathbf{w}^{(l_1)})^2} \right]_{i,jj',kk'} = \frac{1}{m} \phi'' \left( \tilde{\boldsymbol{\alpha}}^{(l_1)} \right) \cdot \boldsymbol{\alpha}_{j'}^{(l_1-1)} \boldsymbol{\alpha}_{k'}^{(l_1-1)} \delta_{ij} \delta_{ik}.
$$

Now,

$$
\left\| \frac{\partial^2 \boldsymbol{\alpha}^{(l_1)}}{(\partial \mathbf{w}^{(l_1)})^2} \right\|_{2,2,1}
$$

$$
= \sup_{\|V_1\|_F=1, \|V_2\|_F=1} \sum_{i=1}^{m} \left| \frac{1}{m} \phi''(\tilde{\boldsymbol{\alpha}}_i^{(l_1)}) V_{1_{ij'}} \boldsymbol{\alpha}_{j'}^{(l_1-1)} V_{2_{ik'}} \boldsymbol{\alpha}_{k'}^{(l_1-1)} \right|
$$

$$
\leq \sup_{\|V_1\|_F=1, \|V_2\|_F=1} \frac{\beta_\phi}{m} \sum_{i=1}^{m} \left| (V_1 \boldsymbol{\alpha}^{(l_1-1)})_i (V_2 \boldsymbol{\alpha}^{(l_1-1)})_i \right| \tag{77}
$$

$$
\leq \sup_{\|V_1\|_F=1, \|V_2\|_F=1} \frac{\beta_\phi}{2m} \left( \left\| V_1 \boldsymbol{\alpha}^{(l_1-1)} \right\|_2^2 + \left\| V_2 \boldsymbol{\alpha}^{(l_1-1)} \right\|_2^2 \right)
$$

$$
\leq \frac{\beta_\phi}{m} \left\| \boldsymbol{\alpha}^{(l_1-1)} \right\|_2^2 \leq \beta_\phi \left( \gamma^{l_1-1} + |\phi(0)| \sum_{i=1}^{l_1-1} \gamma^{i-1} \right)^2.
$$

Finally, we can upper bound the last quantity above as in (56) and complete the proof.

Now, the last result to prove is (61). Note that from (39) we have

$$
\left[ \frac{\partial^2 \boldsymbol{\alpha}^{(l_2)}}{(\partial \mathbf{r}^{(l_2)})^2} \right]_{i,jj',kk'} = \frac{1}{m} \phi'' \left( \tilde{\boldsymbol{\alpha}}^{(l_2)} \right) \cdot F_{ij}^* F_{j'p} \boldsymbol{\alpha}_p^{(l_2-1)} F_{ik}^* F_{k'q} \boldsymbol{\alpha}_q^{(l_2-1)},
$$

where we make use of the Einstein notation. Now,

$$
\left\| \frac{\partial^2 \boldsymbol{\alpha}^{(l_2)}}{(\partial \mathbf{r}^{(l_2)})^2} \right\|_{2,2,1}
$$

$$
= \sup_{\|V_1\|_F=1, \|V_2\|_F=1} \sum_{i=1}^{m} \left| \frac{1}{m} \phi''(\tilde{\boldsymbol{\alpha}}_i^{(l_2)}) F_{ij}^* V_{1_{jj'}} F_{j'p} \boldsymbol{\alpha}_p^{(l_2-1)} F_{ik}^* V_{2_{kk'}} F_{k'q} \boldsymbol{\alpha}_q^{(l_2-1)} \right|
$$

$$
\leq \sup_{\|V_1\|_F=1, \|V_2\|_F=1} \frac{\beta_\phi}{m} \sum_{i=1}^{m} \left| ((F^* V_1 F) \boldsymbol{\alpha}^{(l_2-1)})_i ((F^* V_2 F) \boldsymbol{\alpha}^{(l_2-1)})_i \right| \tag{78}
$$

$$
\leq \sup_{\|V_1\|_F=1, \|V_2\|_F=1} \frac{\beta_\phi}{2m} \left( \left\| F^* V_1 F \boldsymbol{\alpha}^{(l_2-1)} \right\|_2^2 + \left\| F^* V_2 F \boldsymbol{\alpha}^{(l_2-1)} \right\|_2^2 \right)
$$

$$
\leq \frac{\beta_\phi}{m} \left\| \boldsymbol{\alpha}^{(l_2-1)} \right\|_2^2 \leq \beta_\phi \left( \gamma^{l_2-1} + |\phi(0)| \sum_{i=1}^{l_2-1} \gamma^{i-1} \right)^2.
$$

Finally, we can upper bound the last quantity above as in (61) and complete the proof. □

Now we upper bound the terms in equation (54). Thus, we obtain that, with probability at least $1 - \frac{2(L+2)}{m}$,

$$
\mathcal{Q}_\infty(f) = \max_{l \in [L+1]} \left\| \frac{\partial f}{\partial \boldsymbol{\alpha}^{(l)}} \right\|_\infty \overset{(a)}{\leq} \max_{l \in [L+1]} \frac{1}{\sqrt{m}} \gamma^{L+1-l} (1 + \rho_1) \leq \frac{1}{\sqrt{m}} (1 + \gamma^L)(1 + \rho_1), \tag{79}
$$

$$
\mathcal{Q}_2^{(w,r)}(f) \overset{(b)}{\leq} \max_{l \in [L+1]} (\gamma^{l-1} + |\phi(0)| \sum_{i=1}^{l-1} \gamma^{i-1}) \leq (1 + \gamma^L)(1 + L|\phi(0)|), \tag{80}
$$

and

$$
\mathcal{Q}_{2,2,1}^{(w,r)}(f), \mathcal{Q}_{2,2,1}^{(w)}(f), \mathcal{Q}_{2,2,1}^{(r)}(f) \overset{(c)}{\leq} 2\beta_\phi (1 + \gamma^L)^2 (1 + \gamma)^2 (1 + (1 + L|\phi(0)|)^2) + 1, \tag{81}
$$

where (a) follows from a direct adaptation of the results from Section A.6 in (Banerjee et al., 2023a), (b) follows from Lemma 8, and (c) follows from Lemma 9.

We now proceed to analyze the Hessian. We also recall that $\max_{l \in \{2,...,L+1\}} \left\| \frac{\partial \boldsymbol{\alpha}^{(l)}}{\partial \boldsymbol{\alpha}^{(l-1)}} \right\|_2 \leq \gamma$ from Lemma 7.

We introduce some notation. Given an order-3 tensor $T \in \mathbb{R}^{d_1 \times d_2 \times d_3}$, we have that its first dimension has $d_1$ entries, the second has $d_2$ entries, and the third has $d_3$ entries. Consider the matrices $X \in \mathbb{R}^{k_1 \times d_1}$, $Y \in \mathbb{R}^{k_2 \times d_2}$, and $Z \in \mathbb{R}^{k_3 \times d_3}$. We use the notation $(X)(Y)T(Z) \in \mathbb{R}^{k_1 \times k_2 \times k_3}$ to denote $X$ multiplying $A$ long its first dimension, $Y$ along its second dimension, and $Z$ along its third dimension. We use the notation $(X)T(Z) \in \mathbb{R}^{k_1 \times d_2 \times k_3}$ to denote $X$ multiplying $A$ long its first dimension and $Z$ along its third dimension.

**Off-Diagonal Blocks.** For the off-diagonal blocks, we focus on bounding $\|H_{w,r}^{(l_1,l_2)}\|_2$ for (Case 1.A) $l_1 \le l_2$, (Case 1.B) $l_2 \le l_1$. Further, we bound (Case 2.A) $\|H_{v,w}^{(l_1)}\|_2$ and (Case 2.B) $\|H_{v,r}^{(l_2)}\|_2$.

**Case 1.A:** $2 \le l_1 \le l_2 \le L+1$. By building on the form of the gradient, we have

$$H_{w,r}^{(l_1,l_2)} = \frac{\partial^2 \boldsymbol{\alpha}^{(l_1)}}{\partial \mathbf{w}^{(l_1)} \partial \mathbf{r}^{(l_1)}} \frac{\partial f}{\partial \boldsymbol{\alpha}^{(l_1)}} \mathbf{1}_{[l_1=l_2]} + \mathbf{1}_{[l_1<l_2]} \left( \frac{\partial \boldsymbol{\alpha}^{(l_1)}}{\partial \mathbf{w}^{(l_1)}} \prod_{l'=l_1+1}^{l_2-1} \frac{\partial \boldsymbol{\alpha}^{(l')}}{\partial \boldsymbol{\alpha}^{(l'-1)}} \right) \frac{\partial^2 \boldsymbol{\alpha}^{(l_2)}}{\partial \boldsymbol{\alpha}^{(l_2-1)} \partial \mathbf{r}^{(l_2)}} \left( \frac{\partial f}{\partial \boldsymbol{\alpha}^{(l_2)}} \right)$$

$$+ \mathbf{1}_{[l_1<l_2]} \sum_{l=l_2+1}^{L+1} \left( \frac{\partial \boldsymbol{\alpha}^{(l_1)}}{\partial \mathbf{w}^{(l_1)}} \prod_{l'=l_1+1}^{l-1} \frac{\partial \boldsymbol{\alpha}^{(l')}}{\partial \boldsymbol{\alpha}^{(l'-1)}} \right) \left( \frac{\partial \boldsymbol{\alpha}^{(l_2)}}{\partial \mathbf{r}^{(l_2)}} \prod_{l'=l_2+1}^{l-1} \frac{\partial \boldsymbol{\alpha}^{(l')}}{\partial \boldsymbol{\alpha}^{(l'-1)}} \right) \frac{\partial^2 \boldsymbol{\alpha}^{(l)}}{(\partial \boldsymbol{\alpha}^{(l-1)})^2} \left( \frac{\partial f}{\partial \boldsymbol{\alpha}^{(l)}} \right).$$

Then,

$$\|H_{w,r}^{(l_1,l_2)}\|_2 \le \left\| \frac{\partial^2 \boldsymbol{\alpha}^{(l_1)}}{\partial \mathbf{w}^{(l_1)} \partial \mathbf{r}^{(l_1)}} \right\|_{2,2,1} \left\| \frac{\partial f}{\partial \boldsymbol{\alpha}^{(l_1)}} \right\|_\infty \mathbf{1}_{[l_1=l_2]}$$

$$+ \mathbf{1}_{[l_1<l_2]} \left\| \frac{\partial \boldsymbol{\alpha}^{(l_1)}}{\partial \mathbf{w}^{(l_1)}} \right\|_2 \prod_{l'=l_1+1}^{l_2-1} \left\| \frac{\partial \boldsymbol{\alpha}^{(l')}}{\partial \boldsymbol{\alpha}^{(l'-1)}} \right\|_2 \left\| \frac{\partial^2 \boldsymbol{\alpha}^{(l_2)}}{\partial \boldsymbol{\alpha}^{(l_2-1)} \partial \mathbf{r}^{(l_2)}} \right\|_{2,2,1} \left\| \frac{\partial f}{\partial \boldsymbol{\alpha}^{(l_2)}} \right\|_\infty$$

$$+ \mathbf{1}_{[l_1<l_2]} \sum_{l=l_2+1}^{L+1} \left( \left\| \frac{\partial \boldsymbol{\alpha}^{(l_1)}}{\partial \mathbf{w}^{(l_1)}} \right\|_2 \prod_{l'=l_1+1}^{l-1} \left\| \frac{\partial \boldsymbol{\alpha}^{(l')}}{\partial \boldsymbol{\alpha}^{(l'-1)}} \right\|_2 \right) \left( \left\| \frac{\partial \boldsymbol{\alpha}^{(l_2)}}{\partial \mathbf{r}^{(l_2)}} \right\|_2 \prod_{l'=l_2+1}^{l-1} \left\| \frac{\partial \boldsymbol{\alpha}^{(l')}}{\partial \boldsymbol{\alpha}^{(l'-1)}} \right\|_2 \right)$$

$$\times \left\| \frac{\partial^2 \boldsymbol{\alpha}^{(l)}}{(\partial \boldsymbol{\alpha}^{(l-1)})^2} \right\|_{2,2,1} \left\| \frac{\partial f}{\partial \boldsymbol{\alpha}^{(l)}} \right\|_\infty$$

$$\le \left\| \frac{\partial^2 \boldsymbol{\alpha}^{(l_1)}}{\partial \mathbf{w}^{(l_1)} \partial \mathbf{r}^{(l_1)}} \right\|_{2,2,1} \left\| \frac{\partial f}{\partial \boldsymbol{\alpha}^{(l_1)}} \right\|_\infty \mathbf{1}_{[l_1=l_2]}$$

$$+ \mathbf{1}_{[l_1<l_2]} \gamma^{l_2-l_1-1} \left\| \frac{\partial \boldsymbol{\alpha}^{(l_1)}}{\partial \mathbf{w}^{(l_1)}} \right\|_2 \left\| \frac{\partial^2 \boldsymbol{\alpha}^{(l_2)}}{\partial \boldsymbol{\alpha}^{(l_2-1)} \partial \mathbf{r}^{(l_2)}} \right\|_{2,2,1} \left\| \frac{\partial f}{\partial \boldsymbol{\alpha}^{(l_2)}} \right\|_\infty$$

$$+ \mathbf{1}_{[l_1<l_2]} \sum_{l=l_2+1}^{L+1} \gamma^{2l-l_2-l_1-2} \left\| \frac{\partial \boldsymbol{\alpha}^{(l_1)}}{\partial \mathbf{w}^{(l_1)}} \right\|_2 \left\| \frac{\partial \boldsymbol{\alpha}^{(l_2)}}{\partial \mathbf{r}^{(l_2)}} \right\|_2 \left\| \frac{\partial^2 \boldsymbol{\alpha}^{(l)}}{(\partial \boldsymbol{\alpha}^{(l-1)})^2} \right\|_{2,2,1} \left\| \frac{\partial f}{\partial \boldsymbol{\alpha}^{(l)}} \right\|_\infty.$$

Then, based on the definitions in (54), we have

$$\|H_{w,r}^{(l_1,l_2)}\|_2 \le (L+1)(1+\gamma^{2L}) \mathcal{Q}_{2,2,1}^{(w,r)}(f) \mathcal{Q}_\infty(f)$$

$$\overset{(a)}{\le} \frac{(L+1)(1+\rho_1)}{\sqrt{m}} (1+\gamma^{2L})^2 (2\beta_\phi(1+\gamma^L)^2(1+\gamma)^2(1+(1+L|\phi(0)|)^2)+1),$$

where for (a) we used equations (79) and (81).

**Case 1.B:** $2 \le l_2 \le l_1 \le L+1$. By building on the form of the gradient, we have

$$H_{w,r}^{(l_1,l_2)} = \frac{\partial^2 \boldsymbol{\alpha}^{(l_2)}}{\partial \mathbf{w}^{(l_2)} \partial \mathbf{r}^{(l_2)}} \frac{\partial f}{\partial \boldsymbol{\alpha}^{(l_2)}} \mathbf{1}_{[l_1=l_2]} + \mathbf{1}_{[l_2<l_1]} \left( \frac{\partial \boldsymbol{\alpha}^{(l_2)}}{\partial \mathbf{r}^{(l_2)}} \prod_{l'=l_2+1}^{l_1-1} \frac{\partial \boldsymbol{\alpha}^{(l')}}{\partial \boldsymbol{\alpha}^{(l'-1)}} \right) \frac{\partial^2 \boldsymbol{\alpha}^{(l_1)}}{\partial \boldsymbol{\alpha}^{(l_1-1)} \partial \mathbf{w}^{(l_1)}} \left( \frac{\partial f}{\partial \boldsymbol{\alpha}^{(l_1)}} \right)$$

$$+ \mathbf{1}_{[l_2<l_1]} \sum_{l=l_1+1}^{L+1} \left( \frac{\partial \boldsymbol{\alpha}^{(l_2)}}{\partial \mathbf{r}^{(l_2)}} \prod_{l'=l_2+1}^{l-1} \frac{\partial \boldsymbol{\alpha}^{(l')}}{\partial \boldsymbol{\alpha}^{(l'-1)}} \right) \left( \frac{\partial \boldsymbol{\alpha}^{(l_1)}}{\partial \mathbf{w}^{(l_1)}} \prod_{l'=l_1+1}^{l-1} \frac{\partial \boldsymbol{\alpha}^{(l')}}{\partial \boldsymbol{\alpha}^{(l'-1)}} \right) \frac{\partial^2 \boldsymbol{\alpha}^{(l)}}{(\partial \boldsymbol{\alpha}^{(l-1)})^2} \left( \frac{\partial f}{\partial \boldsymbol{\alpha}^{(l)}} \right).$$

Then,

$$
\begin{aligned}
\|H_{w,r}^{(l_1,l_2)}\|_2 &\leq \left\|\frac{\partial^2\boldsymbol{\alpha}^{(l_2)}}{\partial\mathbf{w}^{(l_2)}\partial\mathbf{r}^{(l_2)}}\right\|_{2,2,1}\left\|\frac{\partial f}{\partial\boldsymbol{\alpha}^{(l_2)}}\right\|_\infty \mathbf{1}_{[l_1=l_2]} \\
&+ \mathbf{1}_{[l_2<l_1]}\left\|\frac{\partial\boldsymbol{\alpha}^{(l_2)}}{\partial\mathbf{r}^{(l_2)}}\right\|_2\prod_{l'=l_2+1}^{l_1-1}\left\|\frac{\partial\boldsymbol{\alpha}^{(l')}}{\partial\boldsymbol{\alpha}^{(l'-1)}}\right\|_2\left\|\frac{\partial^2\boldsymbol{\alpha}^{(l_1)}}{\partial\boldsymbol{\alpha}^{(l_1-1)}\partial\mathbf{w}^{(l_1)}}\right\|_{2,2,1}\left\|\frac{\partial f}{\partial\boldsymbol{\alpha}^{(l_1)}}\right\|_\infty \\
&+ \mathbf{1}_{[l_2<l_1]}\sum_{l=l_1+1}^{L+1}\left(\left\|\frac{\partial\boldsymbol{\alpha}^{(l_2)}}{\partial\mathbf{r}^{(l_2)}}\right\|_2\prod_{l'=l_2+1}^{l-1}\left\|\frac{\partial\boldsymbol{\alpha}^{(l')}}{\partial\boldsymbol{\alpha}^{(l'-1)}}\right\|_2\right)\left(\left\|\frac{\partial\boldsymbol{\alpha}^{(l_1)}}{\partial\mathbf{w}^{(l_1)}}\right\|_2\prod_{l'=l_1+1}^{l-1}\left\|\frac{\partial\boldsymbol{\alpha}^{(l')}}{\partial\boldsymbol{\alpha}^{(l'-1)}}\right\|_2\right) \\
&\times \left\|\frac{\partial^2\boldsymbol{\alpha}^{(l)}}{(\partial\boldsymbol{\alpha}^{(l-1)})^2}\right\|_{2,2,1}\left\|\frac{\partial f}{\partial\boldsymbol{\alpha}^{(l)}}\right\|_\infty \\
&\leq \left\|\frac{\partial^2\boldsymbol{\alpha}^{(l_2)}}{\partial\mathbf{w}^{(l_2)}\partial\mathbf{r}^{(l_2)}}\right\|_{2,2,1}\left\|\frac{\partial f}{\partial\boldsymbol{\alpha}^{(l_2)}}\right\|_\infty \mathbf{1}_{[l_1=l_2]} \\
&+ \mathbf{1}_{[l_2<l_1]}\gamma^{l_1-l_2-1}\left\|\frac{\partial\boldsymbol{\alpha}^{(l_2)}}{\partial\mathbf{r}^{(l_2)}}\right\|_2\left\|\frac{\partial^2\boldsymbol{\alpha}^{(l_1)}}{\partial\boldsymbol{\alpha}^{(l_1-1)}\partial\mathbf{w}^{(l_1)}}\right\|_{2,2,1}\left\|\frac{\partial f}{\partial\boldsymbol{\alpha}^{(l_1)}}\right\|_\infty \\
&+ \mathbf{1}_{[l_2<l_1]}\sum_{l=l_1+1}^{L+1}\gamma^{2l-l_1-l_2-2}\left\|\frac{\partial\boldsymbol{\alpha}^{(l_2)}}{\partial\mathbf{r}^{(l_2)}}\right\|_2\left\|\frac{\partial\boldsymbol{\alpha}^{(l_1)}}{\partial\mathbf{w}^{(l_1)}}\right\|_2\left\|\frac{\partial^2\boldsymbol{\alpha}^{(l)}}{(\partial\boldsymbol{\alpha}^{(l-1)})^2}\right\|_{2,2,1}\left\|\frac{\partial f}{\partial\boldsymbol{\alpha}^{(l)}}\right\|_\infty.
\end{aligned}
$$

Then, the upper bound is similar to the case **Case 1.A**,

$$
\begin{aligned}
\|H_{w,r}^{(l_1,l_2)}\|_2 &\leq (L+1)(1+\gamma^{2L})\mathcal{Q}_{2,2,1}^{(w,r)}(f)\mathcal{Q}_\infty(f) \\
&\overset{(a)}{\leq} \frac{(L+1)(1+\rho_1)}{\sqrt{m}}(1+\gamma^{2L})^2(2\beta_\phi(1+\gamma^L)^2(1+\gamma)^2(1+(1+L|\phi(0)|)^2)+1).
\end{aligned}
$$

**Case 2.A:** $1 \leq l_1 \leq L+1$. For Hessian terms involving $(w,v)$, since $\frac{\partial f}{\partial\mathbf{v}} = \frac{1}{\sqrt{m}}\boldsymbol{\alpha}^{(L+1)}$, we have

$$
H_{w,v}^{(l_1)} = \frac{1}{\sqrt{m}}\frac{\partial\boldsymbol{\alpha}^{(L+1)}}{\partial\mathbf{w}^{(l_1)}} = \frac{1}{\sqrt{m}}\left(\frac{\partial\boldsymbol{\alpha}^{(l_1)}}{\partial\mathbf{w}^{(l_1)}}\prod_{l'=l_1+1}^{L+1}\frac{\partial\boldsymbol{\alpha}^{(l')}}{\partial\boldsymbol{\alpha}^{(l'-1)}}\right).
$$

Then,

$$
\|H_{w,v}^{(l_1,L+1)}\|_2 \leq \frac{1}{\sqrt{m}}\left\|\frac{\partial\boldsymbol{\alpha}^{(l_1)}}{\partial\mathbf{w}^{(l_1)}}\right\|_2\prod_{l'=l_1+1}^{L+1}\left\|\frac{\partial\boldsymbol{\alpha}^{(l')}}{\partial\boldsymbol{\alpha}^{(l'-1)}}\right\|_2 \leq \frac{1}{\sqrt{m}}\gamma^L\mathcal{Q}_2^{(w,r)}(f) \overset{(a)}{\leq} \frac{1}{\sqrt{m}}\gamma^L(1+\gamma^L)(1+L|\phi(0)|),
$$

where (a) follows from equation (80).

**Case 2.B:** $2 \leq l_2 \leq L+1$. For Hessian terms involving $(r,v)$, since $\frac{\partial f}{\partial\mathbf{v}} = \frac{1}{\sqrt{m}}\boldsymbol{\alpha}^{(L+1)}$, we have

$$
H_{r,v}^{(l_2)} = \frac{1}{\sqrt{m}}\frac{\partial\boldsymbol{\alpha}^{(L+1)}}{\partial\mathbf{r}^{(l_2)}} = \frac{1}{\sqrt{m}}\left(\frac{\partial\boldsymbol{\alpha}^{(l_2)}}{\partial\mathbf{r}^{(l_2)}}\prod_{l'=l_2+1}^{L+1}\frac{\partial\boldsymbol{\alpha}^{(l')}}{\partial\boldsymbol{\alpha}^{(l'-1)}}\right).
$$

Then,

$$
\|H_{r,v}^{(l_2)}\|_2 \leq \frac{1}{\sqrt{m}}\left\|\frac{\partial\boldsymbol{\alpha}^{(l_2)}}{\partial\mathbf{r}^{(l_2)}}\right\|_2\prod_{l'=l_2+1}^{L+1}\left\|\frac{\partial\boldsymbol{\alpha}^{(l')}}{\partial\boldsymbol{\alpha}^{(l'-1)}}\right\|_2 \overset{(a)}{\leq} \frac{1}{\sqrt{m}}\gamma^L(1+\gamma^L)(1+L|\phi(0)|),
$$

where (a) follows from equation (80).

**Diagonal Blocks.** For the diagonal blocks, we focus only on bounding (Case 3.A) $\|H_w^{(l_1,l_2)}\|_2$ and (Case 3.B) $\|H_r^{(l_1,l_2)}\|_2$ for $l_1 \le l_2$, since the case $l_2 \le l_1$ is just symmetrical and will have the same bounds.

**Case 3.A:** $1 \le l_1 \le l_2 \le L+1$. By building on the form of the gradient, we have

$$
H_w^{(l_1,l_2)} = \frac{\partial^2 \boldsymbol{\alpha}^{(l_1)}}{(\partial \mathbf{w}^{(l_1)})^2} \frac{\partial f}{\partial \boldsymbol{\alpha}^{(l_1)}} \mathbf{1}_{[l_1=l_2]} + \mathbf{1}_{[l_1<l_2]} \left( \frac{\partial \boldsymbol{\alpha}^{(l_1)}}{\partial \mathbf{w}^{(l_1)}} \prod_{l'=l_1+1}^{l_2-1} \frac{\partial \boldsymbol{\alpha}^{(l')}}{\partial \boldsymbol{\alpha}^{(l'-1)}} \right) \frac{\partial^2 \boldsymbol{\alpha}^{(l_2)}}{\partial \boldsymbol{\alpha}^{(l_2-1)} \partial \mathbf{w}^{(l_2)}} \left( \frac{\partial f}{\partial \boldsymbol{\alpha}^{(l_2)}} \right)
$$
$$
+ \mathbf{1}_{[l_1<l_2]} \sum_{l=l_2+1}^{L+1} \left( \frac{\partial \boldsymbol{\alpha}^{(l_1)}}{\partial \mathbf{w}^{(l_1)}} \prod_{l'=l_1+1}^{l-1} \frac{\partial \boldsymbol{\alpha}^{(l')}}{\partial \boldsymbol{\alpha}^{(l'-1)}} \right) \left( \frac{\partial \boldsymbol{\alpha}^{(l_2)}}{\partial \mathbf{w}^{(l_2)}} \prod_{l'=l_2+1}^{l-1} \frac{\partial \boldsymbol{\alpha}^{(l')}}{\partial \boldsymbol{\alpha}^{(l'-1)}} \right) \frac{\partial^2 \boldsymbol{\alpha}^{(l)}}{(\partial \boldsymbol{\alpha}^{(l-1)})^2} \left( \frac{\partial f}{\partial \boldsymbol{\alpha}^{(l)}} \right) .
$$

Then,

$$
\|H_w^{(l_1,l_2)}\|_2 \le \left\| \frac{\partial^2 \boldsymbol{\alpha}^{(l_1)}}{(\partial \mathbf{w}^{(l_1)})^2} \right\|_{2,2,1} \left\| \frac{\partial f}{\partial \boldsymbol{\alpha}^{(l_1)}} \right\|_\infty \mathbf{1}_{[l_1=l_2]}
$$
$$
+ \mathbf{1}_{[l_1<l_2]} \left\| \frac{\partial \boldsymbol{\alpha}^{(l_1)}}{\partial \mathbf{w}^{(l_1)}} \right\|_2 \prod_{l'=l_1+1}^{l_2-1} \left\| \frac{\partial \boldsymbol{\alpha}^{(l')}}{\partial \boldsymbol{\alpha}^{(l'-1)}} \right\|_2 \left\| \frac{\partial^2 \boldsymbol{\alpha}^{(l_2)}}{\partial \boldsymbol{\alpha}^{(l_2-1)} \partial \mathbf{w}^{(l_2)}} \right\|_{2,2,1} \left\| \frac{\partial f}{\partial \boldsymbol{\alpha}^{(l_2)}} \right\|_\infty
$$
$$
+ \mathbf{1}_{[l_1<l_2]} \sum_{l=l_2+1}^{L+1} \left( \left\| \frac{\partial \boldsymbol{\alpha}^{(l_1)}}{\partial \mathbf{w}^{(l_1)}} \right\|_2 \prod_{l'=l_1+1}^{l-1} \left\| \frac{\partial \boldsymbol{\alpha}^{(l')}}{\partial \boldsymbol{\alpha}^{(l'-1)}} \right\|_2 \right) \left( \left\| \frac{\partial \boldsymbol{\alpha}^{(l_2)}}{\partial \mathbf{w}^{(l_2)}} \right\|_2 \prod_{l'=l_2+1}^{l-1} \left\| \frac{\partial \boldsymbol{\alpha}^{(l')}}{\partial \boldsymbol{\alpha}^{(l'-1)}} \right\|_2 \right)
$$
$$
\times \left\| \frac{\partial^2 \boldsymbol{\alpha}^{(l)}}{(\partial \boldsymbol{\alpha}^{(l-1)})^2} \right\|_{2,2,1} \left\| \frac{\partial f}{\partial \boldsymbol{\alpha}^{(l)}} \right\|_\infty
$$
$$
\le \left\| \frac{\partial^2 \boldsymbol{\alpha}^{(l_1)}}{(\partial \mathbf{w}^{(l_1)})^2} \right\|_{2,2,1} \left\| \frac{\partial f}{\partial \boldsymbol{\alpha}^{(l_1)}} \right\|_\infty \mathbf{1}_{[l_1=l_2]}
$$
$$
+ \mathbf{1}_{[l_1<l_2]} \gamma^{l_2-l_1-1} \left\| \frac{\partial \boldsymbol{\alpha}^{(l_1)}}{\partial \mathbf{w}^{(l_1)}} \right\|_2 \left\| \frac{\partial^2 \boldsymbol{\alpha}^{(l_2)}}{\partial \boldsymbol{\alpha}^{(l_2-1)} \partial \mathbf{w}^{(l_2)}} \right\|_{2,2,1} \left\| \frac{\partial f}{\partial \boldsymbol{\alpha}^{(l_2)}} \right\|_\infty
$$
$$
+ \mathbf{1}_{[l_1<l_2]} \sum_{l=l_2+1}^{L+1} \gamma^{2l-l_2-l_1-2} \left\| \frac{\partial \boldsymbol{\alpha}^{(l_1)}}{\partial \mathbf{w}^{(l_1)}} \right\|_2 \left\| \frac{\partial \boldsymbol{\alpha}^{(l_2)}}{\partial \mathbf{w}^{(l_2)}} \right\|_2 \left\| \frac{\partial^2 \boldsymbol{\alpha}^{(l)}}{(\partial \boldsymbol{\alpha}^{(l-1)})^2} \right\|_{2,2,1} \left\| \frac{\partial f}{\partial \boldsymbol{\alpha}^{(l)}} \right\|_\infty .
$$

Then, based on the definitions in (54), we have

$$
\|H_{(w)}^{(l_1,l_2)}\|_2 \le (L+1)(1+\gamma^{2L}) \mathcal{Q}_{2,2,1}^{(w)}(f) \mathcal{Q}_\infty(f)
$$
$$
\overset{(a)}{\le} \frac{(L+1)(1+\rho_1)}{\sqrt{m}} (1+\gamma^{2L})^2 (2\beta_\phi(1+\gamma^L)^2(1+\gamma)^2(1+(1+L|\phi(0)|)^2)+1) ,
$$

where for (a) we used equations (79) and (81).

**Case 3.B:** $2 \le l_1 \le l_2 \le L+1$. By building on the form of the gradient, we have

$$
H_r^{(l_1,l_2)} = \frac{\partial^2 \boldsymbol{\alpha}^{(l_1)}}{(\partial \mathbf{r}^{(l_1)})^2} \frac{\partial f}{\partial \boldsymbol{\alpha}^{(l_1)}} \mathbf{1}_{[l_1=l_2]} + \mathbf{1}_{[l_1<l_2]} \left( \frac{\partial \boldsymbol{\alpha}^{(l_1)}}{\partial \mathbf{r}^{(l_1)}} \prod_{l'=l_1+1}^{l_2-1} \frac{\partial \boldsymbol{\alpha}^{(l')}}{\partial \boldsymbol{\alpha}^{(l'-1)}} \right) \frac{\partial^2 \boldsymbol{\alpha}^{(l_2)}}{\partial \boldsymbol{\alpha}^{(l_2-1)} \partial \mathbf{r}^{(l_2)}} \left( \frac{\partial f}{\partial \boldsymbol{\alpha}^{(l_2)}} \right)
$$
$$
+ \mathbf{1}_{[l_1<l_2]} \sum_{l=l_2+1}^{L+1} \left( \frac{\partial \boldsymbol{\alpha}^{(l_1)}}{\partial \mathbf{r}^{(l_1)}} \prod_{l'=l_1+1}^{l-1} \frac{\partial \boldsymbol{\alpha}^{(l')}}{\partial \boldsymbol{\alpha}^{(l'-1)}} \right) \left( \frac{\partial \boldsymbol{\alpha}^{(l_2)}}{\partial \mathbf{r}^{(l_2)}} \prod_{l'=l_2+1}^{l-1} \frac{\partial \boldsymbol{\alpha}^{(l')}}{\partial \boldsymbol{\alpha}^{(l'-1)}} \right) \frac{\partial^2 \boldsymbol{\alpha}^{(l)}}{(\partial \boldsymbol{\alpha}^{(l-1)})^2} \left( \frac{\partial f}{\partial \boldsymbol{\alpha}^{(l)}} \right) .
$$

Then, we can obtain prove the following upper bound in a similar way to Case 3.A based on the definitions in (54),

$$
\|H_{(r)}^{(l_1,l_2)}\|_2 \le (L+1)(1+\gamma^{2L}) \mathcal{Q}_{2,2,1}^{(r)}(f) \mathcal{Q}_\infty(f)
$$
$$
\overset{(a)}{\le} \frac{(L+1)(1+\rho_1)}{\sqrt{m}} (1+\gamma^{2L})^2 (2\beta_\phi(1+\gamma^L)^2(1+\gamma)^2(1+(1+L|\phi(0)|)^2)+1) ,
$$

where for (a) we used equations (79) and (81).

Putting all the shown results back in (53), we prove equation (40). We also note that all the constants in the Hessian bound depend on $\sigma_{1,w}, \sigma_{1,r}$, the depth $L$, and the radii $\rho_w, \rho_r, \rho_1$, and $\rho_2$. This dependence of this bound reduces to the depth and the radii and becomes polynomial whenever $\gamma \leq 1$, which is equivalent to $\sigma_{1,w} + \sigma_{1,r} \leq 1 - \frac{\rho_w + \rho_r}{\sqrt{m}}$.

Now, we focus on proving the rest of equations in Lemma 4, namely, equations (41) and (42).

**Gradient and predictor bounds.** We observe that for $l \in [L]$, $\frac{\partial f}{\partial \mathbf{w}^{(l)}} = \frac{\partial \boldsymbol{\alpha}^{(l)}}{\partial \mathbf{w}^{(l)}} \left( \prod_{l'=l}^{L} \frac{\partial \boldsymbol{\alpha}^{(l'+1)}}{\partial \boldsymbol{\alpha}^{(l')}} \right) \frac{\partial f}{\partial \boldsymbol{\alpha}^{(L+1)}}$, and so

$$
\begin{aligned}
\left\| \frac{\partial f}{\partial \mathbf{w}^{(l)}} \right\|_2 &\leq \left\| \frac{\partial \boldsymbol{\alpha}^{(l)}}{\partial \mathbf{w}^{(l)}} \right\|_2 \gamma^{L-l+1} \left\| \frac{\partial f}{\partial \boldsymbol{\alpha}^{(L+1)}} \right\|_2 \\
&\leq \left\| \frac{\partial \boldsymbol{\alpha}^{(l)}}{\partial \mathbf{w}^{(l)}} \right\|_2 \gamma^{L-l+1} \frac{1}{\sqrt{m}} (1 + \rho_1) \\
&\leq (1 + \gamma^L)(1 + L|\phi(0)|)\gamma^L \frac{1}{\sqrt{m}} (1 + \rho_1) \,,
\end{aligned}
$$

where the last inequality follows from Lemma 8.

We also have that

$$
\begin{aligned}
\left\| \frac{\partial f}{\partial \mathbf{w}^{(L+1)}} \right\|_2 &= \left\| \frac{\partial \boldsymbol{\alpha}^{(L+1)}}{\partial \mathbf{w}^{(L+1)}} \frac{\partial f}{\partial \boldsymbol{\alpha}^{(L+1)}} \right\|_2 \\
&= \left\| \frac{\partial \boldsymbol{\alpha}^{(L+1)}}{\partial \mathbf{w}^{(L+1)}} \right\|_2 \left\| \frac{\partial f}{\partial \boldsymbol{\alpha}^{(L+1)}} \right\|_2 \\
&\leq (1 + \gamma^L)(1 + L|\phi(0)|) \frac{1}{\sqrt{m}} (1 + \rho_1) \,.
\end{aligned}
$$

Similarly, we can obtain for $l_2 \in \{2, \ldots, L\}$,

$$
\left\| \frac{\partial f}{\partial \mathbf{r}^{(l_2)}} \right\|_2 \leq (1 + \gamma^L)(1 + L|\phi(0)|)\gamma^L \frac{1}{\sqrt{m}} (1 + \rho_1) \,,
$$

and

$$
\left\| \frac{\partial f}{\partial \mathbf{r}^{(L+1)}} \right\|_2 \leq (1 + \gamma^L)(1 + L|\phi(0)|) \frac{1}{\sqrt{m}} (1 + \rho_1) \,.
$$

Using all these derivations,

$$
\begin{aligned}
\|\nabla_{\boldsymbol{\theta}} f\|_2^2 &= \sum_{l=1}^{L+1} \left\| \frac{\partial f}{\partial \mathbf{w}^{(l)}} \right\|_2^2 + \sum_{l=2}^{L+1} \left\| \frac{\partial f}{\partial \mathbf{r}^{(l)}} \right\|_2^2 \\
&\leq \frac{2}{m} (L+1)(1 + \gamma^L)^2 (1 + L|\phi(0)|)^2 (1 + \rho_1)^2 \,,
\end{aligned}
$$

which finishes the proof for equation (41).

Now,

$$
\begin{aligned}
|f| &= \left| \frac{1}{\sqrt{m}} \mathbf{v}^\top \boldsymbol{\alpha}^{(L+1)} \right| \\
&\leq \frac{1}{\sqrt{m}} \|\mathbf{v}\|_2 \|\boldsymbol{\alpha}^{(L+1)}\|_2 \\
&\leq (1 + \rho_1)(1 + \gamma^L)(1 + L|\phi(0)|) \,,
\end{aligned}
$$

which finishes the proof for equation (42). Again, we notice that all these bounds have a polynomial dependence on the depth $L$, and the radii $\rho_w, \rho_r, \rho_1$, and $\rho_2$ whenever $\gamma \leq 1$, i.e., whenever $\sigma_{1,w} + \sigma_{1,r} \leq 1 - \frac{\rho_w + \rho_r}{\sqrt{m}}$.

Thus, we finish the proof for Lemma 4.

### E.2. RSC and Smoothness Results

Using the results from the previous section, we immediately obtain the RSC and smoothness results.

**Theorem 4** (RSC for FNOs). *Consider Assumptions 5, 6, and 7, and $Q_\kappa^t$ as in Definition 3 with $\|\nabla_{\boldsymbol{\theta}} \bar{G}_t\|_2 \neq 0$. Then, the set $B_\kappa^t := Q_\kappa^t \cap B_{\rho_w, \rho_r \rho_1}^{\mathrm{Euc}}(\boldsymbol{\theta}_0) \cap B_{\rho_2}^{\mathrm{Euc}}(\boldsymbol{\theta}_t)$ is a non-empty set that satisfies Condition 1(a) and (b) for suitable $\rho_2$. Moreover, with probability at least $1 - \frac{2(L+2)}{m}$, at step $t$ of GD, the FNO loss $\mathcal{L}$ (6) satisfies equation (8) with*

$$\alpha_t = 2\kappa^2 \|\nabla_{\boldsymbol{\theta}} \bar{G}_t\|_2^2 - \frac{c_1}{\sqrt{m}} ,\tag{14}$$

*where $\nabla_{\boldsymbol{\theta}} \bar{G}_t = \frac{1}{n} \sum_{i=1}^n \frac{1}{R} \sum_{j=1}^R \nabla_{\boldsymbol{\theta}} G_{\boldsymbol{\theta}_t}(u^{(i)})(x_j)$, and for some constant $c_1 > 0$ which depends polynomially on the depth $L$, and the radii $\rho_w$, $\rho_r$, $\rho_1$, and $\rho_2$ whenever $\sigma_{1,w} + \sigma_{1,r} \leq 1 - \frac{\rho_w + \rho_r}{\sqrt{m}}$. Thus, the loss $\mathcal{L}(\boldsymbol{\theta})$ satisfies RSC w.r.t $(B_\kappa^t, \boldsymbol{\theta}_t)$, i.e., Condition 1(c), whenever $\|\nabla_{\boldsymbol{\theta}} \bar{G}_t\|_2^2 = \Omega(\frac{1}{\sqrt{m}})$.*

*Proof.* We start by proving the first part of the theorem's statement. We immediately see that, since $B_\kappa^t \subset B_{\rho_w, \rho_r \rho_1}^{\mathrm{Euc}}(\boldsymbol{\theta}_0)$, we satisfy Condition 1(a). We now need to satisfy Condition 1(b). We particularly focus on Condition 1(b.2). For this, we proceed to show the existence of an element $\boldsymbol{\theta}' \in B_\kappa^t$ that is an element of the set $Q_\kappa^t$ as in Definition 3, i.e., satisfies

$$|\cos(\boldsymbol{\theta}' - \boldsymbol{\theta}_t, \nabla_{\boldsymbol{\theta}} \bar{G}_t)| \geq \kappa ,\tag{82}$$

and that also satisfies the following two conditions:

Condition (A): $\|\boldsymbol{\theta}' - \boldsymbol{\theta}_t\|_2 = \epsilon$ for some $\epsilon < \frac{2\|\nabla_{\boldsymbol{\theta}} \mathcal{L}(\boldsymbol{\theta}_t)\|_2 \sqrt{1-\kappa^2}}{\beta}$; and

Condition (B): the angle $\nu'$ between $(\boldsymbol{\theta}' - \boldsymbol{\theta}_t)$ and $-\nabla_{\boldsymbol{\theta}} \mathcal{L}(\boldsymbol{\theta}_t)$ is acute, so that $\cos(\nu') > 0$.

To show the existence of such element $\boldsymbol{\theta}' \in B_\kappa^t$, we propose two possible constructions:

Choice (A): If the points $\boldsymbol{\theta}_{t+1}$, $\nabla_{\boldsymbol{\theta}} \bar{G}_t + \boldsymbol{\theta}_t$, and $\boldsymbol{\theta}_t$ are not collinear, then they define a hyperplane $\mathcal{P}$ that contains the vectors $\nabla_{\boldsymbol{\theta}} \bar{G}_t$ and $-\nabla_{\boldsymbol{\theta}} \mathcal{L}(\boldsymbol{\theta}_t)$ (recall that $\boldsymbol{\theta}_{t+1} - \boldsymbol{\theta}_t = -\nabla_{\boldsymbol{\theta}} \mathcal{L}(\boldsymbol{\theta}_t)$ by gradient descent). We choose $\boldsymbol{\theta}'$ such that the vector $\boldsymbol{\theta}' - \boldsymbol{\theta}_t$ lies in $\mathcal{P}$ with $\cos(\boldsymbol{\theta}' - \boldsymbol{\theta}_t, \nabla_{\boldsymbol{\theta}} \bar{G}_t) = \kappa$ (i.e., it satisfies condition (82) with equality) while simultaneously satisfying Condition (B). If the points $\boldsymbol{\theta}_{t+1}$, $\nabla_{\boldsymbol{\theta}} \bar{G}_t + \boldsymbol{\theta}_t$, and $\boldsymbol{\theta}_t$ are collinear, we choose $\boldsymbol{\theta}'$ such that it is not collinear with these points, thus defining a hyperplane $\mathcal{P}$ with these other three points, and such that $\boldsymbol{\theta}'$ is also taken so that $\cos(\boldsymbol{\theta}' - \boldsymbol{\theta}_t, \nabla_{\boldsymbol{\theta}} \bar{G}_t) = \kappa$ while simultaneously satisfying Condition (B).

Thus far we have only defined *angle* (or *direction*) conditions on the vector $\boldsymbol{\theta}' - \boldsymbol{\theta}_t$, and so there could be an infinite number of values for $\boldsymbol{\theta}'_f$ satisfying such angle conditions without $\boldsymbol{\theta}'$ belonging to the set $B_{\rho_w, \rho_r, \rho_1}^{\mathrm{Euc}}(\boldsymbol{\theta}_0)$ nor $\boldsymbol{\theta}'$ satisfying Condition (A). To determine the feasible values for $\boldsymbol{\theta}'$, we observe that $\boldsymbol{\theta}_t$ is *strictly inside* the set $B_{\rho_w, \rho_r, \rho_1}^{\mathrm{Euc}}(\boldsymbol{\theta}_0)$ by Assumption 7, and so $\boldsymbol{\theta}'$ can be taken arbitrarily close to $\boldsymbol{\theta}_t$ so that $\boldsymbol{\theta}' \in B_{\rho_w, \rho_r, \rho_1}^{\mathrm{Euc}}(\boldsymbol{\theta}_0)$ and Condition (A) is satisfied.

We remark that, regardless of the collinearity of the points $\boldsymbol{\theta}_{t+1}$, $\nabla_{\boldsymbol{\theta}} \bar{G}_t + \boldsymbol{\theta}_t$, and $\boldsymbol{\theta}_t$, hyperplane $\mathcal{P}$ contains the vectors $\boldsymbol{\theta}' - \boldsymbol{\theta}_t$, $\nabla_{\boldsymbol{\theta}} \bar{G}_t$, and $-\nabla_{\boldsymbol{\theta}} \mathcal{L}(\boldsymbol{\theta}_t)$, all sharing its origin at $\boldsymbol{\theta}_f \in \mathcal{P}$.

Choice (B): We choose $\boldsymbol{\theta}'$ as in Choice (A) but with $\nabla_{\boldsymbol{\theta}} \bar{G}_t$ replaced by $-\nabla_{\boldsymbol{\theta}} \bar{G}_t$.

We immediately notice that $\boldsymbol{\theta}'$ defined by either Choice (A) or Choice (B) satisfies $\boldsymbol{\theta}' \in Q_\kappa^t \cap B_{\rho_w, \rho_r, \rho_1}^{\mathrm{Euc}}(\boldsymbol{\theta}_0)$. To make $\boldsymbol{\theta}'$ belong to the set $B_\kappa^t$, we need to find a radius $\rho_2$ such that $\boldsymbol{\theta}' \in B_{\rho_w, \rho_r, \rho_1}^{\mathrm{Euc}}(\boldsymbol{\theta}_0)$, which is done by taking $\rho_2 > \epsilon$ with $\epsilon$ as in Condition (A). Finally, it is straightforward to verify that such $\boldsymbol{\theta}' \in B_\kappa^t$ defined by either Choice (A) or Choice (B) will always exist, by considering the following cases for the angle $\nu$ between $\nabla_{\boldsymbol{\theta}} \bar{G}_t$ and $-\nabla_{\boldsymbol{\theta}} \mathcal{L}(\boldsymbol{\theta}_t)$:

(i) If $\nu \in [0, \pi/2]$ or $\nu \in [3\pi/2, 2\pi]$, then Choice (A) will be true, since $-\nabla_{\boldsymbol{\theta}_f} \mathcal{L}(\boldsymbol{\theta}_t)$ is in the positive half space[2] of $\nabla_{\boldsymbol{\theta}} \bar{G}_t$; and

---

[2] We say $\mathbf{a}$ is in the positive half-space of $\mathbf{b}$ if $\langle \mathbf{a}, \mathbf{b} \rangle \geq 0$.

(ii) if $\nu \in [\pi/2, \pi]$ or $\nu \in [\pi, 3\pi/2]$, then Choice (B) will be true, since $-\nabla_{\boldsymbol{\theta}_f} \mathcal{L}(\boldsymbol{\theta}_t)$ is in the positive half space of $-\nabla_{\boldsymbol{\theta}} \bar{G}_t$.

Now, let us assume we are in the case of item (i) above, so that $\boldsymbol{\theta}'$ is constructed according to Choice (A) (the rest of the proof can be adapted to the case of Choice (B) and to item (ii) by using a symmetrical argument and so it is omitted). Let $\nu_1$ be the angle between $\boldsymbol{\theta}' - \boldsymbol{\theta}_t$ and $\nabla_{\boldsymbol{\theta}} \bar{G}_t$, so that $\cos(\nu_1) = \kappa$ according to Choice (A). Then, we have that

$$|\cos(\nu')| = |\cos(\nu - \nu_1)| \geq |\cos(\pi/2 - \nu_1)| = |\sin(\nu_1)| = \sqrt{1 - \cos^2(\nu_1)} = \sqrt{1 - \kappa^2} .$$

Further, by the construction in Condition (B), $\cos(\nu') > 0$, which implies $\cos(\nu') \geq \sqrt{1 - \kappa^2} > 0$. Now, by the smoothness property of the empirical loss $\mathcal{L}$ we have

$$
\begin{aligned}
\mathcal{L}(\boldsymbol{\theta}') &\leq \mathcal{L}(\boldsymbol{\theta}_t) - \langle \boldsymbol{\theta}' - \boldsymbol{\theta}_t, -\nabla_{\boldsymbol{\theta}} \mathcal{L}(\boldsymbol{\theta}_t) \rangle + \frac{\beta}{2} \|\boldsymbol{\theta}' - \boldsymbol{\theta}_t\|_2^2 \\
&= \mathcal{L}(\boldsymbol{\theta}_t) - \|\boldsymbol{\theta}' - \boldsymbol{\theta}_t\|_2 \|\nabla_{\boldsymbol{\theta}} \mathcal{L}(\boldsymbol{\theta}_t)\|_2 \cos(\nu') + \frac{\beta}{2} \|\boldsymbol{\theta}' - \boldsymbol{\theta}_t\|_2^2 \\
&= \mathcal{L}(\boldsymbol{\theta}_t) - \epsilon \|\nabla_{\boldsymbol{\theta}} \mathcal{L}(\boldsymbol{\theta}_t)\|_2 \cos(\nu') + \frac{\beta}{2} \epsilon^2 \\
&\leq \mathcal{L}(\boldsymbol{\theta}_t) - \epsilon \|\nabla_{\boldsymbol{\theta}_f} \mathcal{L}(\boldsymbol{\theta}_t)\|_2 \sqrt{1 - \kappa^2} + \frac{\beta}{2} \epsilon^2 \\
&= \mathcal{L}(\boldsymbol{\theta}_t) - \frac{\beta \epsilon}{2} \left( \frac{2 \|\nabla_{\boldsymbol{\theta}} \mathcal{L}(\boldsymbol{\theta}_t)\|_2 \sqrt{1 - \kappa^2}}{\beta} - \epsilon \right) \\
&< \mathcal{L}(\boldsymbol{\theta}_t) .
\end{aligned}
$$

where the last inequality follows by the construction of $\epsilon$ in Condition (A). Thus, we have $\mathcal{L}(\boldsymbol{\theta}') < \mathcal{L}(\boldsymbol{\theta}_t)$, which implies that the constructed $\boldsymbol{\theta}'$ is as described in Condition 1(b.2). This finishes the proof for the first part of the theorem's statement.

The second part of the proof, i.e., the RSC condition over the non-empty set $B_\kappa^t$, follows from a direct adaptation of Theorem 5.1 in (Banerjee et al., 2023a) using Lemma 4. Since we are using Lemma 4, the condition for polynomial dependence on the bounds carries on. $\qquad\square$

**Theorem 5 (Smoothness for FNOs).** *Under Assumptions 5 and 6, with probability at least $1 - \frac{2(L+2)}{m}$, the FNO loss $\mathcal{L}$ (6) is $\beta$-smooth in $B_{\rho_w, \rho_r \rho_1}^{\mathrm{Euc}}(\boldsymbol{\theta}_0)$ with $\beta$ being a positive constant which depends polynomially on the depth $L$, and the radii $\rho_w$, $\rho_r$, and $\rho_1$ whenever $\sigma_{1,w} + \sigma_{1,r} \leq 1 - \frac{\rho_w + \rho_r}{\sqrt{m}}$.*

*Proof.* The proof follows from a direct adaptation of the proof of Theorem 5.2 in (Banerjee et al., 2023a) using Lemma 4, where it can be shown that $\beta = 2\varrho^2 + \frac{\bar{c}}{\sqrt{m}}$ for some positive constant $\bar{c}$ which inherits the dependence on the constants $\sigma_{1,w}$, $\sigma_{1,r}$, the depth $L$ and the radii $\rho_w$, $\rho_r$, and $\rho_1$ from Lemma 4. $\qquad\square$

**Proposition 3 (RSC to smoothness ratio).** *Under the same conditions as in Theorems 4 and 5, we have that $\alpha_t / \beta < 1$ with probability at least $1 - \frac{2(L+2)}{m}$.*

*Proof.* From the direct adaptation of the proof of Theorem 5.2 in (Banerjee et al., 2023a) using Lemma 4, we can obtain $\|\nabla_{\boldsymbol{\theta}} \bar{G}_t\|_2^2 \leq \varrho^2$. Then, $\alpha_t \overset{(a)}{<} 2\kappa^2 \|\nabla_{\boldsymbol{\theta}} \bar{G}_t\|_2^2 \leq 2\kappa^2 \varrho^2 \leq 2\varrho^2 \overset{(b)}{<} \beta$, where (a) follows from (14) and (b) from Theorem 5. This result shows that $\frac{\alpha_t}{\beta} < 1$. $\qquad\square$

# F. Supplementary Information for the Experiments

In this section we expand on the mathematical description of each operator learning problem studied in Section 8. We also present further results on how the accuracy of each neural operator model improves as the width $m$ increases. Finally, we provide details about the hyperparameters and datasets used in the training of the corresponding models.

The associated code for the experiments in Section 8 and the ones presented in this appendix are found in https://github.com/bhaveshshrimali/neuralop_optimization.

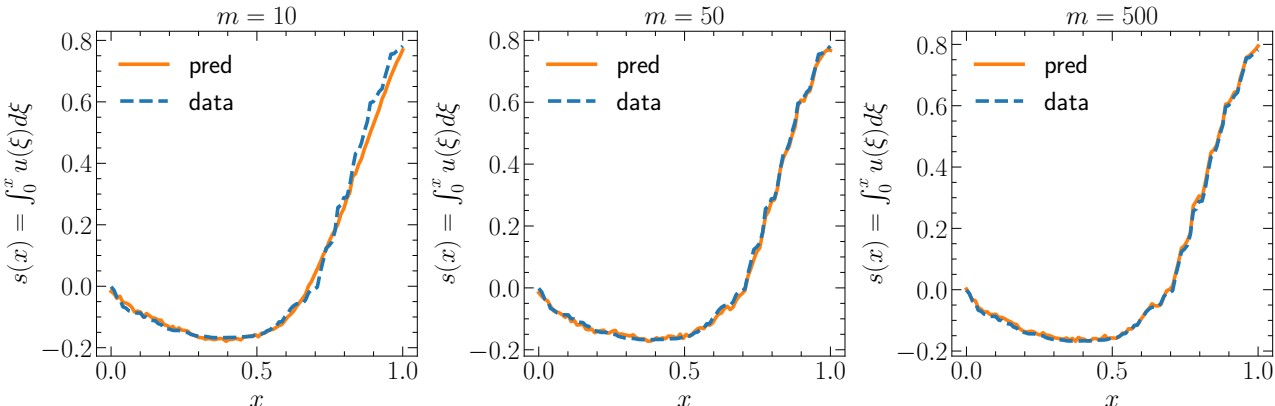

Figure 5: Sample solutions obtained for the Antiderivative operator for DONs for $m \in \{10, 50, 500\}$ at the end of the training process (80,000 epochs) for a randomly chosen input function. The "data" refers to the ground truth (obtained by a standard numerical solver) and "pred" corresponds to the learned operator.

We remark that all experiments with widths $m \in \{10, 50\}$ were run on a personal computer with one NVIDIA Quadro GPU, while the rest of widths were on Google Colab with single NVIDIA L4 and A100 GPUs.

### F.1. Antiderivative Operator

We consider a simple one-dimensional Antiderivative or Integral operator given by

$$s(x) := G(u)(x) = \int_0^x u(\xi) \, d\xi, \qquad x \in [0, 1] \,. \tag{83}$$

Note that $G(u)$ is a linear operator and therefore learnable up to high accuracy. This is evident from the training loss in Figure 1 as well as from the sample solutions presented in Figure 5 for DONs and in Figure 6 for FNOs. We observe that overall an increase in the width $m$ leads to higher training accuracy and lower training loss.

The sample size of the training data is $n = 2000$, with every input function $u^{(i)}$, $i \in [n]$, being a one-dimensional Gaussian Random Fields (GRF). For DON training, we choose $R = 100$ input locations and we choose 100 output locations for each input function, i.e., $q_i = 100$, $i \in [n]$ (according to the notation in Section 3.1).[3] For the FNO, the input function is also sampled across 100 locations (i.e., $\bar{R} = 100$ using the notation in Section 6); however, since we are interested in the model to provide an output of 100 output locations, we modify the FNO architecture to provide this vector-valued output. [4] For all the experiments we fix the learning rate for the Adam optimizer at $10^{-3}$ and with full-batch training, i.e., the batch size of 2000 for both DONs and FNOs. For testing the trained neural operators, we generate another one-dimensional GRF.

### F.2. Diffusion-Reaction Operator

We are interested in learning an operator $G : u(x) \to s(x, t)$ for the solution operator of the one-dimensional Diffusion-Reaction equation implicitly given by

$$\frac{\partial s}{\partial t} = D\frac{\partial^2 s}{\partial x^2} + ks^2 + u(x), \quad (x, t) \in (0, 1] \times (0, 1] \,, \tag{84}$$

---

[3]During training; however, for each $i \in [n]$, instead of averaging the loss over all the $q_i$ points, we simply randomly choose one of the $q_i$ points and evaluate the loss on it. This is strictly done in the interest of computational efficiency, since it is known to not reduce the accuracy of the results for the Antiderivative operator; e.g., see (Lu et al., 2021).

[4]This means that we have $R = 1$ (according to the notation in Section 3.2) with the understanding that for each input function, we output a vector of size 100. This is done as an alternative to an FNO with a scalar output which is averaged across the 100 locations (for which we would have $R = 100$), which is what we described in Section 3. We remark that we considered this modification on the output of the FNO just for the sake of computational efficiency, and this only empirically works for the case of the Antiderivative operator. Again, as in the case of DONs, we randomly sample one of the output locations to compute the loss during training.

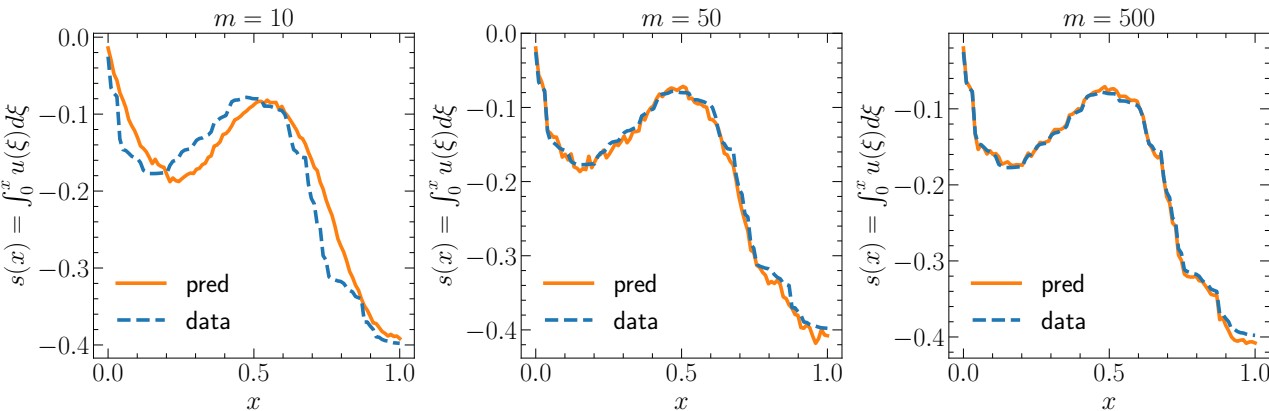

Figure 6: Sample solutions obtained for the Antiderivative operator for FNOs for $m \in \{10, 50, 500\}$. The setting is the same as in Figure 5.

with $D > 0$ and zero initial and boundary conditions, namely,

$$s(0, t) = s(1, t) = 0 \quad \text{and} \quad s(x, 0) = 0 \,,$$

along with a forcing function $u(x)$ defined by a GRF. This is the same setup as in (Wang et al., 2021a; Lu et al., 2021). The corresponding solutions for DONs and FNOs are presented in Figures 7 and 8 respectively. Again, a larger width $m$ leads to a more accurate solution.

The neural operator aims to learn a mapping from the forcing function to the solution at different times in the interval $(0, 1]$, in other words, the forcing function would be the *input function* as defined in Section 3. We make use of a slightly modified solver provided at https://github.com/PredictiveIntelligenceLab/Physics-informed-DeepONets to generate the training data for the equation. We generate solutions for $n = 5000$ input functions which are sampled on 100 points in the space dimension (i.e., the interval $(0, 1]$ for $x$ in (84) is divided in 100 points) so that $R = 100$ for DON and $\bar{R} = 100$ for FNO (according to the notations in Section 3.1 and Section 6 respectively). For computing the solutions, we are interested in computing them at 100 different times $t$ within the time interval $(0, 1]$ in (84) (in order to be able to plot the two-dimensional map on $x$ and $t$ in Figures 7 and 8). This division of both spatial and time dimensions results in a grid of $10,000$ points that can be chosen as output locations. For the training of DONs, for each $i \in [n]$, we only select 100 scattered points from the grid of output locations (out of their $10,000$ points), so that $q^{(i)} = 100$, which will become the input to the trunk net. However, for the training of FNOs, we do choose the full grid as output locations and thus we modify the FNO to provide $10,000$ outputs instead of the scalar output provided in our theoretical analysis.[5] We fix the diffusivity as $D = 0.01$. For all the experiments we use a constant learning rate of $3 \times 10^{-4}$ and Adam optimizer with a batch size of 4000. For testing the trained neural operators, we generate another one-dimensional GRF.

### F.3. Burgers' Equation

The Burgers' equation operator learns an operator $G : u(x) \to s(x, 1)$, where

$$\frac{\partial s}{\partial t} + s\frac{\partial s}{\partial x} - \nu\frac{\partial^2 s}{\partial x^2} = 0, \quad (x, t) \in (0, 2\pi] \times (0, 1]$$
$$s(x, 0) = u(x), \quad x \in (0, 2\pi] \tag{85}$$

with $\nu > 0$ and periodic boundary conditions

$$s(0, t) = s(2\pi, t)$$
$$\frac{\partial s}{\partial x}(0, t) = \frac{\partial s}{\partial x}(2\pi, t) \,.$$

---

[5]A scalar output is needed if we were interested in evaluating the operator at only one specific spatial location $x$ and one specific value of time $t$; however, as can be seen in Figures 7 and 8, we are interested in plotting solutions at multiple locations and times.

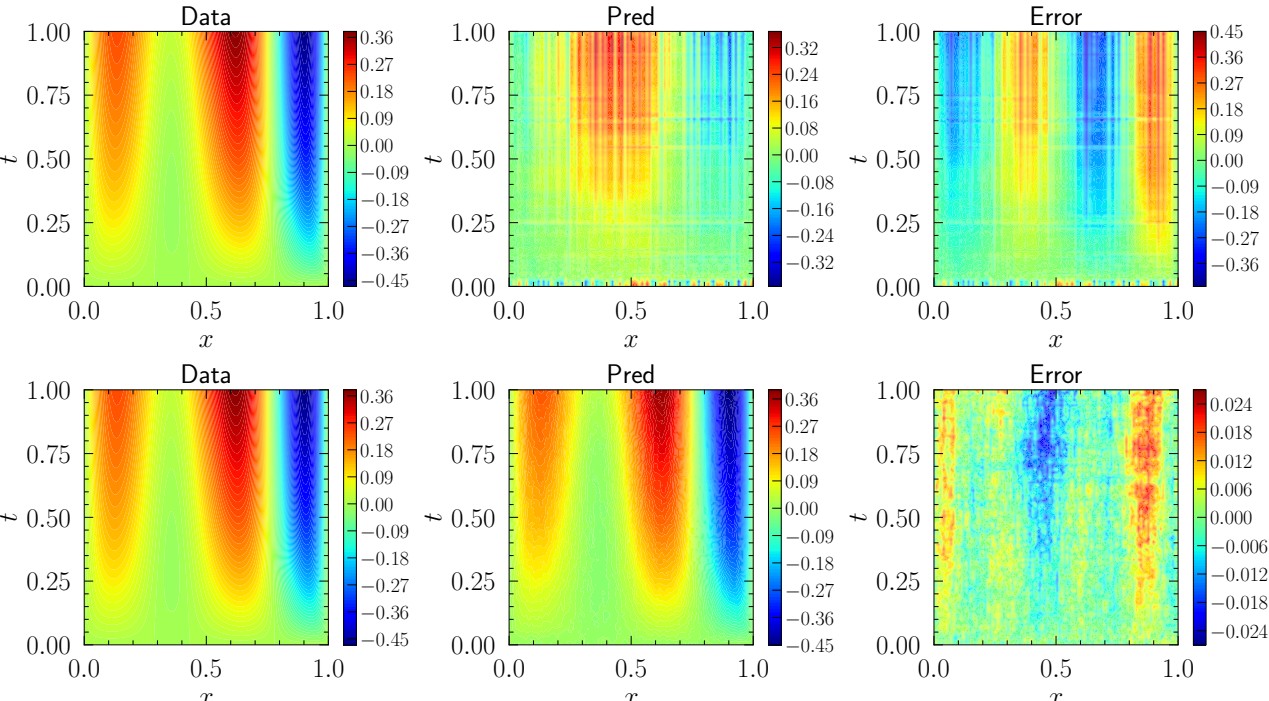

Figure 7: Sample solutions $s(x, t)$ obtained for the Diffusion-Reaction operator for DONs for $m \in \{10, 500\}$ given an input $u(x)$. The top row corresponds to $m = 10$ and the bottom row to $m = 500$. The third column represents the pointwise difference of the ground truth or "Data" (first column) minus the obtained results from the learned DON or "Pred" (second column).

The corresponding solutions for DONs and FNOs are presented in Figures 9 and 10 respectively. Again, a larger width $m$ leads to a more accurate solution.

The neural operator aims to learn a mapping from the initial condition to the solution at time $t = 1$, i.e. the mapping from $u(x)$ to the final solution $s(x, 1)$. This is the operator learning problem originally studied in (Li et al., 2021a). We note that the initial condition would then be the *input function* as defined in Section 3. We make use of the datasets publicly available at https://github.com/neuraloperator/neuraloperator, specifically the Burgers_R10.mat dataset available at https://drive.google.com/drive/folders/1UnbQh2WWc6knEHbLn-ZaXrKUZhp7pjt-, which comprises of 2048 input functions and corresponding final solution (i.e., $u^{(i)}$ with associated solution $s^{(i)}(\cdot, 1)$, $i \in [2048]$). All solutions are calculated for a single viscosity $\nu = 0.01$. For all the experiments we use a constant learning rate of $10^{-3}$ and Adam optimizer with a batch size of 800. We test the trained neural operators on a simple GRF sampled from the training dataset.

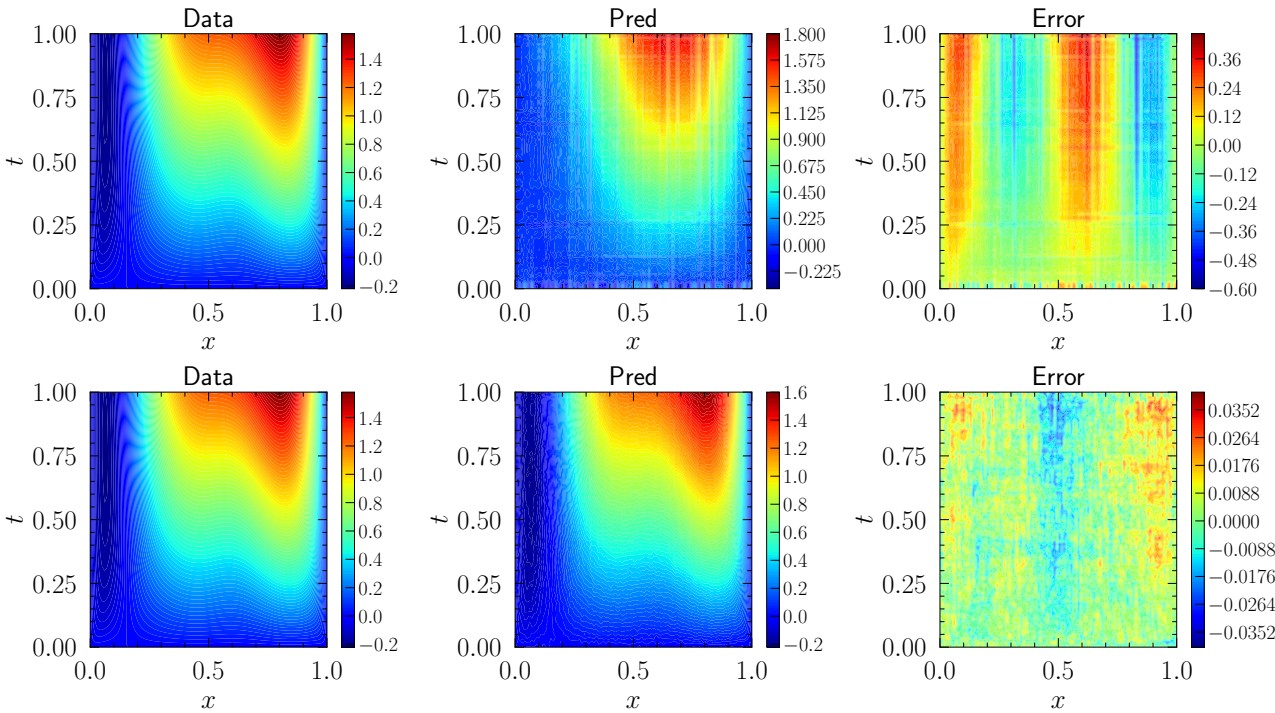

Figure 8: Sample solutions obtained for the Diffusion-Reaction operator $s(x, t)$ for FNOs for $m \in \{10, 500\}$. The setting of the plots is the same as in Figure 7 where the top row corresponds to $m = 10$ and the bottom row to $m = 500$.

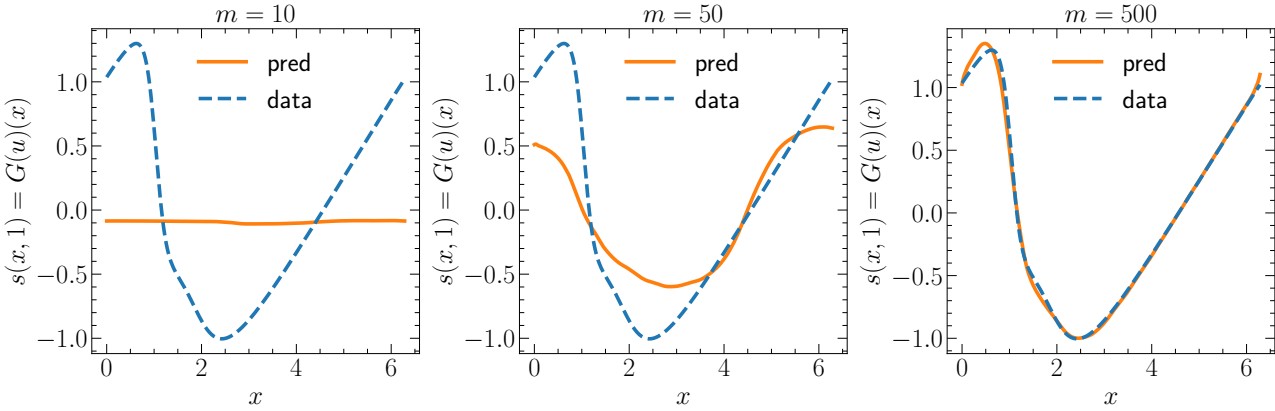

Figure 9: Sample solutions obtained for the Burgers' equation for DONs for $m \in \{10, 50, 500\}$. The setting is similar to the one in Figure 5 where we plot the obtained solution from the learned operator (denoted by "pred") along with the ground truth (denoted by "data") for different widths.

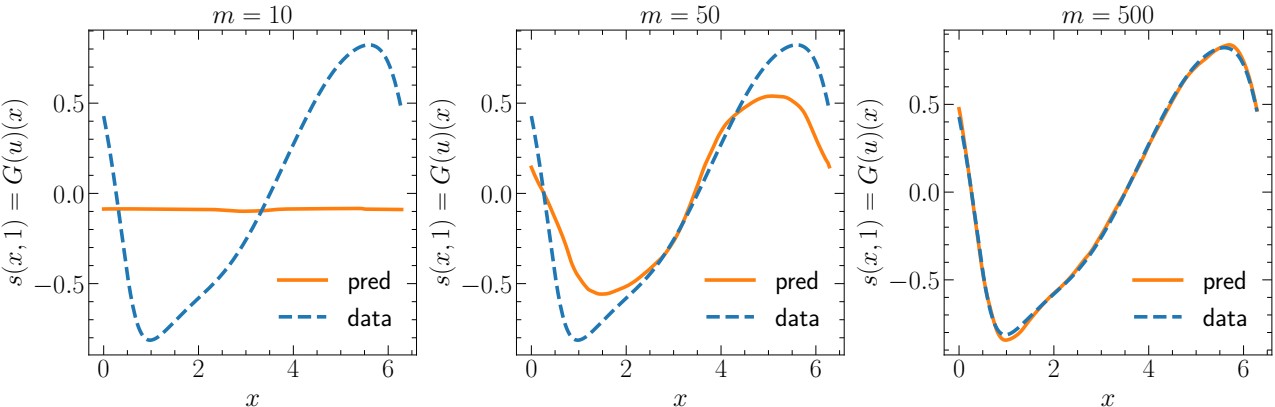

Figure 10: Sample solutions obtained for the Burgers equation for FNOs for $m \in \{10, 50, 500\}$. The setting is the same as in Figure 9.

