# OpenReview forum: "Optimization for Neural Operators can Benefit from Width"
_ICML.cc/2025/Conference — ICML 2025 poster_

### Official Review · Reviewer_bZNF · 2025-03-10

**Overall Recommendation:** 3

**Summary:**

This paper proposes a unified optimization framework using Restricted Strong Convexity (RSC) and smoothness to establish gradient descent convergence guarantees for Deep Operator Networks (DONs) and Fourier Neural Operators (FNOs). The key contributions are as follows:
1. A theoretical proof that the empirical losses for both operators satisfy RSC and smoothness under over-parameterization (Theorems 2-5).
2. A demonstration that wider networks improve optimization convergence, supported by both theoretical analysis and experiments.
3. Empirical validation on three operator learning tasks: antiderivative, diffusion-reaction, and Burgers’ equation.

**Claims And Evidence:**

The authors demonstrate that increasing network width benefits optimization, supported by Hessian/gradient bounds (Theorems 2-3 for DONs and 4-5 for FNOs) and loss reduction trends (Figures 1-2). However, the local loss descent established in Theorems 2 and 4 does not necessarily imply reduced optimization difficulty with increasing network width. Additionally, there are technical concerns in the RSC proof, particularly in the handling of interaction terms.

**Essential References Not Discussed:**

This work lacks a discussion of concurrent work on operator-specific optimization (e.g., Qin et al., 2024 on Fourier spectral improvements).

**Ethical Review Concerns:**

No ethical concerns identified. This work focuses on the theoretical analysis of existing methods.

**Experimental Designs Or Analyses:**

The experimental results clearly show that increasing width improves optimization performance. However, experiments should also be used to validate the key assumptions (Assumptions 4 and 7) to ensure their practical relevance and applicability.

**Methods And Evaluation Criteria:**

This study proposes a unified theoretical and experimental framework to explore the impact of network width on optimization. While empirical results validate the benefits of width expansion, the theoretical analysis relies on overly restrictive assumptions that limit its generalizability. Furthermore, the practical feasibility of these assumptions is insufficiently validated.

**Other Comments Or Suggestions:**

No other comments. Just see "Strengths And Weaknesses" and "Questions".

**Other Strengths And Weaknesses:**

Strengths

Originality: While the paper provides an RSC-based convergence proof for neural operators, much of the theoretical framework closely follows prior work, particularly Banerjee et al. (2023a), with limited novel extensions specific to neural operators.

Significance: Offers practical insights into how network width impacts optimization, which could guide applications in scientific computing.

Weaknesses

Clarity: Proof sketches in Appendix C.2 lack intuitive explanations of cross-network interactions, making the derivations difficult to follow.

Experiment diversity: All tasks rely on L2 loss, and the absence of adversarial or uncertainty-aware metrics limits the scope of the evaluation.

Theoretical novelty: The theoretical contributions are incremental, as the RSC proof heavily builds on Banerjee et al. (2023a) with minimal adaptation to the specific challenges of neural operators.

**Questions For Authors:**

1. There are some issues in the proof of Theorem 2, where the authors aim to demonstrate that $Q^t_\kappa$ is non-empty, characterized by the following conditions:
$| \cos(\theta' - \theta_t, \nabla_\theta \tilde{G}_{\theta_t}) | \geq \kappa$ (cosine similarity condition),

$(\theta_f' - \theta_{f,t})^\top
\left(\frac{1}{n} \sum_{i=1}^n \frac{1}{q_i} \sum_{j=1}^{q_i} \ell_{i,j}'
\sum_{k=1}^K \nabla_{\theta_f} f_k^{(i)} \nabla_{\theta_g} g_{k,j}^{(i)\top}\right)
(\theta_g' - \theta_{g,t}) \geq 0.$

$(\theta_f' - \theta_{f,t})^\top
\left(
\sum_{k=1}^K \nabla_{\theta_f} f_k^{(i)} \nabla_{\theta_g} g_{k,j}^{(i)\top}
\right)
(\theta_g' - \theta_{g,t}) \leq 0, \quad
\forall i \in [n], \forall j \in [q_i].
$

To simplify the analysis, the authors set $\theta_g' = \theta_{g,t}$ and claim that belonging to the $Q^t_\kappa$  set conveniently reduces to the feasibility of the cosine similarity condition as follows:

$\left| \cos(\theta_f' - \theta_{f,t}, \bar{g}_f) \right| \geq \kappa.  $

However, in this case

$\left| \cos(\theta' - \theta_t, \nabla_\theta \tilde{G}_{\theta_t}) \right| =
\frac{\langle \theta_f' - \theta_{f,t}, \bar{g}_f \rangle}{\| \theta_f' - \theta_{f,t} \| \| \nabla_\theta \tilde{G}_{\theta_t} \|}$

which is not equivalent to $\left| \cos(\theta_f' - \theta_{f,t}, \bar{g}_f) \right|$ without additional control
over $\bar{g}_f$.

2. Assumptions 2 and 4 appear to be overly strong. In the convergence analysis of NTK, such assumptions are not required. As this
work aims to extend NTK theory, it would be valuable to theoretically or empirically validate these assumptions. One feasible approach could be
to examine the norm of the training trajectory during the neural network training process. This would provide insights into whether these assumptions hold in practice and help justify their necessity in the proposed framework.

**Relation To Broader Scientific Literature:**

The paper connects to NTK theory (Jacot et al., 2018) and RSC analysis (Banerjee et al., 2023a).

**Theoretical Claims:**

Upon reviewing the proofs related to the DON section, we identified issues in the construction of the RSC proof. Specifically, certain steps in the derivation lack sufficient justification, raising concerns about the validity of the argument. Additional clarification or further details are necessary to ensure the correctness of the proof.

---

> ### Author Rebuttal · Authors · 2025-03-31
>
> We are grateful to the reviewer for taking the time to review our paper.
>
> ---
>
> We start by responding to the reviewer's concerns.
>
> - We respectfully state that our work is not a "limited novel extension" of (Banerjee et al., 2023a): for a detailed justification we refer to our response to Reviewer bY3J, where we justify our claim on three facts:
>   1. we present a more general optimization framework that is used by both (Banerjee et al., 2023a) and our work;
>   2. several works in the literature share the same underlying mathematical approach and are not considered "limited novel extensions" of each other due to their different models; and
>   3. our models have differences that make their analysis complex and non-trivial (please see the details in Appendices D (for DONs) and E (for FNOs)), without automatically following from (Banerjee et al., 2023a).
>
> - Regarding our "experimental diversity", we stated in our paper that the objective of Section 8 "is to show the effect of over-parameterization on the neural operator training and not to present any kind of comparison between the two neural operators." We consider that our experiments already achieve the goal of illustrating and complementing the theory of our paper, and thus, there is no need to include additional performance metrics such as the ones mentioned by the reviewer. Indeed, since our analysis is about gradient descent over the **empirical loss**, which is an **L2 loss**, it is sufficient to study the behavior of this L2 loss to prove the benefits of width during training. Finally, since our theoretical results are the centerpiece of our work, having straightforward yet concrete empirical findings that support the theory helps to ensure that the focus remains firmly on the paper's core theoretical contributions.
>
> - We are grateful to the reviewer for suggesting more discussion on concurrent works including (Qin et al., 2024); we will include them in Appendix A.
>
> ---
>
> We now proceed to respond to the questions about our paper.
>
> **Question 1**:
>
> We are grateful to the reviewer for pointing out this issue. Indeed, as pointed out by the reviewer, the formula in equation (28) of Appendix D.2 (proof of Theorem 2) should be $|\cos(\theta' -  \theta_{t}, \nabla_{\theta}\\bar{G}\_{\\theta_{t}}) |\\geq \\kappa$ with the understanding that
> $\theta'=[{\theta'\_f}^{\top}\\;{\theta\_{g,t}}^{\top}]^{\top}$. Then, the rest of the proof is **still correct** since it becomes virtually equivalent to the proof of non-emptyness of the $Q^t_\kappa$ set for FNOs (Appendix E.2). This follows from the fact that the $Q^t_\kappa$ set for FNOs only depends on the *cosine similarity condition*, which is similar to the reduction  (28) we obtained for FNOs by taking $\theta'$ as above.
> In conclusion, our proof **still holds** after correcting equation (28) and appropriately changing the notation, following the same proof as the one used for FNOs. Again, thanks to the reviewer for spotting issue.
>
> **Question 2**:
>
> We are grateful for the suggested ways to strengthen the practical validation of our assumptions. We start by respectfully clarifying three things:
>   1. Our work's aim is not to "extend NTK theory" since we use a different mathematical approach (namely, RSC theory). Indeed, Section 2 proposes an alternative optimization framework to the NTK one.
>   2. We must emphasize that Assumption 2 and part of Assumption 4 **are actually found** in the NTK-based paper (Liu et al., 2021a).
>   3. We respectfully believe that Assumption 2 is not "overly strong" for *two* reasons. First, it is satisfied for *commonly used* smooth activation functions such as sigmoid, hyperbolic tangent, and Gaussian Error Linear Unit (GELU). Second, besides being used by the NTK-based paper mentioned above, it is used by all RSC-based papers mentioned in Section 2.
>
> We now discuss the empirical validation of Assumption 4 (Assumption 2 is not to be verified empirically since it is satisfied by the design choice of the activation functions). Assumption 4 requires *all* iterations to be within a neighborhood of the initialization point. This could certainly be validated in the way suggested by the reviewer—the task would be to find the $\rho$ and $\rho_1$ radii, which we suspect will depend on both the initialization point and the training data (training is a non-convex problem and its optimization landscape depends on the training data). Nonetheless, Assumption 4 is *only* required if we want our results to hold for *all* iterations of gradient descent—indeed, we could state a weaker version of Assumption 4 where it only holds for some *finite* set of iterations, and still our guarantees would hold for such iterations. Thus, even if we find that Assumption 4 is not satisfied, it is still possible that our theoretical conditions were met for some iterations along the training procedure.
>
> ---
>
> We hope that this rebuttal leads to a more positive assessment of our paper.

---

### Official Review · Reviewer_qXNc · 2025-03-13

**Overall Recommendation:** 3

**Summary:**

This paper addresses the  problem of optimization convergence guarantees for neural operators, specifically Deep Operator Networks (DONs) and Fourier Neural Operators (FNOs), when trained using gradient descent (GD). The authors propose a unified optimization framework based on two key conditions: restricted strong convexity (RSC) and smoothness of the loss function. They demonstrate that these conditions are satisfied for both DONs and FNOs, particularly when the networks are wide. The paper provides theoretical guarantees for the convergence of GD in training these neural operators and supports the theory with empirical results on canonical operator learning problems.

**Claims And Evidence:**

Yes

**Essential References Not Discussed:**

No

**Experimental Designs Or Analyses:**

Yes

**Methods And Evaluation Criteria:**

Yes

**Other Comments Or Suggestions:**

No

**Other Strengths And Weaknesses:**

Strengthes:
1. The paper provides the first formal optimization convergence guarantees for DONs and FNOs, addressing a significant gap in the literature.

2. The authors complement their theoretical results with empirical evaluations, demonstrating that wider networks indeed lead to lower training losses and faster convergence for both DONs and FNOs.

Weaknesses:
1. While the empirical results are promising, the experiments are limited to three canonical problems. It would be beneficial to see how the theory holds up on more complex or real-world operator learning tasks

2. The paper does not discuss the practical implications of the theoretical results in detail. For instance, how does the width of the network affect the generalization performance, and what are the trade-offs between width and computational cost?

**Questions For Authors:**

1. The paper suggests that wider networks lead to better optimization convergence. However, wider networks also increase computational cost. How do the authors suggest balancing width and computational efficiency in practice? Why not deeper networks?

2. The paper assumes smooth activation functions. How sensitive are the results to the choice of activation function? Would non-smooth activations (e.g., ReLU) affect the optimization guarantees?

3. Have the authors considered comparing the performance of gradient descent with other optimization methods, such as SGD?

4. The paper focuses on optimization convergence, but how does the width of the network affect the generalization performance of DONs and FNOs? Are there any theoretical or empirical insights on this?

**Relation To Broader Scientific Literature:**

The contributions of the paper are related to scientific machine learning and optimization.

**Theoretical Claims:**

No

---

> ### Author Rebuttal · Authors · 2025-03-31
>
> We are grateful to the reviewer for taking the time to review our paper and for the list of questions which we now address.
>
> - **Question 1**: **First subquestion:** Our theoretical work establishes sufficient conditions for optimization and shows that the convergence rate may benefit from width, i.e., *less* optimization steps may be needed to achieve a lower loss value as the network increases in width. Nevertheless, increasing width also means that the computational cost increases *per* optimization step. We believe that the tradeoff between width and computational cost should be determined empirically depending on both (i) the user's tolerance for error and (ii) the available computational resources for training. Assume a user can tolerate a certain $\epsilon$ value of error on the empirical loss. If the user has limited computational resources and uses a wide network, it could happen that the *time* it takes to do a single optimization step is long, even though there are only a few steps needed to reach the tolerance level $\epsilon$. In such case, the user should opt for using networks with less width, because, even though there may be more optimization steps to reach the tolerance $\epsilon$, the *total* amount of time spent during training could be less. If, on the other hand, there is abundant and better computational resources, the user can now afford to increase the width more and even decrease the tolerance level.
> Now, it could also happen that increasing the width will not help too much, depending on the application. Thus, our recommendation would be to not start training with very large widths. For example, if we look at the simulations in Section 8, it is clear that how much help is attained by incrementing width is largely application driven—jumping from a width of 10 to 50 helps the Diffusion-Reaction problem much more than it does to the Burger's Equation. **Second subquestion:** We believe that the question of choosing between increasing depth and increasing width is delicate. Though this goes beyond the scope of our work, we can mention a few things in response to the question by the reviewer:
>   - (i) justifying the benefits of depth versus width in optimization requires a different theoretical framework than ours;
>   - (ii) adding more layers (depth) also adds more weights and thus more matrix-matrix multiplications in the gradient computations, so the computational benefit (if any) of increasing depth versus width has to be carefully considered; and
>   - (iii) deeper networks may need extra architectural changes to avoid *vanishing* gradient effects (such as adding residual connections, normalization layers, etc.), which needs to be taken into account and whose effect on the training of neural operators needs to be studied.
>
> - **Question 2**: Our results are based on the calculation of the RSC condition and smoothness of the empirical loss, which requires calculating the Hessian of the empirical loss—and for this, we ultimately make use of the differentiability of the smooth activation functions. Thus, due to their lack of (global) differentiability, using non-smooth activation functions such as ReLUs would require a different analysis approach—for example, we may need to formulate an alternative notion to the RSC condition for non-smooth functions, as well as an alternative to the smoothness requirement (perhaps using *semi-smoothness* as in the work (Allen-Zhu et al., 2019) which uses ReLUs). We believe this is an interesting future direction. Finally, we would like to point out that, even though our paper only covers smooth activation functions, it encompasses commonly used activations such as sigmoid, hyperbolic tangent, and Gaussian Error Linear Unit (GELU).
>
> - **Question 3**: This is something we hope could be the topic of future theoretical work—to the best of our knowledge, no work has explored SGD with restricted strong convexity. In order to study such problem, we would need to start by adapting our general optimization framework in Section 2 to the stochastic gradient descent setting.
>
> - **Question 4**: This is beyond the scope of our paper, however, we hypothesize that generalization can also benefit from width for both DONs and FNOs. It is relevant to mention that (Kontolati et al. (2022)) has empirically shown that over-parameterization benefits generalization for DONs, as mentioned in Section 2.
>
> ---
>
> Finally, we would like to mention that, although the problems we study in our experiments may be regarded as "toy problems", they were chosen because they are **representative** of problems typically found in the operator learning literature. Indeed, the seminal papers on both DONs (Lu et al. (2021)) and FNOs (Li et al. (2021)) report results on Burger's equation, and the paper on DONs also reports results on the Antiderivative operator and the Diffusion-Reaction equation.
>
> ---
>
> We hope that this rebuttal leads to a more positive assessment of our paper.

---

### Official Review · Reviewer_sqV3 · 2025-03-13

**Overall Recommendation:** 3

**Summary:**

The main results of this paper are to derive optimization convergence results for both Deep Operator Networks (DONs) and Fourier Neural Operators (FNOs), under gradient descent (GD). The main techniques in this paper are to show that the empirical loss functions for such two kinds of networks satisfy Restricted strong convexity (RSC) and Smoothness conditions under certain assumptions.

**Claims And Evidence:**

The results in this paper are more on the theoretical side. Yes, the claims made in the submission are supported by clear and convincing evidence, such as theoretical derivation.

**Essential References Not Discussed:**

No. The references are enough to understand the key contributions of the paper.

**Ethical Review Concerns:**

No.

**Experimental Designs Or Analyses:**

The results in this paper are more on the theoretical side. Yes, the experimental designs and analyses are sound.

**Methods And Evaluation Criteria:**

The results in this paper are more on the theoretical side. Yes, the proposed methods make sense.

**Other Comments Or Suggestions:**

No.

**Other Strengths And Weaknesses:**

Strengths

This paper derives optimization convergence results of loss functions using GD for Deep Operator Networks (DONs) and Fourier Neural Operators (FNOs), which are also verified by empirical results.

Weaknesses

The main results in this paper, Theorems 2,3,4,5, are about the step t of GD, and are not deterministic results. Thus, although in each step, the loss decreases with a high probability (which requires the width to be very large), it doesn't guarantee that the loss will decrease after a large number of iterations of GD. Therefore, it will be interesting if the authors can derive some convergence results for the whole training procedure.

**Questions For Authors:**

It will be interesting if the authors can derive some convergence results for the whole training procedure.

**Relation To Broader Scientific Literature:**

This paper first derived the optimization convergence result of loss functions using GD for Deep Operator Networks (DONs) and Fourier Neural Operators (FNOs), which extends the previous work in neural operators.

**Theoretical Claims:**

Yes, I checked and thus believe the proofs for theoretical claims in this paper should be correct.

---

> ### Author Rebuttal · Authors · 2025-03-31
>
> We are grateful to the reviewer for taking the time to review our paper. We now address the points raised by the reviewer.
>
> We are grateful for the question of guaranteeing convergence for the whole training procedure. If we understood correctly (and we kindly ask the reviewer to correct us if we have not), the reviewer is concerned that because our results (i) hold with high probability and (ii) require larger widths, they may not hold for the whole training procedure. Although this is a valid concern, we respectfully mention that conditions (i) and (ii) can be found in the existing literature of optimization guarantees for deep models—for example, among the works cited in Section 2 for both neural tangent kernel (NTK) and restricted strong convexity (RSC) approaches.
>
> We also point out that the reason why we end up with condition (i) is because we have to bound the norms of the gradients of the neural operators, which leads to the appearance of weight norms that can only be upper bounded by some constant *with* high probability. In contrast to our work, the work (Cisneros-Velarde et al., 2025) does not need such upper bounds since it studies feedforward neural networks with weight normalization, and so all of its results are deterministic (though condition (ii) is still needed). Finally, we point out that condition (ii) is expressed differently in each optimization approach: in terms of a gradient norm for the RSC approach and in terms of sample size for the NTK approach.
>
> Having shown that the nature of the conditions that ensure our optimization guarantees is not foreign to the existing literature and is intrinsic to our mathematical approach, we hope the reviewer finds our conditions well justified. Having said that, we believe that deriving deterministic convergence results for the whole training procedure is an important problem—a problem which may require further assumptions and even a different mathematical approach.
>
> We hope that all of our arguments presented in this rebuttal lead to a more positive assessment of our paper.

---

### Official Review · Reviewer_bY3J · 2025-03-17

**Overall Recommendation:** 3

**Summary:**

The paper provides convergence guarantees for neural operator learning, which are valid under assumptions of restricted strong convexity and smoothness of the loss function. The authors demonstrate that two learning operators (DON and FNO) satisfy these conditions. Both theoretical and experimental findings show that networks with greater width achieve superior optimization performance.

## update after rebuttal

The authors have partially addressed my concerns, so I'm leaning toward acceptance.

**Claims And Evidence:**

the claims are supported by clear and convincing evidence.

**Essential References Not Discussed:**

--

**Experimental Designs Or Analyses:**

good

**Methods And Evaluation Criteria:**

--

**Other Comments Or Suggestions:**

"Nevertheless, optimization guarantees for DONs is also an open problem". line 076 typo (not DON)

**Other Strengths And Weaknesses:**

**strenghts**
- clear, well-written
- interesting topic

**weaknesses**
- incremental compared to Banerjee et al 2022
- the code is not provided

**Questions For Authors:**

-

**Relation To Broader Scientific Literature:**

This paper appears incremental compared to https://arxiv.org/pdf/2209.15106 Banerjee et al. 2022. The main difference is that they extend the results to neural operator learning instead of feedforward models with smooth activations. Many of the proofs follow a similar structure to the work of Banerjee et al 2022. The paper's primary contribution seems to be verifying that DONs and FNOs satisfy the RSC and smoothness conditions (in Theorems 2-5), which then allows them to directly apply the convergence framework established by Banerjee et al. 2022.

**Theoretical Claims:**

Theoretical claims are convincing.

---

> ### Author Rebuttal · Authors · 2025-03-31
>
> We are grateful to the reviewer for taking the time to review our paper. We now respond to the reviewer's concerns.
>
> ---
>
> We respectfully argue that our work **is not incremental** to (Banerjee et al., 2023a) [using the reference as cited in our work] for three reasons:
>
> 1. Our work **generalizes** the framework by (Banerjee et al., 2023a), as shown in Section 4. Indeed, we show that the results for feedforward neural networks (FFNs) studied by (Banerjee et al., 2023a) and for the neural operators we study (DONs and FNOs) are **particular** instances of our framework presented in Section 4—despite the architectural differences among FFNs, DONs, and FNOs.
>
> 2. Just the fact that our work and (Banerjee et al., 2023a) use a *similar* approach to obtain optimization guarantees (namely, the use of restricted strong convexity (RSC)) **does not imply** that our work is an "extension" of the other. If that were the case, this would be similar to saying that *every* work that uses the Neural Tangent Kernel (NTK) approach to obtain optimization guarantees is an extension of the seminal paper (Jacot et al., 2018). We remark that Section 2 lists several works that are based on the NTK approach, yet they are not considered a mere extension of each other: they study networks with different architectures or activation functions. Therefore, we feel that this is an unfair criticism of our work, and ignores the extensive set of new results and analysis we have presented for FNOs and DONs (see details in Appendices D and E).
>
> 3. Finally, as carefully explained in Section 1 (lines 063 to 081 of the first column of page 2) and Section 7, the **structural differences** between the neural operators (DONs and FNOs) and FFNs lead to a series of **challenges** in their analysis compared to FFNs. These challenges are reflected in our **involved and non-trivial** analysis: in the case of DONs, they stem from the empirical loss Hessian structure being substantially more complex due to the interactions between two neural networks; in the case of FNOs, they stem from the operator's Hessian structure being substantially more complex due to the Fourier transformations inside the activations. Proving that the RSC method still applies to these neural operators despite the architectural differences **do not** automatically follow from any prior work in the literature, including (Banerjee et al., 2023a). Indeed, before our paper, there was no evident reason to anticipate that the RSC method would be general enough to provide optimization guarantees to neural operators.
>
> For all the presented reasons, we respectfully reiterate that our paper is not incremental to (Banerjee et al., 2023a)—please see the details in our Appendices D (for DONs) and E (for FNOs). We remark that, to the best of our knowledge, our work is the first one showing optimization guarantees for operator learning. We are grateful to the reviewer to consider the justification we just provided. We hope our arguments will lead to a more positive assessment of our paper.
>
> ---
>
> Finally, the code is currently sitting in a private repository and we will provide the code used in our experiments through a public repository if our paper is accepted. We will also address the typo raised by the reviewer.

---

### Decision · Program_Chairs · 2025-05-01

**Decision:**

Accept (poster)

**Comment:**

The paper provides valuable theoretical insights into the optimization of Deep Operator Networks (DONs) and Fourier Neural Operators (FNOs), two critical architectures in operator learning. Its main strength lies in developing a unified framework based on Restricted Strong Convexity (RSC) and smoothness conditions, contributing to a deeper understanding of how network width impacts gradient descent convergence. The empirical validation across standard operator learning tasks demonstrates practical improvements with increased network width, aligning theory with observable performance gains.

Whle there are some concerns regarding the rigor and clarity of certain theoretical proofs—particularly related to RSC derivations and the cosine similarity condition, the core theoretical approach remains meaningful and novel enough to merit further consideration in ICML. Given the practical significance of understanding width-dependent optimization behaviors in neural operators, and the potential for meaningful refinements in theoretical presentation and empirical validation, in my opinion, the strengths outweigh the noted weaknesses. Therefore, I recommend acceptance, contingent upon authors adequately addressing the technical and presentation concerns raised.